# Using RKHS Weight Functions in Random Feature Models

**Gabriel Dubé** *gadub44@ulaval.ca*
*Département d'informatique et de génie logiciel*
*Université Laval*

**Mario Marchand** *mario.marchand@ift.ulaval.ca*
*Département d'informatique et de génie logiciel*
*Université Laval*

**Frédéric LeBlanc** *frederic.leblanc@iid.ulaval.ca*
*Institut intelligence et données*
*Université Laval*

**Reviewed on OpenReview:** *https://openreview.net/forum?id=BCNPJ3WRVI*

## Abstract

We examine the consequences of positing that the weight function $\alpha$ in the classical random feature model formulation $f(x) = \mathbb{E}_{w \sim p} [\alpha(w)\phi(w, x)]$ belongs to a reproducing kernel Hilbert space. Depending on the choice of parameters of the random feature model, this assumption grants the ability to exactly calculate the model instead of relying on the random kitchen sinks method of approximation. We present several such examples. Additionally, using this form of the model, the functional gradient of the loss can be approximated in an unbiased way through sampling of the random features. This allows the use of a stochastic functional gradient descent to learn the weight function. We show that convergence is guaranteed under mild assumptions. Further theoretical analysis shows that the empirical risk minimizer converges with the same $\mathcal{O}\left(\frac{1}{\sqrt{m}} + \frac{1}{\sqrt{T}}\right)$ rate in the work of Rahimi & Recht (2009). We also present two other algorithms for learning the weight function. We run experiments to compare our three learning algorithms, and to compare our random feature model variant to the original random kitchen sinks and other state of the art algorithms.

## 1 Introduction

The simplest but not the least important kind of machine learning models consists of linear predictors. They are easy to optimize and deriving theoretical guarantees on their performance is simple. Kernel methods, such as support vector machines (SVMs) (Steinwart & Christmann, 2008), and kernel ridge regression (Vovk, 2013), build upon linear predictors using the kernel trick to produce highly complex predictors. These methods are powerful, but have an algorithmic complexity which is at least quadratic in the amount of training data, meaning that they scale poorly to big data.

A popular way of circumventing this shortcoming of kernel methods is to approximate the kernel via random features. Such **random feature methods** include random Fourier features (Rahimi & Recht, 2008), random kitchen sinks (Rahimi & Recht, 2009), extreme learning machines (Huang et al., 2006), Fastfood (Le et al., 2013), orthogonal random features (Yu et al., 2016) and others. The algorithm proposed in Rahimi & Recht (2009), for instance, trades the quadratic dependency in the number of training examples for a quadratic dependency in the number of random features, a quantity that can be easily controlled.

A downside of this approach is the loss of the original exact kernel, and therefore of the intended predictor class, which consists of functions of the form (Bach, 2017):

$$f(x) = \mathop{\mathbb{E}}_{w \sim p} \left[ \alpha(w)\phi(w, x) \right]. \tag{1}$$

In this equation, $\phi$ is a **base predictor** or **feature function**, each $w$ is a **feature**, $p$ is a **feature distribution**, and $\alpha$ is a **weight function** over the feature space. In this paper, we investigate a variation of random feature methods which maintains an exact formulation for the model. The key is to consider weight functions which belong to a reproducing kernel Hilbert space (RKHS). Such spaces are well-behaved, and we show that the choices of RKHS, feature function $\phi$, and feature distribution $p$ sometimes allow Equation 1 to have an analytical form. The model can then be used directly, and exactly. The purpose of this paper is to determine whether this provides advantages, such as improved theoretical guarantees or improved empirical performance.

We provide three different algorithms for learning the model in this form. The most theoretically grounded is called stochastic functional gradient descent (SFGD). It is an iterative algorithm with convergence rate of $\mathcal{O}(1/\sqrt{m})$ with respect to the number $m$ of examples needed for learning (under mild assumptions). We also provide multiple bounds on the generalization gap based on the Rademacher complexity of the proposed class of predictors. These theoretical results are obtained using less stringent assumptions than Rahimi & Recht (2009).

Finally, we run experiments to compare this approach to other classic machine learning algorithms. Our approach compares favorably to random kitchen sinks (Rahimi & Recht, 2009) when the number of random features is small, often reaching higher prediction accuracies, but the performance evens out quickly as the number of random features increases. In our experiments, both RKHS weightings and random kitchen sinks also perform fairly similarly to AdaBoost and SVM. These results clearly show that the model and algorithm are viable machine learning tools, though further work is needed to extract the full potential of the model.

To summarize, our contributions in this paper are the following:

- We give multiple examples (calculus included) of random feature models which can be calculated exactly, provided the assumption that the weight function belongs to an RKHS.

- We provide one novel algorithm (Stochastic Functional Gradient Descent (SFGD)) for learning the weight function of an RKHS weighting, and two more algorithms inspired by the random kitchen sinks (Rahimi & Recht, 2009).

- We provide a workaround to the curse of dimensionality which can appear when using RKHS weightings.

- We provide a thorough theoretical analysis of RKHS weightings. This includes Rademacher complexity bounds, PAC bounds, a stability bound for SGFD, and approximation analysis of the empirical risk minimizer.

- We run experiments comparing RKHS weightings to random kitchen sinks (RKS) and other standard methods on several real-world classification and regression datasets.

The paper is structured as follows. We begin in Section 2 by recalling basic notions and notations of functional analysis and machine learning. This includes a short introduction to reproducing kernel Hilbert spaces. We summarize two meaningful prior works on this topic, by Rahimi & Recht (2009) and Bach (2017), in Section 3. We define the model and our assumptions explicitly in Section 4. We include several examples of model instantiations in Section 4.4, complete with the analytical form for the model calculation. We present three learning algorithms in Section 5. We prove several theoretical guarantees in Section 6: two bounds on the generalization gap in Section 6.1, based on the Rademacher complexity of the class of predictors; a bound on the rate of convergence of Algorithm 3 in Section 6.2, using the stability properties of this algorithm; and a bound on the true risk of the empirical risk minimizer in Section 6.3. We compare these results to those of Rahimi & Recht (2009) and Bach (2017) in Section 6.4. We present experimental results in Section 7, demonstrating the soundness of our approach. Finally, we discuss the limitations of our approach and future work in Section 8.

| Acronym | Meaning | Relevant link |
|---|---|---|
| erf | Error function, $\mathrm{erf}(x) := \frac{2}{\sqrt{\pi}} \int_0^x e^{-t^2} \mathrm{d}t$ | |
| MSE | Mean square error | |
| PAC | Probably approximately correct | |
| $R^2$ | Coefficient of determination | |
| ReLU | Rectified linear unit, $\mathrm{ReLU}(z) := \max(0, z)$ | |
| RKHS | Reproducing kernel Hilbert space | Berlinet & Thomas-Agnan (2011) |
| RKS | Random kitchen sinks | Rahimi & Recht (2009) |
| RWExpRelu | RKHS weighting using ReLU and exponential kernel | Table 2 |
| RWExpSign | RKHS weighting using sign function and exponential kernel | Table 2 |
| RWReLU | RKHS weighting using ReLU and gaussian kernel | Table 2 |
| RWSign | RKHS weighting using sign function and gaussian kernel | Table 2 |
| RWStumps | RKHS weighting using decisions stumps | Table 2 |
| SFGD | Stochastic functional gradient descent | Algorithm 3 |
| SVM | Support vector machine | Pedregosa et al. (2011) |
| SVR | Support vector regressor | Pedregosa et al. (2011) |
| SVC | Support vector classifier | Pedregosa et al. (2011) |

Table 1: Acronyms used in this paper.

## 2 Basic notions

We begin in Section 2.1 by recalling the main notions and definitions of supervised machine learning, and establishing the notation to be used in the remainder of the paper. Then, we recall core definitions of functional analysis in Section 2.2, and give a summary description of reproducing kernel Hilbert spaces in Section 2.3. In Section 2.5, we present the notion of the functional gradient, which is crucial to the working of one of the learning algorithms.

### 2.1 Machine learning

Consider an **instance space** $\mathcal{X}$, and a **label space** $\mathcal{Y} \subseteq \mathbb{R}$. A **predictor**, **hypothesis** or **model** is a function $h : \mathcal{X} \to \mathcal{Y}$ (we will equivalently write $h \in \mathcal{Y}^{\mathcal{X}}$, where $\mathcal{Y}^{\mathcal{X}}$ is the space of all functions from $\mathcal{X}$ to $\mathcal{Y}$). We also refer to a set of predictors $\mathcal{F}$ as a **class** of predictors. We are in the presence of a **classification** problem when $\mathcal{Y}$ is a finite set. We are, instead, in the **regression** setting when $\mathcal{Y} = \mathbb{R}$. The most basic and important type of classification problem is **binary classification**, which denotes the situation when $\mathcal{Y}$ contains only two labels. In that case, we can write $\mathcal{Y} = \{-1, 1\}$ without loss of generality.

We call a pair $(x, y) \in \mathcal{X} \times \mathcal{Y}$ an **example**. A **sample** or **training set** is a sequence or set of examples $\mathcal{S} = \{(x_i, y_i)\}_{i=1}^m \subset (\mathcal{X} \times \mathcal{Y})^m$. A learning algorithm $\mathcal{A}$ takes a sample $\mathcal{S}$ as input and outputs a predictor $\mathcal{A}(\mathcal{S}) \in \mathbb{R}^{\mathcal{X}}$. The goal is for this predictor to satisfy some notion of accuracy, and the algorithm will achieve this by optimizing a criterion defined with respect to the training set. To this end, we define a **loss** to be a function $\ell : \mathbb{R} \times \mathcal{Y} \to [0, \infty)$. Given a predictor $h$, we can calculate the loss's value on example $(x, y)$ as $\ell(h(x), y)$. This value can be seen as a penalty for predicting $h(x)$ when the correct value is $y$. Common examples are the **0-1 loss** $\ell(h(x), y) := \mathbb{1}[\mathrm{sign}(h(x)) \neq y]$ and the **square loss**, defined by $\ell(h(x), y) := (h(x) - y)^2$.

We define the **empirical risk** of the predictor $h$ on the sample $\mathcal{S}$ by

$$\mathcal{L}_{\mathcal{S}}(h) := \frac{1}{m} \sum_{i=1}^m \ell(h(x_i), y_i). \tag{2}$$

Furthermore, we suppose that all examples have been sampled i.i.d. from a data-generating probability distribution $\mathcal{D}$ over $\mathcal{X} \times \mathcal{Y}$. This allows us to define the **true risk** of the predictor $h$ by

$$\mathcal{L}_{\mathcal{D}}(h) := \mathop{\mathbb{E}}_{(x,y)\sim\mathcal{D}}[\ell(h(x), y)]. \tag{3}$$

In the binary classification setting, the **classification error** of $h$ is its expected 0-1 loss:

$$\mathcal{L}_{\mathcal{D}}^{01}(h) := \mathop{\mathbb{E}}_{(x,y)\sim\mathcal{D}} \mathbb{1}[\,\mathrm{sign}(h(x)) \neq y], \tag{4}$$

and its **empirical classification error** is the average 0-1 loss on the training set:

$$\mathcal{L}_{\mathcal{S}}^{01}(h) := \frac{1}{m} \sum_{i=1}^{m} \mathbb{1}[\,\mathrm{sign}(h(x_i)) \neq y_i]. \tag{5}$$

The **generalization gap** of a predictor $h$ is the difference $\mathcal{L}_{\mathcal{D}}(h) - \mathcal{L}_{\mathcal{S}}(h)$ between its true risk and its empirical risk. A **generalization gap bound** (or **guarantee)** for a class of predictors $\mathcal{F}$ ensures, with high probability over the sampling of the training set $\mathcal{S}$, that the generalization gap will be small. Typically, a generalization gap bound holds for all models in the class simultaneously, meaning that, with high probability over the choice of $\mathcal{S}$, $\mathcal{L}_{\mathcal{D}}(h) - \mathcal{L}_{\mathcal{S}}(h)$ is small for all $h \in \mathcal{F}$. Such guarantees improve as the training size $m$ increases.

## 2.2 Functional analysis

A **normed space** is a vector space $\mathcal{X}$ equipped with a norm $\|\cdot\|$, often written as a pair $(\mathcal{X}, \|\cdot\|)$. For example, $\mathbb{R}^n$ with the euclidean norm $\|\cdot\|_2$ is a normed space. An important category of normed spaces are the $L^p$ spaces.[1] For $p \in [1, \infty)$ and a measure $\mu$ over $\mathcal{X}$, we define

$$L^p(\mu) := \left\{ f : \mathcal{X} \to \mathbb{R} \,\middle|\, \|f\|_{L^p(\mu)} := \left( \int_{\mathcal{X}} |f(x)|^p \, \mathrm{d}\mu(x) \right)^{1/p} < \infty \right\}. \tag{6}$$

For the special case $p = \infty$, we define:

$$L^\infty(\mathcal{X}) := \left\{ f : \mathcal{X} \to \mathbb{R} \,\middle|\, \|f\|_\infty := \sup_{x\in\mathcal{X}} |f(x)| < \infty \right\}. \tag{7}$$

We will often call $\|\cdot\|_\infty$ the **supremum norm** or the **sup-norm**. We denote **scalar (or inner, or dot) products** on a vector space $\mathcal{X}$ by $\langle\cdot,\cdot\rangle : \mathcal{X} \times \mathcal{X} \to \mathbb{R}$. Given an operator $\Lambda : \mathcal{X} \to \mathcal{Y}$, we usually write $\Lambda x$ instead of $\Lambda(x)$. The **norm** of a linear operator is defined by

$$\|\Lambda\| = \|\Lambda\|_{\mathcal{X}\to\mathcal{Y}} := \sup_{x\in\mathcal{X}:\|x\|_{\mathcal{X}}=1} \|\Lambda x\|_{\mathcal{Y}}. \tag{8}$$

The operator is said to be **bounded** if its norm is finite. Note that a linear operator is bounded if and only if it is continuous. An operator $\Lambda : \mathcal{X} \to \mathbb{R}$ is called a **functional**. The following theorem applies to bounded (i.e. continuous) linear functionals.

**Theorem 2.1** (Riesz representation theorem). *Consider a Hilbert space $\mathcal{H}$, and a bounded linear functional $L : \mathcal{H} \to \mathbb{R}$. There exists a unique $h \in \mathcal{H}$ such that, for all $\alpha \in \mathcal{H}$, $L\alpha = \langle\alpha, h\rangle_{\mathcal{H}}$. This element $h$ is called the representative of the functional $L$.*

This theorem states that bounded linear functionals on a Hilbert space can be written as scalar products. It is a surprisingly powerful result. It forms the basis for understanding reproducing kernel Hilbert spaces, as we will see in the next section.

(See e.g. Atkinson & Han (2005) for more details on functional analysis.)

---

[1] The definition of $L^p$ spaces that we provide is an oversimplification. Technically, an element of an $L^p$ space is an equivalence class of almost-everywhere-equal functions, since the $p$-norm, being defined by an integral, does not distinguish functions which differ on a null set. We do not need this level of detail, and so leave it to this footnote.

### 2.3 Reproducing kernel Hilbert spaces

Formally, a **reproducing kernel Hilbert space** on a set $\mathcal{W}$ is a Hilbert space $\mathcal{H} \subset \mathbb{R}^{\mathcal{W}}$ which has the property that the evaluation functionals are continuous. This means that for any $w \in \mathcal{W}$, the evaluation functional $L_w : \mathcal{H} \to \mathbb{R}$ defined by $L_w(\alpha) := \alpha(w)$ is continuous. By the Riesz representation theorem, for each $w \in \mathcal{W}$, there exists a unique element $\mathcal{K}_w \in \mathcal{H}$ such that

$$\alpha(w) = L_w(\alpha) = \langle \alpha, \mathcal{K}_w \rangle_{\mathcal{H}} \quad \text{for all } \alpha \in \mathcal{H}. \tag{9}$$

Because $\mathcal{K}_w$ is itself a function with domain $\mathcal{W}$, the Riesz representation theorem gives us that

$$\mathcal{K}_w(u) = L_u(\mathcal{K}_w) = \langle \mathcal{K}_w, \mathcal{K}_u \rangle_{\mathcal{H}}. \tag{10}$$

The **reproducing kernel** of the RKHS $\mathcal{H}$ is the function defined by

$$\mathcal{K}(w, u) := \mathcal{K}_w(u) = \langle \mathcal{K}_w, \mathcal{K}_u \rangle_{\mathcal{H}}. \tag{11}$$

As we can see from the definition of the kernel, the element $\mathcal{K}_w$ corresponds exactly to the function $\mathcal{K}(w, \cdot) : \mathcal{W} \to \mathbb{R}$. For clarity, we use this new notation from now on.

The reproducing kernel calculates the scalar product between two elements $w, u \in \mathcal{W}$ following the embedding $w \mapsto \mathcal{K}(w, \cdot)$ into $\mathcal{H}$. Importantly, it is not necessary to calculate this embedding explicitly, which would be impossible in general if $\mathcal{H}$ was infinite-dimensional. In fact, the Moore–Aronszajn theorem (see e.g. Berlinet & Thomas-Agnan (2011)) states that any symmetric positive definite kernel $\mathcal{K} : \mathcal{W} \times \mathcal{W} \to \mathbb{R}$ is the reproducing kernel of an RKHS $\mathcal{H}$ of real-valued functions with domain $\mathcal{W}$, and every function $\alpha \in \mathcal{H}$ can be written as a sum or a convergent series of the form:

$$\alpha = \sum_{i=1}^{\infty} a_i \mathcal{K}(w_i, \cdot), \tag{12}$$

for some real coefficients $a_i$ and some $w_i \in \mathcal{W}$. Finally, given this representation for the elements in $\mathcal{H}$, the scalar product can also be written as a sum (or series). For $\alpha, \beta \in \mathcal{H}$ that can be written as $\alpha = \sum_{i=1}^{\infty} a_i \mathcal{K}(w_i, \cdot)$ and $\beta = \sum_{i=1}^{\infty} b_i \mathcal{K}(u_i, \cdot)$, the scalar product is given by:

$$\langle \alpha, \beta \rangle_{\mathcal{H}} = \sum_{i=1}^{\infty} \sum_{j=1}^{\infty} a_i b_j \mathcal{K}(w_i, u_j). \tag{13}$$

### 2.4 Kernel methods

Suppose that we have a positive definite kernel $\mathcal{K} : \mathcal{X} \times \mathcal{X} \to \mathbb{R}$ over the instance space. As we explained above, this kernel computes the dot product in its corresponding reproducing kernel Hilbert space $\mathcal{H}$,

$$\mathcal{K}(x_i, x_j) = \langle \mathcal{K}(x_i, \cdot), \mathcal{K}(x_j, \cdot) \rangle_{\mathcal{H}}, \tag{14}$$

after the mapping $x \mapsto \mathcal{K}(x, \cdot)$. This is highly useful in the context of machine learning. The training examples $\{(x_i, y_i)\}_{i=1}^{m}$ might not be linearly separable in their original space, but can well be after being embedded into the (usually) infinite-dimensional RKHS. And despite the infinite dimensionality of the RKHS, this dot product can easily be computed in finite time through the kernel. A **kernel method** is a machine learning algorithm that learns a predictor using this trick. The predictor will belong to the RKHS, and be a weighted sum of the mappings of the examples:

$$f(x) = \sum_{i=1}^{m} a_i \mathcal{K}(x_i, x). \tag{15}$$

More precisely, we have:

$$f(x) = \langle \alpha, \mathcal{K}(x, \cdot) \rangle_{\mathcal{H}}, \tag{16}$$

where $\alpha = \sum_{i=1}^{m} a_i \mathcal{K}(x_i, \cdot) \in \mathcal{H}$. This is a linear predictor (linear in the weight function $\alpha$), and benefits from all the advantages of linear models. The most well-known kernel methods are support vector machines (Steinwart & Christmann, 2008) and kernel ridge regression (Vovk, 2013).

## 2.5 Functional gradient

The usual notion of gradient (crucial for gradient descent algorithms) can be generalized to functionals. Consider normed spaces $\mathcal{X}$ and $\mathcal{Y}$. An operator $\Lambda : \mathcal{X} \to \mathcal{Y}$ is **Fréchet differentiable** at $x \in \mathcal{X}$ if and only if there exists a bounded linear operator $A : \mathcal{X} \to \mathcal{Y}$ such that:

$$\Lambda(x + z) = \Lambda x + Az + o(\|z\|), \qquad z \to 0. \tag{17}$$

See Definition 5.3.1 of Atkinson & Han (2005). We call $A$ the **Fréchet derivative** of $\Lambda$ at $x$. If $\Lambda$ is a bounded linear operator, then it is its own Fréchet derivative.

Notice that when $\Lambda$ is a functional, that is when $\mathcal{Y} = \mathbb{R}$, the Fréchet derivative $A$ is a bounded linear functional. If $\mathcal{X}$ is a Hilbert space, then the Riesz representation theorem is applicable, allowing us to write $A$ as a scalar product of the form:

$$Az = \langle z, \nabla_x \Lambda x \rangle_{\mathcal{X}}, \tag{18}$$

for a unique element $\nabla_x \Lambda x \in \mathcal{X}$. Equation (17) becomes:

$$\Lambda(x + z) = \Lambda x + \langle z, \nabla_x \Lambda x \rangle_X + o(\|z\|), \qquad z \to 0. \tag{19}$$

We call $\nabla_x \Lambda x$ the **functional gradient** of $\Lambda$ at $x$.

The functional gradient is a well-defined mathematical object, but it often cannot be calculated in practice. A notable exception is when the input space $\mathcal{X}$ is an RKHS, in which case the functional gradient sometimes admits an analytical form that can be calculated exactly in finite time. The following lemmas calculate the functional gradient of the evaluation functionals and the squared norm in an RKHS.

**Lemma 2.1.** *Consider an RKHS $\mathcal{H}$ of kernel $\mathcal{K}$ over a vector space $\mathcal{W}$. The functional gradient of the evaluation functional $L_w(\alpha) := \alpha(w)$ for some fixed point $w \in \mathcal{W}$ is:*

$$\boldsymbol{\nabla}_\alpha L_w(\alpha) = \boldsymbol{\nabla}_\alpha \alpha(w) = \mathcal{K}(w, \cdot).$$

*Proof.* Notice that $\alpha(w) = L_w(\alpha) = \langle \alpha, \mathcal{K}(w, \cdot) \rangle$ is a bounded linear functional. As mentioned above, a bounded linear functional is its own Fréchet derivative. By comparing to Equation (18), we can immediately conclude that the functional gradient of the evaluation functional must be its representer $\mathcal{K}(w, \cdot)$. $\square$

**Lemma 2.2.** *Consider an RKHS $\mathcal{H}$ and $\alpha \in \mathcal{H}$. Then $\boldsymbol{\nabla}_\alpha \|\alpha\|_{\mathcal{H}}^2 = 2\alpha$.*

*Proof.* Let $\varepsilon > 0$ and $\beta \in \mathcal{H}$, with $\|\beta\|_{\mathcal{H}} = 1$. We have:

$$\begin{aligned}
\|\alpha + \varepsilon\beta\|_{\mathcal{H}}^2 &= \langle \alpha + \varepsilon\beta, \alpha + \varepsilon\beta \rangle_{\mathcal{H}} \\
&= \langle \alpha, \alpha \rangle_{\mathcal{H}} + 2\langle \alpha, \varepsilon\beta \rangle_{\mathcal{H}} + \langle \varepsilon\beta, \varepsilon\beta \rangle_{\mathcal{H}} \\
&= \|\alpha\|_{\mathcal{H}}^2 + \langle 2\alpha, \varepsilon\beta \rangle_{\mathcal{H}} + \varepsilon^2 \|\beta\|_{\mathcal{H}}^2 \\
&= \|\alpha\|_{\mathcal{H}}^2 + \langle 2\alpha, \varepsilon\beta \rangle_{\mathcal{H}} + \varepsilon^2.
\end{aligned}$$

Comparing this to Equation (19), we can conclude that the functional gradient is $2\alpha$. $\square$

## 3 Related work

Suppose that we have a parameter space $\mathcal{W}$ and **feature function** (or **base predictor**) $\phi : \mathcal{W} \times \mathcal{X} \to \mathbb{R}$, as well as a probability distribution $p$ on $\mathcal{W}$. In this paper, we are interested in models of the form:

$$x \mapsto \mathbb{E}_{w \sim p} [\alpha(w)\phi(w, x)], \tag{20}$$

where $\alpha$ is a **weight function** over the parameter space. This is a well-known form, studied notably by Rahimi & Recht (2009) and Bach (2017). In this work, we are particularly concerned with the assumptions that can be made on the weight function, and the effect that these assumptions have on the approximation capability of the model and the ability to learn the model algorithmically with good generalization gap guarantees. Note that we will usually write these models as $\Lambda\alpha$, where $\Lambda : \mathbb{R}^{\mathcal{W}} \to \mathbb{R}^{\mathcal{X}}$ is the operator that transforms a weight function $\alpha : \mathcal{W} \to \mathbb{R}$ into a predictor with domain $\mathcal{X}$ according to Equation (20).

### 3.1 Random kitchen sinks

Rahimi & Recht (2009) define the following class of predictors (cf. Equation (6) of Rahimi & Recht (2009)):

$$\mathcal{F}_p := \left\{ x \mapsto \mathbb{E}_{w \sim p} \left[ \alpha(w)\phi(w, x) \right] \Big| \ \|\alpha\|_\infty \le B_\infty \right\}. \tag{21}$$

They assume that the weight function is bounded, so that $|\alpha(w)| \le B_\infty$ for all $w$. Since the functions in $\mathcal{F}_p$ cannot be evaluated exactly in general (because of the expectation), Rahimi & Recht (2009) also define the following approximation class:

$$\hat{\mathcal{F}}_w := \left\{ x \mapsto \frac{1}{T} \sum_{t=1}^{T} a_t \phi(w_t, x) \Big| \ \forall t, |a_t| \le B_\infty \right\}, \tag{22}$$

for $(w_1, \ldots, w_T)$ which have been randomly sampled according to the distribution $p$. Assuming that $\sup_{w,x} |\phi(w, x)| \le 1$, and that the loss is $\rho$-Lipschitz, the main result of Rahimi & Recht (2009) states that the output $\hat{f}$ of their Algorithm 1 is such that

$$\mathcal{L}_\mathcal{D}(\hat{f}) - \min_{f \in \mathcal{F}_p} \mathcal{L}_\mathcal{D}(f) \le \mathcal{O}\left( \left( \frac{1}{\sqrt{m}} + \frac{1}{\sqrt{T}} \right) \rho B_\infty \sqrt{\log \frac{2}{\delta}} \right), \tag{23}$$

with probability $1 - \delta$ on the sampling of the training set and the features $(w_1, \ldots, w_T)$. However, they use a different, more convenient algorithm in their experiments to learn the weights $\mathbf{a} := (a_1, \ldots, a_T)$. For any $f \in \hat{\mathcal{F}}_w$ with $f(x) = \frac{1}{T} \sum_{t=1}^{T} a_t \phi(w_t, x)$, consider the $\ell_2$-regularized empirical risk:[2]

$$\mathcal{L}_\mathcal{S}^{\ell_2}(f) := \mathcal{L}_\mathcal{S}(f) + \frac{\lambda_2}{T} \|\mathbf{a}\|_2^2. \tag{24}$$

If we also specifically use the squared loss, then the weights can be determined analytically. Defining the $m \times T$ matrix $\Phi$ with elements $\Phi_{it} := \phi(w_t, x_i)$, the vector of coefficients $\mathbf{a}$ which minimizes the above expression is the solution of the following linear problem:

$$\left( \Phi^\top \Phi + \frac{m\lambda_2}{T} I \right) \mathbf{a} = \Phi^\top \mathbf{y}, \tag{25}$$

where $\mathbf{y} := (y_1, \ldots, y_m)$. This yields the **random kitchen sinks algorithm**, summarized in Algorithm 1.

---

**Algorithm 1** Random kitchen sinks

---

**input** Feature function $\phi$, feature distribution $p$, $\lambda_2 > 0$, number of features $T$, sample $\mathcal{S}$ of size $m$
    Sample $(w_1, \ldots, w_T) \sim p^T$
    Calculate the matrix $\Phi \in \mathbb{R}^{m \times T}$, where $\Phi_{it} := \phi(w_t, x_i)$
    Get $(a_1, \ldots, a_T)$ by solving Equation (25)
**output** $x \mapsto \frac{1}{T} \sum_{t=1}^{T} a_t \phi(w_t, x)$

---

Note that whenever we refer to the random kitchen sinks algorithm (or **RKS algorithm**), we refer to Algorithm 1 rather than Algorithm 1 of Rahimi & Recht (2009). Finally, the $\ell_2$-regularization is also sometimes replaced by an $\ell_1$-regularization. In other words, the optimization objective becomes:

$$\mathcal{L}_\mathcal{S}^{\ell_1}(\Lambda\alpha) = \mathcal{L}_\mathcal{S}(\Lambda\alpha) + \frac{\lambda_1}{T} \|\mathbf{a}\|_1 \tag{26}$$

$$= \frac{1}{m} \|\Phi\mathbf{a} - \mathbf{y}\|_2^2 + \frac{\lambda_1}{T} \|\mathbf{a}\|_1. \tag{27}$$

Equation (27) can be solved using the Lasso (Hastie et al., 2015). Doing so yields a sparser predictor, where many of the coefficients $(a_1, \ldots, a_T)$ are zero. The pseudo-code for this version of the random kitchen sinks can be found in Algorithm 2.

---

[2]The atypical presence of $\frac{1}{T}$ in the regularization term is explained in Section 3.2.

---

**Algorithm 2** Random kitchen sinks (Lasso)

---

**input** Feature function $\phi$, feature distribution $p$, $\lambda_1 > 0$, number of features $T$, sample $\mathcal{S}$ of size $m$

    Sample $(w_1, \ldots, w_T) \sim p^T$

    Calculate the matrix $\Phi \in \mathbb{R}^{m \times T}$, where $\Phi_{it} := \phi(w_t, x_i)$

    Get $(a_1, \ldots, a_T)$ by solving Equation (27) using the Lasso.

**output** $x \mapsto \frac{1}{T} \sum_{t=1}^{T} a_t \phi(w_t, x)$

---

## 3.2 The link with kernel methods

It is not immediately evident from Equations (21) and (22) that the random kitchen sinks approximate a kernel method. However, it is in fact a direct consequence of (a variant of) the representer theorem for kernel methods (Schölkopf et al., 2001). First, notice that the model can be written as a scalar product in $L^2(p)$:

$$\Lambda \alpha(x) = \mathop{\mathbb{E}}_{w \sim p} [\alpha(w)\phi(w, x)] = \langle \alpha, \phi(\cdot, x) \rangle_{L^2(p)}. \tag{28}$$

We will see in Section 6, specifically in Theorem 6.5, that the generalization gap of this predictor is bounded by a function of the $L^2(p)$-norm of $\alpha$. In order to control this norm, we can minimize the $L^2(p)$-regularized empirical risk over the training dataset $\mathcal{S}$:

$$\mathcal{L}_{\mathcal{S}}^{L^2(p)}(\Lambda \alpha) := \mathcal{L}_{\mathcal{S}}(\Lambda \alpha) + \lambda_2 \|\alpha\|_{L^2(p)}^2$$

$$= \frac{1}{m} \sum_{i=1}^{m} \ell(\langle \alpha, \phi(\cdot, x_i) \rangle_{L^2(p)}, y_i) + \lambda_2 \|\alpha\|_{L^2(p)}^2. \tag{29}$$

The weight function $\alpha$ which minimizes this expression must be a weighted sum of the form:

$$\alpha = \sum_{i=1}^{m} b_i \phi(\cdot, x_i). \tag{30}$$

(Any component of $\alpha$ outside of $\{\sum_{i=1}^{m} c_i \phi(\cdot, x_i) \mid \forall i, c_i \in \mathbb{R}\}$ adds to $\|\alpha\|_{L^2(p)}^2$, but cannot affect the empirical risk. That component must therefore be 0 in the weight function that minimizes the regularized empirical risk.) Substituting this expression back into Equation (28), we get:

$$\Lambda \alpha(x) = \left\langle \sum_{i=1}^{m} b_i \phi(\cdot, x_i), \phi(\cdot, x) \right\rangle_{L^2(p)}$$

$$= \sum_{i=1}^{m} b_i \mathop{\mathbb{E}}_{w \sim p} [\phi(w, x_i)\phi(w, x)]. \tag{31}$$

We can define the following kernel:

$$k(x_i, x_j) = \mathop{\mathbb{E}}_{w \sim p} [\phi(w, x_i)\phi(w, x_j)]. \tag{32}$$

The optimal predictor is therefore an element of the RKHS of kernel $k$. Assuming that we cannot analytically calculate the above expectation, we can sample $(w_1, \ldots, w_T)$ from the distribution $p$ to approximate the kernel, the model, and the $L^2(p)$-norm of $\alpha$:

$$k(x_i, x_j) \approx \frac{1}{T} \sum_{t=1}^{T} \phi(w_t, x_i)\phi(w_t, x_j), \tag{33}$$

$$\Lambda \alpha(x) \approx \frac{1}{T} \sum_{t=1}^{T} \sum_{i=1}^{m} b_i \phi(w_t, x_i)\phi(w_t, x), \tag{34}$$

$$\|\alpha\|_{L^2(p)}^2 \approx \frac{1}{T} \sum_{t=1}^{T} \sum_{i=1}^{m} \sum_{j=1}^{m} b_i b_j \phi(w_t, x_i)\phi(w_t, x_j). \tag{35}$$

We can simplify the expressions by noticing that $a_t := \sum_{i=1}^m b_i \phi(w_t, x_i)$ is constant with regard to $x$. Writing $\mathbf{a} := (a_1, \ldots, a_T)$, we obtain:

$$\Lambda\alpha(x) \approx \frac{1}{T} \sum_{t=1}^T a_t \phi(w_t, x), \tag{36}$$

$$\|\alpha\|_{L^2(p)}^2 \approx \frac{1}{T} \sum_{t=1}^T a_t^2 = \frac{1}{T} \|\mathbf{a}\|_2^2. \tag{37}$$

Notice that the model in Equation (36) is precisely the form that functions in $\hat{\mathcal{F}}_w$, the approximation class of Rahimi & Recht (2009), take. Furthermore, using these approximations in the $L^2(p)$-regularized empirical risk, we get:

$$\mathcal{L}_{\mathcal{S}}^{L^2(p)}(\Lambda\alpha) = \mathcal{L}_{\mathcal{S}}(\Lambda\alpha) + \lambda_2 \|\alpha\|_{L^2(p)}^2$$
$$\approx \frac{1}{m} \sum_{i=1}^m \ell\left(\frac{1}{T} \sum_{t=1}^T a_t \phi(w_t, x), y_i\right) + \frac{\lambda_2}{T} \|\mathbf{a}\|_2^2. \tag{38}$$

The (squared) euclidean norm of the coefficients ($\|\mathbf{a}\|_2^2$) approximates the (squared) $L^2(p)$-norm of the weight function being learned. Note that this final expression differs slightly from the objective usually being minimized by the random kitchen sinks algorithm: a factor $\frac{1}{T}$ has appeared in the regularization term. The classical expressions are obtained by instead defining $a_t := \sum_{i=1}^m b_i \frac{\phi(w_t, x_i)}{\sqrt{T}}$, and $\Lambda\alpha(x) \approx \sum_{t=1}^T a_t \frac{\phi(w_t, x)}{\sqrt{T}}$. The regularization term would then be $\lambda_2 \|\mathbf{a}\|_2^2$. We opt for the formulation $\Lambda\alpha(x) \approx \frac{1}{T} \sum_{t=1}^T a_t \phi(w_t, x)$ for two main reasons. It directly highlights the Monte Carlo approximation being made, and also allows directly correlating the sup-norm of the weight function $\alpha$ in Equation (21) to the maximal coefficient in the approximation class (Equation (22)). Less importantly, it avoids the strange-looking and unintuitive $\sqrt{T}$ in the equations.

### 3.3 Square-integrable weight functions

Bach (2017) investigates the more general assumption that the weight function is simply square-integrable with respect to the probability measure $p$. He considers two main model classes:

$$\mathcal{F}_1 := \left\{ x \mapsto \mathop{\mathbb{E}}_{w \sim p}[\alpha(w)\phi(w, x)] \,\middle|\, \alpha \in L^2(p) \text{ and } \gamma_1(\Lambda\alpha) := \|\alpha\|_{L^1(p)} := \mathop{\mathbb{E}}_{w \sim p}\left[|\alpha(w)|\right] \le B_1 \right\}, \tag{39}$$

$$\mathcal{F}_2 := \left\{ x \mapsto \mathop{\mathbb{E}}_{w \sim p}[\alpha(w)\phi(w, x)] \,\middle|\, \alpha \in L^2(p) \text{ and } \gamma_2(\Lambda\alpha) := \|\alpha\|_{L^2(p)} := \sqrt{\mathop{\mathbb{E}}_{w \sim p}\left[\alpha(w)^2\right]} \le B_2 \right\}. \tag{40}$$

He shows that $\gamma_1$ and $\gamma_2$ are norms, and refers to them as **variational norms**.[3] The core result of Bach (2017) is Proposition 6, reproduced here for convenience:

**Proposition 6 of Bach (2017) (Approximation of Lipschitz-continuous functions)** *For $\delta$ larger than a constant that depends only on $d$ and $\alpha$, for any function $f : \mathbb{R}^d \to \mathbb{R}$ such that for all $x, y$ such that $\|x\|_q \le R$ and $\|y\|_q \le R$, $|f(x)| \le \eta$ and $|f(x) - f(y)| \le \eta R^{-1}\|x - y\|_q$, there exists $g \in \mathcal{F}_2$ such that $\gamma_2(g) \le \delta$ and*

$$\sup_{\|x\|_q \le R} |f(x) - g(x)| \le C(d, \alpha)\eta \left(\frac{\delta}{\eta}\right)^{-1/(\alpha+(d-1)/2)} \log\left(\frac{\delta}{\eta}\right). \tag{41}$$

Here, $\phi(w, x)$ is assumed to be the **rectified linear unit** (ReLU) raised to the power $\alpha$, that is, $\phi(w, x) = \text{ReLU}(w, x)^\alpha$ where $\text{ReLU}(w, x) := \max(0, \langle w, x \rangle)$. Moreover, $p$ is assumed to be the uniform distribution on

---

[3] We simplified the definition of the variational norms. Since multiple weight functions $\alpha$ can potentially yield the same predictor $f = \Lambda\alpha$, up to almost-everywhere equality, the variational norm is actually defined as the infimum of the $L^1(p)$ or $L^2(p)$-norm of these weight functions. This detail will not come into play in this paper, and is thus relegated to this footnote.

the unit sphere in $\mathbb{R}^n$. This Proposition 6 shows that the class $\mathcal{F}_2$ can successfully approximate Lipschitz-continuous functions. By making more specific assumptions on the target predictor $f$, for example that it is affine or a projection pursuit, he refines this result, yielding tighter approximation guarantees. He also proves corresponding generalization gap guarantees for the class $\mathcal{F}_2$. He combines both approximation and generalization bounds in his Table 2, showing the best tradeoff that can be achieved between the two (a larger class will have better approximation capabilities, but worse generalization gap guarantees). In the best cases, these guarantees are in $\mathcal{O}(1/\sqrt{m})$, as in the work of Rahimi & Recht (2009). However, if the target function $f$ is more complex than affine, the guarantees are no better than $\mathcal{O}(1/\sqrt[4]{m})$. Note that these are still meaningful results, as Theorem 1 of Rahimi & Recht (2009) does not include an approximation guarantee. On the other hand, Bach (2017) does not provide algorithms that can achieve these rates, and defers to Rahimi & Recht (2009) for practical applications.

## 4    RKHS weightings of functions

In this section, we describe the random feature method variation which we call **RKHS weightings**. It takes the form of an operator $\Lambda$ that takes as input an RKHS element $\alpha$, which is a function with domain $\mathcal{W}$, and outputs a predictor $\Lambda\alpha$ that has domain $\mathcal{X}$. The model and hypothesis classes are defined in Section 4.1. Because of the importance of the operator norm in the theoretical guarantees of Section 6, we examine the operator norm of $\Lambda$ in detail in Section 4.2. Then, we define all the assumptions that we make in Section 4.3. We give explicit examples of instantiations of the model, complete with formulas for calculating the model output, in Section 4.4. We explain the link between RKHS weightings and kernel methods in Section 4.5. Finally, Section 4.6 provides a method for selecting the model hyperparameters that takes into account the curse of dimensionality.

### 4.1    Model definition

As we alluded to in Section 3, in this paper we are interested in functions of the form:

$$\Lambda\alpha(x) := \mathop{\mathbb{E}}_{w\sim p} [\alpha(w)\phi(w,x)]. \tag{42}$$

Making a prediction for a given $x$ requires calculating the expectation in Equation (42), which can be difficult to do. In this paper, we explore the case where the weight function $\alpha$ is assumed to belong to an RKHS. We will see that this can lead to analytically solvable integrals, meaning that we can evaluate Equation (42) exactly. Let us therefore consider $\mathcal{K} : \mathcal{W} \times \mathcal{W} \to \mathbb{R}$, a positive definite kernel over $\mathcal{W}$. Denote the RKHS of kernel $\mathcal{K}$ by $\mathcal{H}$. We call the tuple $(\mathcal{W}, \phi, \mathcal{K}, p)$ an **instantiation** of the model, and define the following class of predictors:

$$\Lambda\mathcal{H} := \left\{ \Lambda\alpha \,\middle|\, \alpha \in \mathcal{H} \right\} = \left\{ x \mapsto \Lambda\alpha(x) := \mathop{\mathbb{E}}_{w\sim p} [\alpha(w)\phi(w,x)] \,\middle|\, \alpha \in \mathcal{H} \right\}. \tag{43}$$

We call functions in $\Lambda\mathcal{H}$ **RKHS weightings**. Similarly to Rahimi & Recht (2009), who bound the supremum norm of the weight function, and to Bach (2017), who limits the variational norm of the predictor, it will be necessary to limit the norm of the weight function $\alpha \in \mathcal{H}$. Therefore, for a constant $B_{\mathcal{H}} > 0$, we define the set

$$\mathcal{H}_B := \left\{ \alpha \in \mathcal{H} \,\middle|\, \|\alpha\|_{\mathcal{H}} \le B_{\mathcal{H}} \right\}, \tag{44}$$

and the following class of predictors:

$$\Lambda\mathcal{H}_B := \left\{ \Lambda\alpha \,\middle|\, \alpha \in \mathcal{H}, \ \|\alpha\|_{\mathcal{H}} \le B_{\mathcal{H}} \right\}. \tag{45}$$

In order to algorithmically control the norm $\|\alpha\|_{\mathcal{H}}$, we will typically add the regularization term $\lambda_{\mathcal{H}}\|\alpha\|_{\mathcal{H}}^2$ (with $\lambda_{\mathcal{H}} > 0$) to the empirical risk. Hence we define the **regularized empirical risk** by:

$$\mathcal{L}_{\mathcal{S}}^{\mathrm{reg}}(\Lambda\alpha) := \mathcal{L}_{\mathcal{S}}(\Lambda\alpha) + \lambda_{\mathcal{H}}\|\alpha\|_{\mathcal{H}}^2. \tag{46}$$

Furthermore, since weight functions $\alpha \in \mathcal{H}$ take the form $\alpha = \sum_{t=1}^{\infty} a_t \mathcal{K}(w_t, \cdot)$ for real coefficients $a_t$ and $w_t \in \mathcal{W}$, we can rewrite the model as

$$
\begin{aligned}
\Lambda\alpha(x) &:= \underset{w \sim p}{\mathbb{E}} \left[ \alpha(w)\phi(w, x) \right] \\
&= \underset{w \sim p}{\mathbb{E}} \left[ \sum_{t=1}^{\infty} a_t \mathcal{K}(w_t, w)\phi(w, x) \right] \\
&= \sum_{t=1}^{\infty} a_t \underset{w \sim p}{\mathbb{E}} \left[ \mathcal{K}(w_t, w)\phi(w, x) \right].
\end{aligned}
\tag{47}
$$

(Note that swapping the sum and integral in the expressions above is non-trivial. The proof is in Appendix A.3.) In practice, a finite number of features $\{w_t\}_{t=1}^{T}$ will be sampled from distribution $p$, and the coefficients $(a_t)_{t=1}^{T}$ will be learned. The model will then take the form

$$
\Lambda\alpha(x) = \sum_{t=1}^{T} a_t \underset{w \sim p}{\mathbb{E}} \left[ \mathcal{K}(w_t, w)\phi(w, x) \right].
\tag{48}
$$

To calculate this expression exactly, the expectation

$$
\underset{w \sim p}{\mathbb{E}} \left[ \mathcal{K}(u, w)\phi(w, x) \right]
\tag{49}
$$

must admit an analytical form for all $u \in \mathcal{W}$ and $x \in \mathcal{X}$. We give examples in Section 4.4.

So far, we have simply assumed that the model is well-defined (the expectation converges). In the next section, we explore under what conditions the operator $\Lambda$ is continuous. This will ensure that the model is well-defined, and give us useful results for deriving our theoretical guarantees in Section 6.

### 4.2 Norm of the operator

A crucial quantity, which appears in some way in all guarantees that we prove in Section 6, is the operator norm of $\Lambda$. The following theorem upper bounds that norm.

**Theorem 4.1.** *Consider an instance space $\mathcal{X}$, a parameter space $\mathcal{W}$, a function $\phi : \mathcal{W} \times \mathcal{X} \to \mathbb{R}$, an RKHS $\mathcal{H} \subset \mathbb{R}^{\mathcal{W}}$ of kernel $\mathcal{K}$, a distribution $p$ over $\mathcal{W}$ and the operator $\Lambda$ defined by*

$$
\Lambda\alpha := \underset{w \sim p}{\mathbb{E}} \left[ \alpha(w)\phi(w, \cdot) \right].
$$

*Consider the constant $\kappa$ defined by*

$$
\kappa := \sup_{x \in \mathcal{X}} \sqrt{\underset{w \sim p}{\mathbb{E}} \left[ \|\phi(w, x)\mathcal{K}(w, \cdot)\|_{\mathcal{H}}^2 \right]} = \sup_{x \in \mathcal{X}} \sqrt{\underset{w \sim p}{\mathbb{E}} \left[ \mathcal{K}(w, w)\phi(w, x)^2 \right]}.
\tag{50}
$$

*Suppose that $\kappa$ is finite. Then $\Lambda\alpha \in L^{\infty}(\mathcal{X})$ for all $\alpha \in \mathcal{H}$. Furthermore, the constant*

$$
\theta := \sup_{x \in \mathcal{X}} \left\| \underset{w \sim p}{\mathbb{E}} \left[ \phi(w, x)\mathcal{K}(w, \cdot) \right] \right\|_{\mathcal{H}} = \sup_{x \in \mathcal{X}} \sqrt{\underset{w \sim p}{\mathbb{E}} \underset{u \sim p}{\mathbb{E}} \left[ \mathcal{K}(u, w)\phi(u, x)\phi(w, x) \right]}
\tag{51}
$$

*is well-defined and we have:*

$$
\|\Lambda\| := \|\Lambda\|_{\mathcal{H} \to L^{\infty}(\mathcal{X})} \leq \theta \leq \kappa.
\tag{52}
$$

*Proof.* The ultimate goal is to prove that $\|\Lambda\| \leq \theta \leq \kappa$. This can be achieved by showing that $\|\Lambda\alpha\|_{\infty} \leq \theta\|\alpha\|_{\mathcal{H}} \leq \kappa\|\alpha\|_{\mathcal{H}}$ for any $\alpha \in \mathcal{H}$. To do this, we prove that $|\Lambda\alpha(x)| \leq \theta\|\alpha\|_{\mathcal{H}} \leq \kappa\|\alpha\|_{\mathcal{H}}$ for all $x$. We begin

by showing that $\|\Lambda\| \leq \kappa$. Consider any $\alpha \in \mathcal{H}$ and $x \in \mathcal{X}$. We have:

$$
\begin{aligned}
|\Lambda\alpha(x)| &= \left| \underset{w\sim p}{\mathbb{E}} [\alpha(w)\phi(w,x)] \right| \\
&\leq \underset{w\sim p}{\mathbb{E}} \left[ |\alpha(w)\phi(w,x)| \right] \\
&= \underset{w\sim p}{\mathbb{E}} \left[ |\langle \alpha, \mathcal{K}(w,\cdot)\rangle_{\mathcal{H}}\, \phi(w,x)| \right] && \text{(Reproducing property)} \\
&\leq \underset{w\sim p}{\mathbb{E}} \left[ \|\alpha\|_{\mathcal{H}} \|\mathcal{K}(w,\cdot)\|_{\mathcal{H}} |\phi(w,x)| \right] && \text{(Cauchy–Schwarz inequality)} \\
&= \underset{w\sim p}{\mathbb{E}} \left[ \|\mathcal{K}(w,\cdot)\|_{\mathcal{H}} |\phi(w,x)| \right] \|\alpha\|_{\mathcal{H}} \\
&\leq \sqrt{\underset{w\sim p}{\mathbb{E}} \left[ \|\phi(w,x)\mathcal{K}(w,\cdot)\|_{\mathcal{H}}^2 \right]} \|\alpha\|_{\mathcal{H}} && \text{(Jensen inequality)} \\
&\leq \kappa\|\alpha\|_{\mathcal{H}}.
\end{aligned}
$$

Therefore $\Lambda\alpha$ is a bounded function of $x$, and we have $\|\Lambda\| := \|\Lambda\|_{\mathcal{H}\to L^\infty(\mathcal{X})} \leq \kappa$. Additionally, we can observe that $\alpha \mapsto \Lambda\alpha(x)$, seen as an operator from $\mathcal{H}$ to $\mathbb{R}$, is a bounded linear functional of norm at most $\kappa$. The Riesz representation theorem tells us that we can write

$$\Lambda\alpha(x) = \langle \alpha, \psi(x)\rangle_{\mathcal{H}}, \tag{53}$$

for some $\psi(x) \in \mathcal{H}$. In fact, we have that

$$\psi(x) = \underset{w\sim p}{\mathbb{E}} [\phi(w,x)\mathcal{K}(w,\cdot)], \tag{54}$$

since, by the reproducing property of the RKHS, we can write, for all $u \in \mathcal{W}$,

$$
\begin{aligned}
\psi(x)(u) &= \langle \mathcal{K}(u,\cdot), \psi(x)\rangle_{\mathcal{H}} \\
&= \Lambda\mathcal{K}(u,\cdot)(x) && \text{(Equation (53))} \\
&= \underset{w\sim p}{\mathbb{E}} [\mathcal{K}(w,u)\phi(w,x)].
\end{aligned}
$$

Using the Cauchy–Schwarz inequality, we have that

$$|\Lambda\alpha(x)| \leq \|\alpha\|_{\mathcal{H}} \|\psi(x)\|_{\mathcal{H}} = \|\alpha\|_{\mathcal{H}} \left\| \underset{w\sim p}{\mathbb{E}} [\phi(w,x)\mathcal{K}(w,\cdot)] \right\|_{\mathcal{H}} \leq \theta\|\alpha\|_{\mathcal{H}}. \tag{55}$$

This shows that $\|\Lambda\| \leq \theta$. Finally, we need to show that $\theta \leq \kappa$. This is a direct consequence of Jensen's inequality:

$$
\begin{aligned}
\theta &= \sup_{x\in\mathcal{X}} \left\| \underset{w\sim p}{\mathbb{E}} [\phi(w,x)\mathcal{K}(w,\cdot)] \right\|_{\mathcal{H}} \\
&\leq \sup_{x\in\mathcal{X}} \underset{w\sim p}{\mathbb{E}} \left[ \|\phi(w,x)\mathcal{K}(w,\cdot)\|_{\mathcal{H}} \right] \\
&\leq \sup_{x\in\mathcal{X}} \sqrt{\underset{w\sim p}{\mathbb{E}} \left[ \|\phi(w,x)\mathcal{K}(w,\cdot)\|_{\mathcal{H}}^2 \right]} = \kappa. && \square
\end{aligned}
$$

We give examples of the values (or upper bounds) of $\theta$ and $\kappa$ for different instantiations of the model in Section 4.4. A recurring pattern is that $\theta$ is more difficult to calculate, but yields a much tighter bound on the operator norm of $\Lambda$.

Under the more general condition that $\alpha \in L^2(p)$, satisfied in the works of Rahimi & Recht (2009) and Bach (2017), we can derive a similar bound on the operator norm of $\Lambda$. We do so in the next lemma.

**Lemma 4.1.** *Consider an instance space $\mathcal{X}$, a parameter space $\mathcal{W}$, a function $\phi : \mathcal{W} \times \mathcal{X} \to \mathbb{R}$, a distribution $p$ over $\mathcal{W}$ and the operator $\Lambda : L^2(p) \to \mathbb{R}^{\mathcal{X}}$ defined by*

$$\Lambda\alpha(x) = \mathop{\mathbb{E}}_{w \sim p} [\alpha(w)\phi(w, x)].$$

*Define the constant*

$$\tau := \sup_{x \in \mathcal{X}} \|\phi(\cdot, x)\|_{L^2(p)} = \sup_{x \in \mathcal{X}} \sqrt{\mathop{\mathbb{E}}_{w \sim p} [\phi(w, x)^2]}. \tag{56}$$

*Suppose that $\tau < \infty$. Then $\Lambda\alpha \in L^{\infty}(\mathcal{X})$ for all $\alpha \in L^2(p)$, and $\|\Lambda\|_{L^2(p) \to L^{\infty}(\mathcal{X})} \leq \tau$.*

*Proof.* Consider any $\alpha \in L^2(p)$. We need to show that $|\Lambda\alpha(x)| \leq \tau \|\alpha\|_{L^2(p)}$ for all $x$. We have that

$$
\begin{aligned}
|\Lambda\alpha(x)| &= \left| \mathop{\mathbb{E}}_{w \sim p} [\alpha(w)\phi(w, x)] \right| \\
&= \left| \langle \alpha, \phi(\cdot, x) \rangle_{L^2(p)} \right| \\
&\leq \|\alpha\|_{L^2(p)} \|\phi(\cdot, x)\|_{L^2(p)} \qquad\qquad \text{(Cauchy–Schwarz)} \\
&\leq \tau \|\alpha\|_{L^2(p)}.
\end{aligned}
$$

$\square$

Lemma4.1 also applies to RKHS weightings if the weight function is square-integrable. Because we always have the inequality

$$\|\alpha\|_{L^2(p)}^2 = \mathop{\mathbb{E}}_{w \sim p} [\alpha(w)^2] \leq \|\alpha\|_{\mathcal{H}}^2 \mathop{\mathbb{E}}_{w \sim p} [\mathcal{K}(w, w)], \tag{57}$$

we can ensure that $\|\alpha\|_{L^2(p)} \leq \infty$ by adding the assumption that $\mathbb{E}_{w \sim p} [\mathcal{K}(w, w)] < \infty$.

### 4.3 Assumptions

The theoretical results in this paper use some or all of the following assumptions.

**Assumption 1 (A1).** Consider an instance space $\mathcal{X}$, an instantiation $(\mathcal{W}, \phi, \mathcal{K}, p)$ of the model such that the constants $\kappa$ defined in Equation (50) and $\tau$ defined in Equation (56) are finite, $\mathcal{H}$ the RKHS of kernel $\mathcal{K}$, the operator $\Lambda$ defined by Equation (42) and the predictor class $\Lambda\mathcal{H}_B$ defined by Equation (45). Additionally suppose that $\mathbb{E}_{w \sim p} [\mathcal{K}(w, w)] < \infty$.

**Assumption 2 (A2).** The loss $\ell : \mathbb{R} \times \mathbb{R} \to [0, \infty)$ is $\rho$-Lipschitz in the first argument (the output of the model).

**Assumption 3 (A3).** The loss $\ell : \mathbb{R} \times \mathbb{R} \to [0, \infty)$ is convex and differentiable in its first argument. We write $\ell'(z, y) := \frac{\partial \ell}{\partial z}(z, y)$ for the partial derivative with regard to the first argument.

### 4.4 Examples of instantiations of the model

An instantiation of the model defined in Equation (42) is a tuple $(\mathcal{W}, \phi, \mathcal{K}, p)$. This is a highly flexible model family. Table 2 contains several examples of instantiations of the model, along with their constants $\theta$ and $\kappa$. Additionally, Table 3 provides the analytical form for the expectation

$$\mathop{\mathbb{E}}_{w \sim p} [\mathcal{K}(u, w)\phi(w, x)], \tag{58}$$

which is required to evaluate the model (see Equation (48)). All of the calculations can be found in the appendix, except for the instantiation RWRelu, which was introduced by Dubé & Marchand (2025).

| | RWSign | RWExpSign | RWRelu | RWExpRelu | RWStumps |
|---|---|---|---|---|---|
| $\mathcal{W}$ | $\mathbb{R}^n$ | $\mathbb{R}^n$ | $\mathbb{R}^n$ | $\mathbb{R}^n$ | $\{1,\dots,n\} \times \mathbb{R}$ |
| $p$ | $\mathcal{N}(0,\sigma^2 I)$ | $\mathcal{N}(0,\sigma^2 I)$ | $\mathcal{N}(0,\sigma^2 I)$ | $\mathcal{N}(0,\sigma^2 I)$ | $\mathcal{U}(\{1,\dots,n\}) \times \mathcal{N}(0,\sigma^2)$ |
| $\phi(w,x)$ | $\mathrm{sign}(\langle w,x\rangle)$ | $\mathrm{sign}(\langle w,x\rangle)$ | $\mathrm{ReLU}(w,x)$ | $\mathrm{ReLU}(w,x)$ | $\mathrm{sign}(x_{w_1} - w_2)$ |
| $\mathcal{K}(u,w)$ | $\exp\left(-\frac{\|u-w\|_2^2}{2\gamma^2}\right)$ | $\exp\left(\frac{\langle u,w\rangle}{2\gamma^2}\right)$ | $\exp\left(-\frac{\|u-w\|_2^2}{2\gamma^2}\right)$ | $\exp\left(\frac{\langle u,w\rangle}{2\gamma^2}\right)$ | $\mathbb{1}[u_1 = w_1]\exp\left(-\frac{(u_2-w_2)^2}{2\gamma^2}\right)$ |
| $\theta \leq$ | $\left(1+\frac{2\sigma^2}{\gamma^2}\right)^{-\frac{n}{4}}$ | $\left(1-\frac{\sigma^2}{2\gamma^2}\right)^{-\frac{n}{2}}$ | $\frac{\sigma}{\sqrt{2\pi}}\sup_x\|x\|\left(1+\frac{2\sigma^2}{\gamma^2}\right)^{\frac{1-n}{4}}$ | $\frac{\sigma}{\sqrt{2\pi}}\sup_x\|x\|\left(1-\frac{\sigma^2}{2\gamma^2}\right)^{\frac{1-n}{2}}$ | $\frac{1}{\sqrt{n}}\left(1+\frac{2\sigma^2}{\gamma^2}\right)^{-\frac{1}{4}}$ |
| $\kappa =$ | $1$ | $\left(1-\frac{\sigma^2}{\gamma^2}\right)^{-\frac{n}{4}}$ | $\frac{\sigma}{\sqrt{2}}\sup_x\|x\|$ | $\frac{\sigma}{\sqrt{2}}\sup_x\|x\|\left(1-\frac{\sigma^2}{\gamma^2}\right)^{-\frac{n+2}{4}}$ | $1$ |

Table 2: Five RKHS weighting instantiations. For RWStumps, each $w$ is a tuple $(w_1, w_2)$, where $w_1$ is an integer, indicating which variable the decision stump considers, and $w_2$ is the threshold, a real number.

| | $\mathbb{E}_{w\sim p}\left[\mathcal{K}(u,w)\phi(w,x)\right]$ | Proof |
|---|---|---|
| RWSign | $\left(1+\frac{\sigma^2}{\gamma^2}\right)^{-n/2} e^{\frac{-\|u\|_2^2}{2\sigma^2+2\gamma^2}}\,\mathrm{erf}\left(\frac{\langle u',x\rangle}{\sqrt{2}\zeta\|x\|_2}\right)$ | Appendix B.1 |
| RWExpSign | $e^{\frac{\sigma^2}{8\gamma^4}\|u\|^2}\,\mathrm{erf}\left(\frac{\sigma}{\sqrt{8\pi\gamma^2}}\frac{\langle u,x\rangle}{\|x\|}\right)$ | Appendix B.2 |
| RWRelu | $\left(1+\frac{\sigma^2}{\gamma^2}\right)^{-n/2} e^{\frac{-\|u\|_2^2}{2\sigma^2+2\gamma^2}}\frac{\|x\|}{2\sqrt{\pi}}\left(\sqrt{2}\zeta e^{-\frac{\langle u',x\rangle^2}{2\zeta^2\|x\|^2}} + \sqrt{\pi}\frac{\langle u',x\rangle}{\|x\|}\left[1+\mathrm{erf}\left(\frac{\langle u',x\rangle}{\sqrt{2}\zeta\|x\|}\right)\right]\right)$ | Dubé & Marchand (2025) |
| RWExpRelu | $\left(\frac{1}{\sqrt{2\pi\sigma^2}}\right)e^{\frac{\sigma^2}{8\gamma^4}\|u\|_2^2}\left[\frac{\sigma\|x\|}{\sqrt{2}}\left(\sqrt{2}\sigma e^{-\frac{\langle u,x\rangle^2}{2\sigma^2\|x\|^2}} + \sqrt{\pi}\frac{\langle u,x\rangle}{\|x\|}\left[1+\mathrm{erf}\left(\frac{\langle u,x\rangle}{\sqrt{2}\sigma\|x\|}\right)\right]\right)\right]$ | Appendix B.3 |
| RWStumps | $\frac{\zeta}{\sigma n}e^{\frac{-u_2^2}{2\sigma^2+2\gamma^2}}\,\mathrm{erf}\left(\frac{x_{u_1}-u_2'}{\sqrt{2}\zeta}\right)$ | Appendix B.4 |

Table 3: Analytical form for the expectation. The constant $\zeta$ is defined by the relationship $\frac{1}{2\zeta^2} = \frac{1}{2\gamma^2} + \frac{1}{2\sigma^2}$, and $u' := \left(1+\frac{\gamma^2}{\sigma^2}\right)^{-1} u$.

## 4.5 RKHS weightings are implicitly a kernel method

Using the representer theorem for kernel methods, we showed in Section 3 that random feature methods approximate kernel methods when minimizing the $L^2(p)$-norm-regularized empirical risk (see Equation (29)). A similar reasoning applies to RKHS weightings paired with an RKHS-norm regularization. We show below that choosing an instantiation $(\mathcal{W}, \phi, \mathcal{K}, p)$ implicitly defines an RKHS over the instance space $\mathcal{X}$, and that the optimal predictor belongs to that RKHS. Indeed, by selecting a valid instantiation $(\mathcal{W}, \phi, \mathcal{K}, p)$, we get Equation (53):

$$\Lambda\alpha(x) = \langle\alpha, \psi(x)\rangle_{\mathcal{H}},$$

where $\psi(x) := \mathbb{E}_{w\sim p}\left[\phi(w,x)\mathcal{K}(w,\cdot)\right] \in \mathcal{H}$. Using this form for $\Lambda\alpha(x)$ in the expression of the regularized empirical risk, we get:

$$\begin{aligned}
\mathcal{L}_{\mathcal{S}}^{\mathrm{reg}}(\Lambda\alpha) &= \mathcal{L}_{\mathcal{S}}(\Lambda\alpha) + \lambda_{\mathcal{H}}\|\alpha\|_{\mathcal{H}}^2 \\
&= \frac{1}{m}\sum_{i=1}^m \ell(\Lambda\alpha(x_i), y_i) + \lambda_{\mathcal{H}}\|\alpha\|_{\mathcal{H}}^2 \\
&= \frac{1}{m}\sum_{i=1}^m \ell(\langle\alpha, \psi(x_i)\rangle_{\mathcal{H}}, y_i) + \lambda_{\mathcal{H}}\|\alpha\|_{\mathcal{H}}^2.
\end{aligned} \tag{59}$$

The representer theorem for kernel methods (Schölkopf et al., 2001) tells us that the optimal weight function $\alpha_{\mathcal{S}} \in \mathcal{H}$ which minimizes the previous expression must be a linear combination of the $\{\psi(x_i)\}_{i=1}^m$. In other words, we must have $\alpha_{\mathcal{S}} \in \mathcal{H}_{\mathcal{S}}$, with

$$\mathcal{H}_{\mathcal{S}} := \left\{\sum_{i=1}^m a_i\psi(x_i)\right\}. \tag{60}$$

Inserting this form for the optimal predictor back into Equation (53), we obtain:

$$\Lambda\alpha_{\mathcal{S}}(x) = \langle \alpha_{\mathcal{S}}, \psi(x) \rangle_{\mathcal{H}}$$

$$= \left\langle \sum_{i=1}^{m} a_i \psi(x_i), \psi(x) \right\rangle_{\mathcal{H}}$$

$$= \sum_{i=1}^{m} a_i \langle \psi(x_i), \psi(x) \rangle_{\mathcal{H}}. \tag{61}$$

If we define the kernel $\mathcal{K}_{\mathcal{X}} : \mathcal{X} \times \mathcal{X} \to \mathbb{R}$ as:

$$\mathcal{K}_{\mathcal{X}}(x_i, x_j) := \langle \psi(x_i), \psi(x_j) \rangle_{\mathcal{H}} \tag{62}$$

$$= \left\langle \mathop{\mathbb{E}}_{w \sim p} [\phi(w, x_i)\mathcal{K}(w, \cdot)], \mathop{\mathbb{E}}_{w \sim p} [\phi(w, x_j)\mathcal{K}(w, \cdot)] \right\rangle_{\mathcal{H}}$$

$$= \mathop{\mathbb{E}}_{u \sim p} \mathop{\mathbb{E}}_{w \sim p} [\phi(w, x_i)\phi(u, x_j) \langle \mathcal{K}(u, \cdot), \mathcal{K}(w, \cdot) \rangle_{\mathcal{H}}]$$

$$= \mathop{\mathbb{E}}_{u \sim p} \mathop{\mathbb{E}}_{w \sim p} [\mathcal{K}(u, w)\phi(u, x_i)\phi(w, x_j)], \tag{63}$$

then Equation (53) becomes:

$$\Lambda\alpha_{\mathcal{S}}(x) = \sum_{i=1}^{m} a_i \mathcal{K}_{\mathcal{X}}(x_i, x), \tag{64}$$

or, equivalently:

$$\Lambda\alpha_{\mathcal{S}} = \sum_{i=1}^{m} a_i \mathcal{K}_{\mathcal{X}}(x_i, \cdot). \tag{65}$$

We call $\mathcal{K}_{\mathcal{X}}$ the **implicit kernel** of the instantiation $(\mathcal{W}, \phi, \mathcal{K}, p)$. We can also show that $\mathcal{K}_{\mathcal{X}}$ is a positive definite kernel. Given any set of instances $\{x_i\}_{i=1}^{m}$, we need to prove that the following holds for all $(c_1, \ldots, c_m) \in \mathbb{R}^m$:

$$\sum_{i=1}^{m} \sum_{j=1}^{m} c_i c_j \mathcal{K}_{\mathcal{X}}(x_i, x_j) \geq 0. \tag{66}$$

This immediately follows from the definition of $\mathcal{K}_{\mathcal{X}}$, as we can write:

$$\sum_{i=1}^{m} \sum_{j=1}^{m} c_i c_j \mathcal{K}_{\mathcal{X}}(x_i, x_j) = \sum_{i=1}^{m} \sum_{j=1}^{m} c_i c_j \langle \psi(x_i), \psi(x_j) \rangle_{\mathcal{H}}$$

$$= \left\langle \sum_{i=1}^{m} c_i \psi(x_i), \sum_{j=1}^{m} c_j \psi(x_j) \right\rangle_{\mathcal{H}}$$

$$= \left\| \sum_{i=1}^{m} c_i \psi(x_i) \right\|_{\mathcal{H}}^{2} \geq 0.$$

In fact, this shows that the norm $\|\Lambda\alpha\|_{\mathcal{H}_{\mathcal{X}}}$ of the predictor $\Lambda\alpha$, with $\alpha = \sum_{i=1}^{m} c_i \psi(x_i)$ is equal to the RKHS norm $\|\alpha\|_{\mathcal{H}} = \|\sum_{i=1}^{m} c_i \psi(x_i)\|_{\mathcal{H}}$ of the weight function. We encapsulate this fact in the following lemma.

**Lemma 4.2.** *Assume A1. Suppose we have a dataset $\mathcal{S} \subset \mathcal{X} \times \mathcal{Y}$. If $\alpha \in \mathcal{H}_{\mathcal{S}}$, then $\Lambda\alpha \in \mathcal{H}_{\mathcal{X}}$, and $\|\Lambda\alpha\|_{\mathcal{H}_{\mathcal{X}}} = \|\alpha\|_{\mathcal{H}}$.*

By the Moore–Aronsajn theorem, $\mathcal{K}_{\mathcal{X}}$ is the reproducing kernel of an RKHS $\mathcal{H}_{\mathcal{X}}$, which contains functions precisely of the form of Equation (65). This means that $\Lambda\alpha_{\mathcal{S}} \in \mathcal{H}_{\mathcal{X}}$.

To conclude this section, we address the obvious idea of pairing the implicit kernel $\mathcal{K}_{\mathcal{X}}$ with a standard kernel method in order to learn the model. This can only be done if there is an analytical form for Equation (63). It requires solving a highly difficult integral even for simple instantiations. We do not pursue that avenue in this paper. Alternatively, Equation (63) can be approximated via Monte Carlo sampling. We will cover that possibility in a subsequent paper.

### 4.6 Curse of dimensionality and choice of parameters

RKHS Weightings suffer from the curse of dimensionality. This can be seen explicitly in the value of the constants $\theta$ and $\kappa$ in Table 2, and in the analytical expectations in Table 3. For example, we have the upper bound:

$$\theta \leq \left(1 + \frac{2\sigma^2}{\gamma^2}\right)^{-n/4}$$

for RWSign, where $\sigma^2$ and $\gamma^2$ are the variances of the Gaussian distribution and kernel, respectively. Because the output of the model is upper bounded as follows:

$$|\Lambda\alpha(x)| \leq \theta\|\alpha\|_{\mathcal{H}}, \tag{Theorem 4.1}$$

the predictions $\Lambda\alpha(x)$ will quickly vanish to zero as the dimensionality $n$ grows and $\theta$ goes to 0 exponentially fast. To compensate, the RKHS norm of the weight function $\alpha$ would need to be exponentially large. However, as we will see in the theoretical guarantees in Section 6, for example Theorem 6.7, this would require an exponential number of examples to ensure generalization.

We can work around the curse of dimensionality by choosing the model hyperparameters adequately. To see this, let us continue with our example of RWSign, which uses the Gaussian kernel (see Table 2). If $\gamma$ is too large, then the kernel value $\mathcal{K}(w, u)$ will be close to 1 for most pairs $(w, u)$. The weight functions, of the form $\alpha = \sum_{t=1} a_t \mathcal{K}(w_t, \cdot)$, will be roughly constant. This yields a non-expressive predictor class $\Lambda\mathcal{H}_B$. If $\gamma$ is too small, then the model vanishes to a constant value of 0, as we explained above.

What we propose is a way to find the small range of values for the hyperparameters where the constants $\theta$ and $\kappa$ take on reasonable values. For RWSign and RWRelu, where

$$\theta \propto \left(1 + \frac{2\sigma^2}{\gamma^2}\right)^{-n/4},$$

we can choose $\theta$ and $\sigma$, and then find $\gamma$ using the following relationship:

$$\gamma^2 = \frac{2\sigma^2}{\theta^{-4/n} - 1}. \tag{67}$$

For RWExpSign and RWExpRelu, the problem is the opposite; the model explodes to infinity with the dimensionality. Instead of controlling the smallest constant $\theta$, we need to limit the largest constant $\kappa$, which also comes into play in Section 6 guarantees. In both cases, we have that

$$\kappa \propto \left(\frac{1}{1 - \frac{\sigma^2}{\gamma^2}}\right)^{n/4}.$$

We can therefore take

$$\gamma^2 = \frac{\sigma^2}{1 - \kappa^{-4/n}}. \tag{68}$$

Finally, RWStumps is an exception. It has no exponential dependency on dimensionality, since the distribution is only over a 2-dimensional space. The hyperparameters can be chosen freely. Table 4 summarizes how to set the hyperparameters for all the instantiations of Table 2.

It is clear that the curse of dimensionality must be taken into account when choosing an instantiation to model a given problem. One way to avoid the problem entirely is to use a smaller parameter space. RWStumps is such an example. The parameter space $\mathcal{W}$ is not $\mathbb{R}^n$; it is instead a two-dimensional space, regardless of the dimensionality of the instance space $\mathcal{X}$. In return, it does not exhibit an exponential behavior in $n$. Indeed, we have the simple upper bound $\theta \leq \frac{1}{\sqrt{n}}\left(1 + \frac{2\sigma^2}{\gamma^2}\right)^{-1/4}$.

RWStumps represents a form of prior knowledge applied to the model. By using RWStumps, we assume that decision stumps on individual variables are sufficient for good prediction. For problems on which that assumption is true, RWStumps will provide better learning guarantees, and avoid the curse of dimensionality.

|  | Formula for $\gamma$ | Allowed values |
|---|---|---|
| RWSign | $\gamma^2 = \frac{2\sigma^2}{\theta^{-4/n}-1}$ | $\theta \in (0,1)$ |
| RWRelu | $\gamma^2 = \frac{2\sigma^2}{\theta^{-4/n}-1}$ | $\theta \in (0,1)$ |
| RWExpSign | $\gamma^2 = \frac{\sigma^2}{1-\kappa^{-4/n}}$ | $\kappa > 1$ |
| RWExpRelu | $\gamma^2 = \frac{\sigma^2}{1-\kappa^{-4/n}}$ | $\kappa > 1$ |
| RWStumps | N/A | N/A |

Table 4: Formulas for finding reasonable kernel parameter values from the distribution parameter.

## 5 Learning the model

Learning the model of Equation (42) consists in finding the weight function $\alpha$ which minimizes an optimization objective, such as the regularized empirical risk. We present three such algorithms in this paper.

### 5.1 Stochastic functional gradient descent

The first algorithm is a stochastic gradient descent in RKHS space. This requires calculating the functional gradient of the empirical risk functional (Equation (2)) with regard to weight function. We start with the following lemma, which gives the functional gradient of the evaluation functionals.

**Lemma 5.1.** *Assume A1. Then* $\boldsymbol{\nabla}_\alpha \Lambda\alpha(x) = \psi(x) := \mathbb{E}_{w\sim p}\left[\phi(w,x)\mathcal{K}(w,\cdot)\right]$.

*Proof.* See Equation (53). $\qquad\square$

We can use Lemma 5.1, the linearity of the gradient, as well as the chain rule, to calculate the gradient of the empirical risk functional. We do this in the next theorem.

**Theorem 5.1.** *Assume A1, A2 and A3. Suppose we have a sample* $\mathcal{S} \subseteq (\mathcal{X} \times \mathcal{Y})^m$. *Then:*

$$\boldsymbol{\nabla}_\alpha(\mathcal{L}_\mathcal{S}(\Lambda\alpha)) = \mathop{\mathbb{E}}_{w\sim p}\left[\frac{1}{m}\sum_{i=1}^{m}\ell'(\Lambda\alpha(x_i),y_i)\phi(w,x_i)\mathcal{K}(w,\cdot)\right]. \tag{69}$$

*Proof.* We have:

$$\boldsymbol{\nabla}_\alpha(\mathcal{L}_\mathcal{S}(\Lambda\alpha)) = \frac{1}{m}\sum_{i=1}^{m}\boldsymbol{\nabla}_\alpha\ell(\Lambda\alpha(x_i),y_i) \qquad\qquad \text{(Linearity of the gradient)}$$

$$= \frac{1}{m}\sum_{i=1}^{m}\ell'(\Lambda\alpha(x_i),y_i)\boldsymbol{\nabla}_\alpha\Lambda\alpha(x_i) \qquad\qquad \text{(Chain rule)}$$

$$= \frac{1}{m}\sum_{i=1}^{m}\ell'(\Lambda\alpha(x_i),y_i)\mathop{\mathbb{E}}_{w\sim p}\left[\phi(w,x_i)\mathcal{K}(w,\cdot)\right] \qquad\qquad \text{(Lemma 5.1)}$$

$$= \mathop{\mathbb{E}}_{w\sim p}\left[\frac{1}{m}\sum_{i=1}^{m}\ell'(\Lambda\alpha(x_i),y_i)\phi(w,x_i)\mathcal{K}(w,\cdot)\right]. \qquad \text{(Linearity of the expectation)}$$

$$\square$$

By the linearity of the gradient and Lemma 2.2, we have that:

$$\boldsymbol{\nabla}_\alpha\mathcal{L}_\mathcal{S}^{\text{reg}}(\Lambda\alpha) = \boldsymbol{\nabla}_\alpha\mathcal{L}_\mathcal{S}(\Lambda\alpha) + \boldsymbol{\nabla}_\alpha\lambda_\mathcal{H}\|\alpha\|_\mathcal{H}^2 = \boldsymbol{\nabla}_\alpha\mathcal{L}_\mathcal{S}(\Lambda\alpha) + 2\lambda_\mathcal{H}\alpha. \tag{70}$$

Because the functional gradients of Equations (69) and (70) are expectations over the choice of $w$, they are difficult objects to work with in practice. However, we can easily extract unbiased approximations of the

gradients through simple random sampling. By sampling a feature $u$ according to the distribution $p$, and a data batch $\mathcal{B}$ of size $b$ uniformly (with replacement) from $\mathcal{S}$ (we denote the uniform distribution over $\mathcal{S}$ by $U(\mathcal{S})$), we can define the following unbiased approximation for the functional gradient $\boldsymbol{\nabla}_\alpha \mathcal{L}_\mathcal{S}^{\mathrm{reg}}(\Lambda\alpha)$:

$$v(\alpha, u, \mathcal{B}) := \left( \frac{1}{b} \sum_{(x,y)\in\mathcal{B}} \ell'(\Lambda\alpha(x), y)\phi(u,x) \right) \mathcal{K}(u,\cdot) + 2\lambda_\mathcal{H}\alpha. \tag{71}$$

This approximation is indeed unbiased since, by Equations (69) and (70), we get that:

$$\mathop{\mathbb{E}}_{u\sim p} \mathop{\mathbb{E}}_{\mathcal{B}\sim U(\mathcal{S})^b} [v(\alpha, u, \mathcal{B})] = \boldsymbol{\nabla}_\alpha \mathcal{L}_\mathcal{S}^{\mathrm{reg}}(\Lambda\alpha). \tag{72}$$

Assuming that the loss $\ell$ is convex, we can apply a stochastic gradient descent algorithm using this approximation, with an update at iteration $t$ of the form

$$\alpha \leftarrow \alpha - \eta_t v(\alpha, w_t, \mathcal{B}_t), \tag{73}$$

for a given stepsize $\eta_t$, and be guaranteed to converge to the optimal solution. In fact, $\mathcal{L}_\mathcal{S}^{\mathrm{reg}}(\Lambda\alpha)$ is $\lambda_\mathcal{H}$-strongly convex in $\alpha$, by convexity of $\ell$ and linearity of $\Lambda\alpha$. We therefore propose using a stochastic functional gradient descent algorithm for $\lambda_\mathcal{H}$-strongly convex functions, similar to the one found in Section 14.4.4 of Shalev-Shwartz & Ben-David (2014). See Algorithm 3 for the pseudocode and Theorem 6.7 for a convergence guarantee.

---

**Algorithm 3** Stochastic functional gradient descent for learning the weight function

---

**input** Instantiation $(\mathcal{W}, \phi, \mathcal{K}, p)$, $\lambda_\mathcal{H} > 0$, $B_\mathcal{H} > 0$, number of iterations $T$, sample $\mathcal{S}$, batch size $b$

$\quad \alpha^{(0)} \leftarrow 0 \in \mathcal{H}$

$\quad$ **for** $t = 1, \ldots, T$ **do**

$\quad\quad$ Sample $\mathcal{B}_t \sim U(\mathcal{S})^b$

$\quad\quad$ Sample $w_t \sim p$

$\quad\quad \eta_t \leftarrow \frac{1}{\lambda_\mathcal{H} t}$

$\quad\quad v_t \leftarrow v\big(\alpha^{(t-1)}, w_t, \mathcal{B}_t\big)$ (see Equation (71))

$\quad\quad \alpha^{(t-\frac{1}{2})} \leftarrow \alpha^{(t-1)} - \eta_t v_t$

$\quad\quad \alpha^{(t)} \leftarrow \min\left(1, \dfrac{B_\mathcal{H}}{\left\|\alpha^{(t-\frac{1}{2})}\right\|_\mathcal{H}}\right)\alpha^{(t-\frac{1}{2})}$ (projection step)

$\quad$ **end for**

**output** $\bar{\alpha} := \frac{1}{T+1}\sum_{t=0}^T \alpha^{(t)}$

---

To analyse the algorithmic complexity of Algorithm 3, we can notice that the bottleneck is the projection step. This requires calculating the norm $\left\|\alpha^{(t-\frac{1}{2})}\right\|_\mathcal{H}$, which can be done in $\mathcal{O}(t^2)$, as it involves doing a matrix multiplication using the $t \times t$ Gram matrix of the parameters at iteration $t$. Over $T$ iterations, the projection step therefore costs $\mathcal{O}(T^3)$ in computation time. However, a simple optimization can reduce this cost by a factor of $T$ to $\mathcal{O}(T^2)$, in line with the rest of the algorithm. Indeed, notice that the update formula is:

$$\alpha^{(t-\frac{1}{2})} \leftarrow \alpha^{(t-1)} - \eta_t v_t. \tag{74}$$

Referring back to Equation (71) and using the shorthand $c_t := \left( \frac{1}{b} \sum_{(x,y)\in\mathcal{B}_t} \ell'(\Lambda\alpha(x), y)\phi(w_t, x) \right)$, this can be rewritten as:

$$\alpha^{(t-\frac{1}{2})} \leftarrow \alpha^{(t-1)} - \eta_t\big(c_t\mathcal{K}(w_t,\cdot) + 2\lambda_\mathcal{H}\alpha^{(t-1)}\big) = (1 - 2\eta_t\lambda_\mathcal{H})\alpha^{(t-1)} - \eta_t c_t\mathcal{K}(w_t,\cdot). \tag{75}$$

Table 5: Detailed breakdown of the algorithmic complexity of Algorithm 3.

| Operation | Complexity |
|---|---|
| Calculating the model output $\Lambda\alpha(x)$ for one instance at iteration $t$. | $\mathcal{O}(nt)$ |
| Calculating the model output $\Lambda\alpha(x)$ for all instances in the batch $\mathcal{B}_t$ at iteration $t$. | $\mathcal{O}(bnt)$ |
| Calculating the functional gradient approximation (Equation (71)) at iteration $t$. | $\mathcal{O}(bnt)$ |
| Calculating the functional gradient approximation (Equation (71)) for $t$ iterations. | $\mathcal{O}(bnT^2)$ |
| Updating the weight function at iteration $t$. | $\mathcal{O}(1)$ |
| Updating the weight function for $T$ iterations. | $\mathcal{O}(T)$ |
| Projection step at iteration $t$. | $\mathcal{O}(t)$ |
| Projection step for $T$ iterations. | $\mathcal{O}(T^2)$ |
| Total cost of Algorithm 3. | $\mathcal{O}(bnT^2 + T + T^2)$ |

Considering the squared norm, we have:

$$
\begin{aligned}
\left\|\alpha^{(t-\frac{1}{2})}\right\|_{\mathcal{H}}^2 &= \left\|(1 - 2\eta_t\lambda_{\mathcal{H}})\alpha^{(t-1)} - \eta_t c_t \mathcal{K}(w_t, \cdot)\right\|_{\mathcal{H}}^2 \\
&= (1 - 2\eta_t\lambda_{\mathcal{H}})^2\left\|\alpha^{(t-1)}\right\|_{\mathcal{H}}^2 - 2(1 - 2\eta_t\lambda_{\mathcal{H}})\eta_t c_t \left\langle\alpha^{(t-1)}, \mathcal{K}(w_t, \cdot)\right\rangle_{\mathcal{H}} + \eta_t^2 c_t^2 \|\mathcal{K}(w_t, \cdot)\|_{\mathcal{H}}^2 \\
&= (1 - 2\eta_t\lambda_{\mathcal{H}})^2\left\|\alpha^{(t-1)}\right\|_{\mathcal{H}}^2 - 2(1 - 2\eta_t\lambda_{\mathcal{H}})\eta_t c_t \alpha^{(t-1)}(w_t) + \eta_t^2 c_t^2 \mathcal{K}(w_t, w_t).
\end{aligned}
\tag{76}
$$

Calculating $\alpha^{(t-1)}(w_t)$ is in $\mathcal{O}(t)$, and $\mathcal{K}(w_t, w_t)$ takes constant time to compute (with regard to $t$). We can therefore reduce the learning time of the algorithm from $\mathcal{O}(T^3)$ to $\mathcal{O}(T^2)$ by simply keeping in memory the norm of each iterate, and using Equation (76) to calculate the norm of the next one. Table 5 contains a precise breakdown of the algorithmic complexity of Algorithm 3.

### 5.1.1 Sampling multiple random features at each iteration

Algorithm 3 samples multiple examples at each iteration in order to obtain a better functional gradient approximation and speed up convergence. Why not also sample multiple random features? The core issue is that the model size increases with each sampled random feature. If $c$ random features are sampled at each iteration, then the model contains $tc$ terms after iteration $t$. For a total number of iterations $T$ and $c$ features sampled at each iteration, the training cost of SFGD is in $\mathcal{O}(nbT^2c)$ (averaging Equation (71) at iteration $t$ requires $b$ model evaluations, each costing $\mathcal{O}(ntc)$ operations; this is done for each value of $t$ from 1 to $T$). This is the same cost as training with a single feature per iteration for $T\sqrt{c}$ iterations. This is therefore only advantageous, time-wise, if using a batch size of $c$ reduces the required number of iterations by a factor at least $\sqrt{c}$, from $T\sqrt{c}$ iterations with a batch size of 1, to $T$ iterations with a batch size of $c$. However, this still leads to a larger model (with $Tc$ terms rather than $T\sqrt{c}$), which means slower inference.[4]

### 5.2 Least squares fit of the random features

Many random feature methods first generate a large number of random parameters according to the sampling distribution, then learn the weight coefficients by analytically solving a convex optimization problem (Rahimi & Recht, 2009; Huang et al., 2006). This is possible when using the squared loss $\ell(h(x), y) := (h(x) - y)^2$. This idea works for RKHS Weightings as well. We can define the operator $\varphi$ which, given $(w_1, \ldots, w_T)$ sampled from the distribution $p$, embeds an instance $x$ into a higher dimensional space:

$$
\varphi(x) := \left(\mathop{\mathbb{E}}_{w\sim p}[\mathcal{K}(w_1, w)\phi(w, x)], \ldots, \mathop{\mathbb{E}}_{w\sim p}[\mathcal{K}(w_T, w)\phi(w, x)]\right)^{\top}.
$$

---

[4]The interested reader can look at Figures 7 and 8 in the appendix for an empirical look at the effect of increasing the random feature batch size.

The output of the model is then simply a linear function in the embedding space:

$$\Lambda\alpha(x) := \sum_{t=1}^{T} a_t \mathop{\mathbb{E}}_{w\sim p}\left[\mathcal{K}(w_t, w)\phi(w, x)\right] = \langle a, \varphi(x)\rangle, \tag{77}$$

For some sample $\mathcal{S} = \{(x_i, y_i)\}_{i=1}^{m}$, the regularized empirical squared loss is:

$$\mathcal{L}_{\mathcal{S}}^{\text{reg}}(\Lambda\alpha) := \frac{1}{m}\sum_{i=1}^{m}(\langle a, \varphi(x_i)\rangle - y_i)^2 + \lambda_{\mathcal{H}}\|\alpha\|_{\mathcal{H}}^2. \tag{78}$$

Denoting $\Phi := (\varphi(x_1), \ldots, \varphi(x_m))^{\top} \in \mathbb{R}^{m\times T}$, so that $\Phi_{i,t} = \mathbb{E}_{w\sim p}\left[\mathcal{K}(w_t, w)\phi(w, x_i)\right]$, also denoting $\mathbf{y} := (y_1, \ldots, y_m)^{\top}$ the vector of labels and $\mathbf{a} := (a_1, \ldots, a_T)$ the vector of coefficients, and finally denoting $G \in \mathbb{R}^{T\times T}$ the matrix defined by $G_{i,j} := \mathcal{K}(w_i, w_j)$, we can simplify:

$$\mathcal{L}_{\mathcal{S}}^{\text{reg}}(\Lambda\alpha) = \frac{1}{m}\|\Phi\mathbf{a} - \mathbf{y}\|_2^2 + \lambda_{\mathcal{H}}\mathbf{a}^{\top}G\mathbf{a}. \tag{79}$$

The minimizer $\mathbf{a} \in \mathbb{R}^T$ of this expression is the solution to the linear problem:

$$\left(\Phi^{\top}\Phi + m\lambda_{\mathcal{H}}G\right)\mathbf{a} = \Phi^{\top}\mathbf{y}. \tag{80}$$

Solving the linear system of Equation 80 requires $\mathcal{O}(T^3)$ operations, which is slower than the $\mathcal{O}(T^2)$ required by Algorithm 3. On the other hand, solving the linear system yields the optimal weights, requiring fewer sampled parameters to get the same accuracy. See Figure 1 in Section 7. Algorithm 4 summarizes this section,[5] and Table 6 contains a breakdown of the algorithmic complexity of Algorithm 4.

---

**Algorithm 4** Least squares fit of the weight function coefficients

**input** Instantiation $(\mathcal{W}, \phi, \mathcal{K}, p)$, $\lambda_{\mathcal{H}} > 0$, number of features $T$, sample $\mathcal{S}$ of size $m$
    Sample $(w_1, \ldots, w_T) \sim p^T$
    Calculate the matrix $\Phi \in \mathbb{R}^{m\times T}$, where $\Phi_{it} := \mathbb{E}_{w\sim p}\left[\mathcal{K}(w_t, w)\phi(w, x_i)\right]$
    Calculate the matrix $G \in \mathbb{R}^{T\times T}$, where $G_{ij} := \mathcal{K}(w_i, w_j)$
    Get $(a_1, \ldots, a_T)$ by solving Equation (80)
**output** $\alpha := \sum_{t=1}^{T} a_t\mathcal{K}(w_t, \cdot)$

---

### 5.3 Lasso fit of the random features

We can replace the Tikhonov regularizer $\lambda_{\mathcal{H}}\|\alpha\|_{\mathcal{H}}^2$ in Equation (79) by the $\ell_1$-regularizer on the norm of the coefficients, $\lambda_1\|\mathbf{a}\|_1$, giving us the new minimization objective:

$$\mathcal{L}_{\mathcal{S}}^{\ell_1}(\Lambda\alpha) = \frac{1}{m}\|\Phi\mathbf{a} - \mathbf{y}\|_2^2 + \lambda_1\|\mathbf{a}\|_1. \tag{83}$$

The solution $\mathbf{a}$ of this problem can be obtained by applying the Lasso algorithm (Hastie et al., 2015). By the nature of minimizing with an $\ell_1$-regularizer, the vector of coefficients $\mathbf{a}$ obtained this way will be sparse, which is an interesting advantage. Smaller models have a faster inference time and lower memory requirements, and are more readily interpretable. (Note that the algorithmic complexity of Algorithm 5 is dominated by the Lasso algorithm, in $\mathcal{O}(T^3)$.)

---

[5]In practice, rare numerical instability issues can arise when both the regularization parameter $\lambda_{\mathcal{H}}$ and the smallest eigenvalue of the matrix $G$ are too small. Our solution is to add a very small $\ell_2$-regularizer to the regularized empirical risk:

$$\mathcal{L}_{\mathcal{S}}^{\text{reg}}(\Lambda\alpha) = \frac{1}{m}\|\Phi\mathbf{a} - \mathbf{y}\|_2^2 + \lambda_{\mathcal{H}}\mathbf{a}^{\top}G\mathbf{a} + \epsilon\mathbf{a}^{\top}I\mathbf{a}. \tag{81}$$

In our experiments, we used $\epsilon = 10^{-10}$. Equation (80) becomes:

$$\left(\Phi^{\top}\Phi + m\lambda_{\mathcal{H}}G + m\epsilon I\right)\mathbf{a} = \Phi^{\top}\mathbf{y}. \tag{82}$$

Table 6: Detailed breakdown of the algorithmic complexity of Algorithm 4.

| Operation | Complexity |
|---|---|
| Calculating each $\Phi_{it} := \mathbb{E}_{w \sim p}\left[\mathcal{K}(w_t, w)\phi(w, x_i)\right]$. | $\mathcal{O}(n)$ |
| Calculating the matrix $\Phi$. | $\mathcal{O}(mnT)$ |
| Calculating the matrix $G$. | $\mathcal{O}(nT^2)$ (if $\mathcal{W} = \mathbb{R}^n$) |
| Calculating the matrix $\Phi^\top \Phi$. | $\mathcal{O}(mT^2)$ |
| Calculating the matrix $\Phi^\top \mathbf{y}$. | $\mathcal{O}(mT)$ |
| Solving Equation (80). | $\mathcal{O}(T^3)$ |
| Total cost of Algorithm 4. | $\mathcal{O}(mnT + nT^2 + mT^2 + T^3)$ |

---

**Algorithm 5** Lasso fit of the weight function coefficients

---

**input** Instantiation $(\mathcal{W}, \phi, \mathcal{K}, p)$, $\lambda_1 > 0$, number of features $T$, sample $\mathcal{S}$ of size $m$
    Sample $(w_1, \ldots, w_T) \sim p^T$
    Calculate the matrix $\Phi \in \mathbb{R}^{m \times T}$, where $\Phi_{it} := \mathbb{E}_{w \sim p}\left[\mathcal{K}(w_t, w)\phi(w, x_i)\right]$
    Get $(a_1, \ldots, a_T)$ by minimizing Equation (83) using the Lasso
**output** $\alpha := \sum_{t=1}^{T} a_t \mathcal{K}(w_t, \cdot)$

---

### 5.4 Comparison to existing algorithms

The three algorithms for learning RKHS weightings that we presented above are similar to or inspired by existing algorithms.

The Stochastic Functional Gradient Descent (SFGD), Algorithm 3, relies on the observation that an unbiased approximation of the functional gradient of the loss can be obtained when the function space being searched is an RKHS. This is somewhat similar to the work of Dai et al. (2014). First, Dai et al. (2014) assume that the model (rather than the weight function in our case) belongs to the RKHS $\mathcal{H}_k$ of kernel $k(x_1, x_2) = \mathbb{E}_{w \sim p}\left[\phi(w, x_1)\phi(w, x_2)\right]$. They then use a combination of RKHS properties and Bochner's Theorem to show that an unbiased approximation of the functional gradient at $f \in \mathcal{H}_k$ of the true risk $\mathcal{L}_{\mathcal{D}}(f) := \mathbb{E}_{(x,y) \sim \mathcal{D}}\left[\ell(f(x), y)\right]$ is

$$\nabla_f \mathcal{L}_{\mathcal{D}}(f) \approx \ell'(f(x), y)\phi(w, x)\phi(w, \cdot), \tag{84}$$

with $\ell'$ being the derivative of the loss with regard to its first argument. Finally, an SGD can be implemented using this functional gradient approximation. In contrast, Algorithm 3 searches for the weight function of the random feature model. This yields a few meaningful differences. For instance, Algorithm 3 always searches within the RKHS, while the functional gradient (and therefore the outputted model) in Dai et al. (2014) leaves the RKHS. This makes our theoretical analysis much simpler. Also, Algorithm 3 works with any well-defined kernel (as long as the condition that $\kappa < \infty$ is met), not just those of the form $k(x_1, x_2) = \mathbb{E}_{w \sim p}\left[\phi(w, x_1)\phi(w, x_2)\right]$. This makes Algorithm 3 applicable with a wider selection of kernels.

Our second algorithm, the least squares fit (Algorithm 4), can be seen as a variation of the random kitchen sinks (RKS) algorithm. Where the RKS would use the base predictions $\Phi_{it} := \phi(w_t, x_i)$ directly, we use the expectations $\Phi_{it} := \mathbb{E}_{w \sim p}\left[\mathcal{K}(w_t, w)\phi(w, x_i)\right]$. And where the RKS regularizes the euclidean norm of the vector of coefficients $(a_1, \ldots, a_T)$ being learned, we instead regularize the RKHS norm of the weight function that these coefficients imply. Similarly, the Lasso fit, Algorithm 5, simply replaces the regularization by an $\ell_1$-regularizer, which can also be done for the random kitchen sinks.

## 6 Theoretical guarantees

In this section, we demonstrate various theoretical guarantees. Section 6.1 contains two bounds on the generalization gap based on the Rademacher complexity of the class of predictors: one bound, valid only for RKHS weightings, depends on the RKHS norm of the weight function; the other, valid in general, depends on the $L^2(p)$-norm of the weight function. Section 6.2 contains a single result, a guarantee on the convergence of

Algorithm 3. Section 6.3 provides a more elaborate analysis of the generalization gap of RKHS weightings, relying on the implicit kernel formulation of the model (see Section 4.5). Afterward, we offer in Section 6.4 an in-depth comparison of these guarantees to Rahimi & Recht (2009) and Bach (2017).

## 6.1 Bounding the generalization gap using Rademacher complexity

A large generalization gap is the result of overfitting, which is usually a sign that the complexity of the class of predictors is inappropriately high in relation to the amount of available training data. Knowing this complexity allows us to choose model parameters adequately to limit overfitting. Our first theorem upper bounds the **empirical Rademacher complexity** of class $\Lambda \mathcal{H}_B$:

$$\widehat{\mathcal{R}}_S(\Lambda \mathcal{H}_B) := \frac{1}{m} \underset{\sigma \sim \{\pm 1\}^m}{\mathbb{E}} \left[ \sup_{\Lambda \alpha \in \Lambda \mathcal{H}_B} \sum_{i=1}^m \sigma_i \Lambda \alpha(x_i) \right], \tag{85}$$

and its **expected Rademacher complexity**:

$$\mathcal{R}_m(\Lambda \mathcal{H}_B) := \underset{S \sim \mathcal{D}^m}{\mathbb{E}} \left[ \widehat{\mathcal{R}}_S(\Lambda \mathcal{H}_B) \right]. \tag{86}$$

(See e.g. Shalev-Shwartz & Ben-David (2014) or Mohri et al. (2012) for a rigorous exposition of the Rademacher theory for bounding the generalization gap.)

**Theorem 6.1.** *Given assumptions A1 and A2, we have for any sample $S := \{(x_i, y_i)\}_{i=1}^m \subset (\mathcal{X} \times \mathcal{Y})^m$ that:*

$$\widehat{\mathcal{R}}_S(\Lambda \mathcal{H}_B) \leq \frac{\theta B_\mathcal{H}}{\sqrt{m}}, \tag{87}$$

*and:*

$$\mathcal{R}_m(\Lambda \mathcal{H}_B) \leq \frac{\theta B_\mathcal{H}}{\sqrt{m}}. \tag{88}$$

The proof is in Appendix A.4.

We see that the Rademacher complexity is characterized by two model-dependent constants. The first is $\theta$, which is defined by the instantiation of the model (Equation (51)). The second is $B_\mathcal{H}$, the maximal RKHS norm for the weight function, which acts as a hyperparameter of the algorithm, and can be used to control overfitting. We can convert the Rademacher complexity of $\Lambda \mathcal{H}_B$ into the following uniform bound on the generalization gap.

**Theorem 6.2.** *Given assumptions A1, A2, and A3, we have with probability at least $1 - \delta$ over the choice of $S \sim \mathcal{D}^m$ that the following holds for all $\Lambda \alpha \in \Lambda \mathcal{H}_B$:*

$$\mathcal{L}_\mathcal{D}(\Lambda \alpha) \leq \mathcal{L}_S(\Lambda \alpha) + \frac{2\rho \theta B_\mathcal{H}}{\sqrt{m}} \left( 1 + \sqrt{\frac{\log \frac{1}{\delta}}{2}} \right). \tag{89}$$

The proof is in Appendix A.5.

It will be useful in the proof of the next theorem to have a bound on $\mathcal{L}_S(\Lambda \alpha) - \mathcal{L}_\mathcal{D}(\Lambda \alpha)$ rather than $\mathcal{L}_\mathcal{D}(\Lambda \alpha) - \mathcal{L}_S(\Lambda \alpha)$. This is the following corollary.

**Corollary 6.2.1.** *Given assumptions A1, A2, and A3, we have with probability at least $1 - \delta$ over the choice of $S \sim \mathcal{D}^m$ that the following holds for all $\Lambda \alpha \in \Lambda \mathcal{H}_B$:*

$$\mathcal{L}_S(\Lambda \alpha) \leq \mathcal{L}_\mathcal{D}(\Lambda \alpha) + \frac{2\rho \theta B_\mathcal{H}}{\sqrt{m}} \left( 1 + \sqrt{\frac{\log \frac{1}{\delta}}{2}} \right). \tag{90}$$

The proof is in Appendix A.6.

Using this corollary and Theorem 6.2, we can derive a Probably Approximately Correct bound which justifies empirical risk minimization.

**Theorem 6.3.** *Assume A1, A2, and A3. For any $\mathcal{S} \in (\mathcal{X} \times \mathcal{Y})^m$, define:*

$$\alpha_{\mathcal{S}} := \underset{\alpha \in \mathcal{H}_B}{\operatorname{argmin}} \mathcal{L}_{\mathcal{S}}(\Lambda \alpha),$$

$$\alpha^* := \underset{\alpha \in \mathcal{H}_B}{\operatorname{argmin}} \mathcal{L}_{\mathcal{D}}(\Lambda \alpha).$$

*Then with probability at least $1 - \delta$ over the choice $\mathcal{S} \sim \mathcal{D}^m$, we have:*

$$\mathcal{L}_{\mathcal{D}}(\Lambda \alpha_{\mathcal{S}}) \leq \mathcal{L}_{\mathcal{D}}(\Lambda \alpha^\star) + \frac{4\rho\theta B_{\mathcal{H}}}{\sqrt{m}}\left(1 + \sqrt{\frac{\log \frac{2}{\delta}}{2}}\right). \tag{91}$$

*Proof.* We can write:

$$\begin{aligned}
\mathcal{L}_{\mathcal{D}}(\Lambda \alpha_{\mathcal{S}}) - \mathcal{L}_{\mathcal{D}}(\Lambda \alpha^\star) = &\mathcal{L}_{\mathcal{D}}(\Lambda \alpha_{\mathcal{S}}) - \mathcal{L}_{\mathcal{S}}(\Lambda \alpha_{\mathcal{S}}) \\
&+ \mathcal{L}_{\mathcal{S}}(\Lambda \alpha_{\mathcal{S}}) - \mathcal{L}_{\mathcal{S}}(\Lambda \alpha^\star) \\
&+ \mathcal{L}_{\mathcal{S}}(\Lambda \alpha^\star) - \mathcal{L}_{\mathcal{D}}(\Lambda \alpha^\star).
\end{aligned} \tag{92}$$

From Theorem 6.2, the first term is bounded by $\frac{2\rho\theta B_{\mathcal{H}}}{\sqrt{m}}\left(1 + \sqrt{\frac{\log \frac{1}{\delta}}{2}}\right)$ with probability at least $1 - \delta$. From Corollary 6.2.1, the third term is also bounded by $\frac{2\rho\theta B_{\mathcal{H}}}{\sqrt{m}}\left(1 + \sqrt{\frac{\log \frac{1}{\delta}}{2}}\right)$ with probability at least $1 - \delta$. Both inequalities hold together with probability at least $1 - 2\delta$. The second term is smaller than 0, since $\Lambda \alpha_{\mathcal{S}}$ is the empirical risk minimizer. We obtain the result by replacing $\delta$ by $\frac{\delta}{2}$. $\square$

The previous three theorems can be replicated with only minor differences using the condition that the weight function $\alpha$ be square-integrable with respect to the distribution $p$. In other words, the condition is now that $\|\alpha\|_{L^2(p)} \leq B_2$ rather than that $\alpha$ belongs to an RKHS. We can therefore derive the Rademacher complexity of class $\mathcal{F}_2$ of Bach (2017). This will apply by extension to the class $\mathcal{F}_p$ of Rahimi & Recht (2009), since $\|\alpha\|_\infty \leq B_\infty$ implies that $\|\alpha\|_{L^2(p)} \leq B_\infty$.

**Theorem 6.4.** *Given assumptions A1 and A2, we have for any sample $\mathcal{S} := \{(x_i, y_i)\}_{i=1}^m \subset (\mathcal{X} \times \mathcal{Y})^m$ that:*

$$\widehat{\mathcal{R}}_{\mathcal{S}}(\mathcal{F}_2) \leq \frac{\tau B_2}{\sqrt{m}}, \tag{93}$$

*and:*

$$\mathcal{R}_m(\mathcal{F}_2) \leq \frac{\tau B_2}{\sqrt{m}}. \tag{94}$$

(The proof is in Appendix A.7.) This implies a very similar generalization bound as Theorem 6.2.

**Theorem 6.5.** *Given assumptions A1, A2, and A3, we have with probability at least $1 - \delta$ over the choice of $\mathcal{S} \sim \mathcal{D}^m$ that the following holds for all $\alpha \in L^2(p)$ with $\|\alpha\|_{L^2(p)} \leq B_2$:*

$$\mathcal{L}_{\mathcal{D}}(\Lambda \alpha) \leq \mathcal{L}_{\mathcal{S}}(\Lambda \alpha) + \frac{2\rho\tau B_2}{\sqrt{m}}\left(1 + \sqrt{\frac{\log \frac{1}{\delta}}{2}}\right). \tag{95}$$

Note also that since $\mathcal{H} \subset L^2(p)$, this previous theorem also applies for weight functions taken from $\mathcal{H}$. We can take the minimum of $\{\theta\|\alpha\|_{\mathcal{H}}, \tau\|\alpha\|_{L^2(p)}\}$ in the Rademacher bound for the tightest possible guarantee. We then have the following corollary:

**Corollary 6.5.1.** *Given assumptions A1, A2, and A3, we have with probability at least $1 - \delta$ over the choice of $\mathcal{S} \sim \mathcal{D}^m$ that the following holds for all $\alpha \in L^2(p)$ with $\|\alpha\|_{L^2(p)} \leq B$:*

$$\mathcal{L}_{\mathcal{S}}(\Lambda \alpha) \leq \mathcal{L}_{\mathcal{D}}(\Lambda \alpha) + \frac{2\rho\tau B_2}{\sqrt{m}}\left(1 + \sqrt{\frac{\log \frac{1}{\delta}}{2}}\right). \tag{96}$$

Finally, we also have the PAC bound:

**Theorem 6.6.** *Assume A1, A2, and A3. For any $\mathcal{S} \in (\mathcal{X} \times \mathcal{Y})^m$, denote:*

$$\Lambda\alpha_{\mathcal{S}} := \underset{\Lambda\alpha \in \mathcal{F}_2}{\operatorname{argmin}} \mathcal{L}_{\mathcal{S}}(\Lambda\alpha),$$

$$\Lambda\alpha^* := \underset{\Lambda\alpha \in \mathcal{F}_2}{\operatorname{argmin}} \mathcal{L}_{\mathcal{D}}(\Lambda\alpha).$$

*Then with probability at least $1 - \delta$ over the choice $\mathcal{S} \sim \mathcal{D}^m$, we have:*

$$\mathcal{L}_{\mathcal{D}}(\Lambda\alpha_{\mathcal{S}}) \leq \mathcal{L}_{\mathcal{D}}(\Lambda\alpha^\star) + \frac{4\rho\tau B_2}{\sqrt{m}}\left(1 + \sqrt{\frac{\log\frac{1}{\delta}}{2}}\right). \tag{97}$$

These bounds, Theorems 6.3 and 6.6, justify empirical risk minimization with a norm constraint. For RKHS weightings, the bound depends on the maximal RKHS norm $B_{\mathcal{H}}$ of the weight function, and is the rationale for minimizing the RKHS norm-regularized empirical risk in practice. Similarly, Theorem 6.6 justifies the $L^2(p)$-regularization of random feature methods in general (see Section 3.2).

## 6.2 Stability analysis of the stochastic functional gradient descent

The following theorem describes the convergence in expectation of Algorithm 3 with regard to the number of examples in the sample, and the number of iterations. (The proof is in appendix A.8.)

**Theorem 6.7.** *Assume A1, A2, A3. The output $\bar{\alpha}$ of Algorithm 3 is such that:*

$$\mathbb{E}\left[\mathcal{L}_{\mathcal{D}}(\Lambda\bar{\alpha})\right] - \min_{\Lambda\alpha \in \Lambda\mathcal{H}_B} \mathcal{L}_{\mathcal{D}}(\Lambda\alpha) \leq \frac{2\rho\theta B_{\mathcal{H}}}{\sqrt{m}} + \lambda_{\mathcal{H}}B_{\mathcal{H}}^2 + \frac{8\rho^2}{\lambda_{\mathcal{H}}m} + \frac{(\rho\kappa + 2\lambda_{\mathcal{H}}B_{\mathcal{H}})^2}{2\lambda_{\mathcal{H}}T}(1 + \log(T)), \tag{98}$$

*where the expectation is taken over the choice of sample and all sampled parameters and batches, that is $\mathcal{S} \sim \mathcal{D}^m, (w_1, \ldots, w_T) \sim p^T$, and $(\mathcal{B}_1, \ldots, \mathcal{B}_T) \sim U(\mathcal{S})^{T \times b}$. By choosing $\lambda_{\mathcal{H}} = \sqrt{\frac{8\rho^2}{B_{\mathcal{H}}^2 m}}$, we obtain:*

$$\mathbb{E}\left[\mathcal{L}_{\mathcal{D}}(\Lambda\bar{\alpha})\right] - \min_{\Lambda\alpha \in \Lambda\mathcal{H}_B} \mathcal{L}_{\mathcal{D}}(\Lambda\alpha) \leq \frac{\rho B_{\mathcal{H}}}{\sqrt{m}}\left(\sqrt{32} + 2\theta + \frac{m}{\sqrt{32}T}\left(\kappa + \sqrt{\frac{32}{m}}\right)^2 (1 + \log(T))\right) \tag{99}$$

$$\in \mathcal{O}\left(\frac{1}{\sqrt{m}} + \frac{\sqrt{m}}{T}\log T\right).$$

This bound guarantees convergence to the optimal given enough data and iterations. While it is a result on the expected risk, it can be turned into a probabilistic bound via Markov's inequality or other more sophisticated methods, such as the one given in exercise 13.1 of Shalev-Shwartz & Ben-David (2014).

Note also that, in practice, it is easier to use Equation (98) rather than Equation (99), since it allows using an arbitrary value for $\lambda_{\mathcal{H}}$ (chosen by cross-validation, for instance). It is also slightly less constraining, due to the ability to calculate the bound a posteriori with $B_{\mathcal{H}}$ replaced by the largest iterate norm. This avoids needing to specify the maximal norm $B_{\mathcal{H}}$ before learning, and skips the projection step of Algorithm 3.

## 6.3 Near-optimality of the empirical risk minimizer

As we argued in Section 4.5, the weight function which minimizes the regularized empirical risk $\mathcal{L}_{\mathcal{S}}^{\text{reg}}$ belongs to the space $\mathcal{H}_{\mathcal{S}} := \{\sum_{i=1}^m a_i\psi(x_i)\} \subset \mathcal{H}$, where $\psi(x_i) = \mathbb{E}_{w \sim p}[\phi(w, x_i)\mathcal{K}(w, \cdot)] \in \mathcal{H}$. Furthermore, the space $\Lambda\mathcal{H}_{\mathcal{S}}$ is a subset of the RKHS $\mathcal{H}_{\mathcal{X}}$. However, our inability to exactly evaluate the kernel $\mathcal{K}_{\mathcal{X}}$ means that we do not directly have access to $\Lambda\mathcal{H}_{\mathcal{S}}$. In practice, the weight function will instead take the form $\sum_{t=1}^T a_t\mathcal{K}(w_t, \cdot)$, where the set $\mathcal{U} := \{w_1, \ldots, w_T\}$ has been sampled from $p^T$. This is the case for all the learning algorithms laid out in Section 5. We denote the space of available weight functions as:

$$\mathcal{H}_{\mathcal{U}} := \left\{\sum_{t=1}^T a_t\mathcal{K}(w_t, \cdot)\right\}. \tag{100}$$

The following theorem is a guarantee for the empirical risk minimizer in $\Lambda\mathcal{H}_\mathcal{U}$. It shows that the predictor class $\Lambda\mathcal{H}_\mathcal{U}$ allows approximating $\Lambda\mathcal{H}_\mathcal{S}$, and the approximation quality increases with the sampling size $T$.

**Theorem 6.8.** *Assume A1, A2. For any $\alpha = \sum_{i=1}^m a_i \psi(x_i) \in \mathcal{H}_\mathcal{S}$, define the pseudo-norm of $\alpha$ as:*

$$[\![\alpha]\!] := \sum_{i=1}^m |a_i|. \tag{101}$$

*For any $\mathcal{S} \in (\mathcal{X} \times \mathcal{Y})^m$, there exists*

$$\alpha_\mathcal{S} \in \mathcal{H}_\mathcal{S} \cap \operatorname*{argmin}_{\alpha \in \mathcal{H}:[\![\alpha]\!] \leq B_1} \mathcal{L}_\mathcal{S}(\Lambda\alpha). \tag{102}$$

*Further consider $\Lambda\bar{\alpha}_{\mathcal{S},\mathcal{U}}$ the minimizer of the empirical risk $\mathcal{L}_\mathcal{S}(\Lambda\bar{\alpha}_{\mathcal{S},\mathcal{U}})$ within $\Lambda\mathcal{H}_\mathcal{U}$, with the constraint that $\|\bar{\alpha}_{\mathcal{S},\mathcal{U}}\|_\mathcal{H} \leq \theta B_1 + \frac{\kappa B_1}{\sqrt{T}}\left(1 + 2\sqrt{2\log\frac{1}{\delta}}\right) =: B'$:*

$$\bar{\alpha}_{\mathcal{S},\mathcal{U}} := \operatorname*{argmin}_{\alpha \in \mathcal{H}_\mathcal{U}:\|\alpha\|_\mathcal{H} \leq B'} \mathcal{L}_\mathcal{S}(\Lambda\alpha). \tag{103}$$

*Then we have with probability at least $1 - \delta$ over the choices of $\mathcal{S} \sim \mathcal{D}^m$ and $\mathcal{U} \sim p^T$ that:*

$$\mathcal{L}_\mathcal{D}(\Lambda\bar{\alpha}_{\mathcal{S},\mathcal{U}}) \leq \mathcal{L}_\mathcal{S}(\Lambda\alpha_\mathcal{S}) + \rho\theta B_1 \left[ \frac{\kappa}{\sqrt{T}}\left(1 + \frac{2}{\sqrt{m}}\left(1 + \sqrt{\frac{\log\frac{2}{\delta}}{2}}\right)\right)\left(1 + 2\sqrt{2\log\frac{2}{\delta}}\right) \right.$$

$$\left. + \frac{2\theta}{\sqrt{m}}\left(1 + \sqrt{\frac{\log\frac{2}{\delta}}{2}}\right) \right] \tag{104}$$

$$= \mathcal{L}_\mathcal{S}(\Lambda\alpha_\mathcal{S}) + \tilde{\mathcal{O}}\left(\rho\theta B_1\left(\frac{\kappa}{\sqrt{T}} + \frac{\theta}{\sqrt{m}}\right)\right). \tag{105}$$

The proof is in Appendix A.9.

Theorem 6.8 is similar to Theorem 6.3, the PAC bound that we obtained through classical Rademacher complexity theory, but differs in a few meaningful ways. The most important difference is that Theorem 6.8 concerns the empirical risk minimizer in $\Lambda\mathcal{H}_\mathcal{U}$ (with RKHS norm at most $B'$), a predictor to which we have access in practice, while Theorem 6.3 concerns the empirical risk minimizer in $\Lambda\mathcal{H}_B$, which we usually cannot find in practice.[6] In other words, minimizing over $\mathcal{H}_\mathcal{U}$ is possible, while minimizing over $\mathcal{H}$ is not. Therefore Theorem 6.8 is much more relevant to practical applications and algorithms. It is also more informative, as it ties into the bound the number of terms $T$ used to write the weight function. As we see in Equation (105), the dataset size $m$ is tied to the constant $\theta$, and the number of sampled features $T$ is tied to $\kappa$. Since we know that $\theta \leq \kappa$ (Theorem 4.1), the bound suggests that having large $T$ is slightly more important than having large $m$.

The second important difference is to what $\mathcal{L}_\mathcal{D}(\Lambda\bar{\alpha}_{\mathcal{S},\mathcal{U}})$ is being compared. Interestingly, Theorem 6.8 looks at the empirical risk of the empirical risk minimizer $\Lambda\alpha_\mathcal{S}$ (with $\ell_1$-norm at most $B_1$), as opposed to Theorem 6.3, which looks at the true risk of the true risk minimizer $\Lambda\alpha^\star$ (with RKHS norm at most $B_\mathcal{H}$). We usually expect the empirical risk to be lower than the true risk, since the model only has to succesfully label a finite number of samples, rather than generalize to the entire population, an unambiguously harder task. To be noted that this comparison is not quite so straightforward, as the two theorems apply a different norm constraint on the weight function. We proved the relationship $\|\alpha_\mathcal{S}\|_\mathcal{H} \leq \theta[\![\alpha_\mathcal{S}]\!]$, which means that constraining the $\ell_1$-norm is a more restrictive assumption. This can degrade the term $\mathcal{L}_\mathcal{S}(\Lambda\alpha_\mathcal{S})$.

## 6.4 Comparison to other work

In this section, we compare our hypothesis class, assumptions and theoretical results to Rahimi & Recht (2009) and Bach (2017), two major contributions in the domain of random feature methods. Recall from

---

[6]The empirical risk minimizer in $\Lambda\mathcal{H}_B$ can be found in practice if the implicit kernel $\mathcal{K}_\mathcal{X}$ can be computed exactly.

Section 3 the two prediction classes from Rahimi & Recht (2009), the class $\mathcal{F}_2$ from Bach (2017), and the class $\Lambda \mathcal{H}_B$ introduced in this paper:

$$\mathcal{F}_p := \left\{ x \mapsto \Lambda\alpha(x) := \underset{w \sim p}{\mathbb{E}}\left[\alpha(w)\phi(w,x)\right] \,\middle|\, \|\alpha\|_\infty \leq B_\infty \right\}, \tag{21}$$

$$\hat{\mathcal{F}}_w := \left\{ x \mapsto \frac{1}{T}\sum_{t=1}^{T} a_t \phi(w_t, x) \,\middle|\, \forall t, |a_t| \leq B_\infty \right\}, \tag{22}$$

$$\mathcal{F}_2 := \left\{ x \mapsto \Lambda\alpha(x) = \underset{w \sim p}{\mathbb{E}}\left[\alpha(w)\phi(w,x)\right] \,\middle|\, \alpha \in L^2(p), \|\alpha\|_{L^2(p)} \leq B_2 \right\}, \tag{40}$$

$$\Lambda\mathcal{H}_B := \left\{ x \mapsto \Lambda\alpha(x) := \underset{w \sim p}{\mathbb{E}}\left[\alpha(w)\phi(w,x)\right] \,\middle|\, \alpha \in \mathcal{H}, \|\alpha\|_{\mathcal{H}} \leq B_{\mathcal{H}} \right\}. \tag{45}$$

The difference between the three main classes ($\mathcal{F}_p$, $\mathcal{F}_2$ and $\Lambda\mathcal{H}_B$) lies in the condition on the weight function. Where Rahimi & Recht (2009) assume that the weight function is bounded, and RKHS weightings assume that it belongs to an RKHS, Bach (2017) simply postulate that it is a square-integrable function with regard to the distribution $p$. Note that this last assumption is the most general: any bounded function $\alpha$ or RKHS function is square-integrable with respect to a probability measure. [7]

Boundedness (the random kitchen sinks assumption) limits the subspace of $L^2(p)$ available for selecting a weight function. However, we can show that such is not the case for RKHS weightings when the kernel $\mathcal{K}$ is $L^2$-universal, [8] which is the case for the Gaussian kernel.

**Lemma 6.1.** *Assume A1, A2. Further assume that $\mathcal{K}$ is $L^2$-universal (e.g. the Gaussian kernel). Consider any $\alpha_p \in L^2(p)$. Then for all $\varepsilon > 0$, there exists $\alpha_{\mathcal{H}} \in \mathcal{H}$ such that:*

$$\|\alpha_{\mathcal{H}}\|_{L^2(p)} \leq \|\alpha_p\|_{L^2(p)} + \frac{\varepsilon}{\rho\tau}, \tag{106}$$

*and:*

$$|\mathcal{L}_{\mathcal{D}}(\Lambda\alpha_p) - \mathcal{L}_{\mathcal{D}}(\Lambda\alpha_{\mathcal{H}})| < \varepsilon. \tag{107}$$

*Proof.* Since $\mathcal{K}$ is $L^2$-universal, its RKHS $\mathcal{H}$ is dense in $L^2(p)$. Therefore, because $\alpha_p \in L^2(p)$, we can find $\alpha_{\mathcal{H}} \in \mathcal{H}$ such that:

$$\|\alpha_p - \alpha_{\mathcal{H}}\|_{L^2(p)} < \frac{\varepsilon}{\rho\tau}, \tag{108}$$

where $\rho$ is the Lipschitz constant of the loss. Equation (106) follows immediately from the triangle inequality, and we obtain Equation (107) because:

$$
\begin{aligned}
|\mathcal{L}_{\mathcal{D}}(\Lambda\alpha_p) - \mathcal{L}_{\mathcal{D}}(\Lambda\alpha_{\mathcal{H}})| &= \left| \underset{(x,y)\sim\mathcal{D}}{\mathbb{E}}\left[\ell(\Lambda\alpha_p(x),y) - \ell(\Lambda\alpha(x),y)\right] \right| \\
&\leq \rho \underset{(x,y)\sim\mathcal{D}}{\mathbb{E}}\left[|\Lambda\alpha_p(x) - \Lambda\alpha(x)|\right] && (\ell \text{ is } \rho\text{-Lipschitz}) \\
&\leq \rho\tau\|\alpha_p - \alpha\|_{L^2(p)} && (\text{Lemma 4.1}) \\
&< \varepsilon.
\end{aligned}
$$

$\square$

---

[7] RKHS weightings weight functions are square-integrable because we assume that $\mathbb{E}_{w\sim p}[\mathcal{K}(w,w)] < \infty$. (See assumption A1.) By Equation (57) in Section 4.2, we have $\|\alpha\|_{L^2(p)}^2 \leq \|\alpha\|_{\mathcal{H}}^2 \mathbb{E}_{w\sim p}[\mathcal{K}(w,w)]$.

[8] From Sriperumbudur et al. (2011), a measurable and bounded positive definite kernel $\mathcal{K} : \mathcal{W} \times \mathcal{W} \to \mathbb{R}$ is $L^2$-**universal** if its corresponding RKHS $\mathcal{H}$ is dense in $L^2(\mu)$ for all Borel probability measures $\mu$ on $\mathcal{W}$. (A Borel probability measure is one defined on all open sets.) In our situation, if $p$ is a Borel probability measure, then for all $\varepsilon > 0$ and all $f \in L^2(p)$, there exists $h \in \mathcal{H}$ such that $\|f - h\|_{L^2(p)} < \varepsilon$. Radial kernels, such as the Gaussian kernel, are $L^2$-universal, and the Gaussian distribution is a Borel probability measure.

The implication of this lemma is that any predictor $\Lambda\alpha_p \in \mathcal{F}_2$ (this includes predictors in $\mathcal{F}_p$) can be successfully approximated by an RKHS weighting $\Lambda\alpha_{\mathcal{H}}$, where $\alpha_{\mathcal{H}}$ has an $L^2(p)$-norm almost equal to that of $\alpha_p$. Therefore, the approximation capability of RKHS weightings is in theory equivalent to the unrestricted assumption that the weight function simply be square-integrable. And as Bach (2017) showed (Proposition 6 and its derivatives), the approximation capability of $\mathcal{F}_2$ is high, and generalization is guaranteed by Theorem 6.5.

However, there is no guarantee that $\|\alpha_{\mathcal{H}}\|_{\mathcal{H}}$ will be small. In fact, $\|\alpha_{\mathcal{H}}\|_{\mathcal{H}}$ can be arbitrarily high despite its small $L^2(p)$-norm.[9] This is a problem, since the learning algorithms that we presented all regularize the RKHS norm, either directly (Algorithms 3 and Algorithm 4) or indirectly (Algorithm 5).[10] By setting any regularization parameter larger than 0, we set an implicit upper bound on the RKHS norm of the weight function that can be returned by these algorithms. If this implicit upper bound is smaller than the unknown, and potentially extremely large, $\|\alpha_{\mathcal{H}}\|_{\mathcal{H}}$, then the algorithm will not find this $\alpha_{\mathcal{H}} \in \mathcal{H}$ close to $\alpha_p$. This suggests that the RKHS norm regularization, as well as Theorems 6.2, 6.7, 6.8, which depend on the RKHS norm of the weight function, are not best suited to ensure finding the best possible RKHS weighting weight function. Ideally, the RKHS norm regularization should be replaced by an $L^2(p)$-norm regularization. Since this is not a trivial task, we leave this for future work.

In principle, however, there exists a combination of a distribution $p$, ($L^2$-universal) kernel $\mathcal{K}$, and base predictor $\phi$ which optimizes the tradeoff between the risk $\mathcal{L}_{\mathcal{D}}(\Lambda\alpha_{\mathcal{H}})$, and the RKHS norm $\|\alpha_{\mathcal{H}}\|_{\mathcal{H}}$ of the optimal weight function $\alpha_{\mathcal{H}} \in \mathcal{H}$ of norm at most $B_{\mathcal{H}}$. This suggests trying as many instantiations as possible, and using cross-validation to choose the best parameters for the distribution and kernel, with the goal of finding the best possible combination of $(\mathcal{W}, \phi, \mathcal{K}, p)$ for a given problem.

This highlights a key difference with Bach (2017). Proposition 6 of Bach (2017), their core result, provides a powerful approximation guarantee, but is limited to one specific instantiation: the ReLU as the base predictor $\phi$ (or an integer power of the ReLU), and the uniform distribution $p$ on the sphere. Proposition 6 of Bach (2017) does not preclude the existence of an RKHS weighting instantiation $(\mathcal{W}, \phi, \mathcal{K}, p)$ which can perform better on a given problem. The generality of our results, which apply to any valid instantiation, allows searching through the space of instantiations for the best one for the problem at hand.

Additionally, Bach (2017) is mostly concerned with the theory of learning with random features, and defers to Rahimi & Recht (2009) for practical applications. We therefore take the time to make a few more observations comparing RKHS weightings to random kitchen sinks.

The most meaningful difference concerns the weight functions allowed by the assumptions. If the kernel $\mathcal{K}$ is unbounded, then the weight functions in $\mathcal{H}$ can also be unbounded[11]. Such a choice of kernel therefore gives us access to weight functions that are disallowed in Rahimi & Recht (2009). This has the potential of increasing the expressivity of the predictor class, thus improving the best predictor in the class. We investigate this possibility in our experiments (Section 7.3), where we test the performance of all the instantiations of Table 2. This includes RWExpSign and RWExpRelu, which use the unbounded exponential kernel.

Another potentially meaningful difference between RKHS weightings and random kitchen sinks is that RKHS weighting algorithms yield an actual weight function. One advantage that this has is the ability to approximate the $L^2(p)$-norm of the weight function, yielding better bound values. By contrast, the RKS algorithm only learns the coefficients $(a_1, \ldots, a_T)$ for the sampled features $(w_1, \ldots, w_T)$. Since these coefficients correspond to the weight function values $\alpha(w_t)$ at those features, the $L^2(p)$-norm must be loosely upper bounded by the maximal coefficient (the sup-norm).[12] (Note that we do not leverage the existence of the weight

---

[9]For example, take $\mathcal{K}(w, u) = \exp\left(-\|w - u\|^2\right)$, and $\alpha = c\mathcal{K}(cw, \cdot)$ for some $w \in \mathbb{R}^n$ and $c \in \mathbb{R}$, and $p$ the standard normal distribution. The RKHS norm of $\alpha$ is equal to $c$, and is therefore linear in $c$. However, $p(cw)$ decreases exponentially in $c$. The $L^2(p)$-norm of $\alpha$ therefore decreases exponentially in $c$. As $\|\alpha\|_{\mathcal{H}}$ tends to infinity, $\|\alpha\|_{L^2(p)}$ tends to 0.

[10]The Lasso fit of the coefficients regularizes the $\ell_1$-norm of the coefficients. We have $\|\alpha\|_{\mathcal{H}}^2 = \sum_{i=1}^T \sum_{j=1}^T a_i a_j \mathcal{K}(w_i, w_j) \leq \|\mathbf{a}\|_1^2 \max_i \mathcal{K}(w_i, w_i)$. The $\ell_1$-regularization is therefore implicitly an RKHS-norm regularization as well.

[11]Take for example $\mathcal{K}(w, u) := \exp(\langle w, u \rangle)$, and $\alpha = \mathcal{K}(w, \cdot)$ for some $w \in \mathcal{W} = \mathbb{R}^n$. Then $\alpha$ is not bounded as a function of $\mathcal{W}$, since $\alpha(cw) = \exp\left(c\|w\|_2^2\right) \to \infty$ when $c \to \infty$.

[12]To be more precise, we have from Equation (22) that $\max_t |a_t| \leq B_\infty$. We can approximate $\|\alpha\|_{L^2(p)} \leq B_\infty \approx \max_t |a_t|$.

Table 7: Summary of the various assumptions on the weight function and their implications.

|  | Bach (2017) | Rahimi & Recht (2009) | RKHS weightings |
|---|---|---|---|
| Basic assumption | $\alpha \in L^2(p)$ | $\alpha \in L^\infty(\mathcal{W})$ | $\alpha \in \mathcal{H}$ |
| Constraint | $\|\alpha\|_{L^2(p)} \leq B_2$ | $\|\alpha\|_\infty \leq B_\infty$ | $\|\alpha\|_{\mathcal{H}} \leq B_{\mathcal{H}}$ |
| Further assumptions | $\phi$ is power of ReLU | $|\phi(w,x)| \leq 1$ | $\kappa < \infty$ |
|  | $p$ uniform on the sphere |  | Analytical expectations |
| Unbounded $\alpha$ | Yes | No | Yes |
| Unbounded $\phi$ | Yes | No | Yes |
| Approximation guarantees | Often in $\mathcal{O}(1/\sqrt[4]{m})$ | None | None |
| Generalization gap | In $\mathcal{O}\left(\frac{B_2}{\sqrt{m}}\right)$ | In $\mathcal{O}\left(\frac{B_\infty}{\sqrt{m}}\right)$ | In $\mathcal{O}\left(\frac{B_{\mathcal{H}}}{\sqrt{m}}\right)$ |
| Dedicated algorithms | None | RKS | Algorithms 3, 4, 5 |
| Algorithm convergence | N/A | $\mathcal{O}\left(\frac{1}{\sqrt{m}} + \frac{1}{\sqrt{T}}\right)$ | $\mathcal{O}\left(\frac{1}{\sqrt{m}} + \frac{1}{\sqrt{T}}\right)$ |

function further in this paper. It would be interesting to explore in further work other potential uses for the weight function.)

Finally, let us directly compare the theoretical guarantees that we proved in this Section to those of Rahimi & Recht (2009) and Bach (2017). Theorem 6.7 shows a convergence rate of $\mathcal{O}\left(\frac{1}{\sqrt{m}} + \frac{\sqrt{m}}{T}\log T\right)$ for Algorithm 3, where $m$ is the number of samples and $T$ is the number of random features, compared to $\mathcal{O}\left(\frac{1}{\sqrt{m}} + \frac{1}{\sqrt{T}}\right)$ for Algorithm 1 of Rahimi & Recht (2009). When $T$ is large (commensurate to $m$), these bounds have the same complexity, up to a log factor. Theorem 6.8 reveals the same $\mathcal{O}\left(\frac{1}{\sqrt{m}} + \frac{1}{\sqrt{T}}\right)$ rate as Rahimi & Recht (2009), but with much more explicit constants. Indeed, the simple assumption that $|\phi(w,x)| \leq 1$ of Rahimi & Recht (2009) is replaced by the two much tighter instantiation-dependent constants $\theta$ and $\kappa$.

Table 7 summarizes this long comparison of RKHS weightings with previous work on random features. In short, the predictor class $\Lambda\mathcal{H}_B$ examined in this work appears expressive and flexible, but obtaining good performance in practical applications will likely require trying many instantiations, through the choices of the building blocks $(\mathcal{W}, \phi, \mathcal{K}, p)$ and their hyperparameters.

# 7 Experiments

We perform various experiments on a variety of binary classification and regression datasets taken from the UC Irvine Machine Learning Repository (Dua & Graff, 2017), Scikit-learn (Pedregosa et al., 2011), as well as MNIST (Deng, 2012). Dataset information is summarized in Tables 8 and 9. (Note that **mnist17** consists of only the digits 1 and 7 of the MNIST dataset.)

Table 8: Binary classification datasets used in this paper.

|  | Training size | Test size | Dimensionality | Source (clickable) |
|---|---|---|---|---|
| adults | 32561 | 16281 | 108 | UCI |
| cancer | 426 | 143 | 30 | Scikit-learn |
| marketing | 33908 | 11303 | 47 | UCI |
| mnist17 | 13007 | 2163 | 784 | Kaggle |
| phishing | 8291 | 2764 | 30 | UCI |
| skin | 183792 | 61265 | 3 | UCI |
| telescope | 14265 | 4755 | 10 | UCI |

In Section 7.1, we compare the three RKHS weightings learning algorithms that we presented in Section 5. The quick takeaway is that the least squares fit (Algorithm 4) is both fast and accurate. The Lasso fit

Table 9: Regression datasets used in this paper.

|  | Training size | Test size | Dimensionality | Source (clickable) |
|---|---|---|---|---|
| abalone | 3132 | 1045 | 10 | UCI |
| diabetes | 331 | 111 | 10 | Scikit-learn |
| housing | 15480 | 5160 | 8 | Scikit-learn |
| concrete | 772 | 258 | 8 | UCI |
| conductivity | 15947 | 5316 | 81 | UCI |
| wine | 133 | 45 | 13 | UCI |

(Algorithm 5) is accurate, but slow. The Stochastic Functional Gradient Descent (Algorithm 3) is slow and inaccurate (in comparison to the other two).

In Section 7.2, we illustrate the need for and efficacy of the method described in Section 4.6 for setting RKHS weighting hyperparameters.

In Section 7.3, we thoroughly compare RKHS weightings to random kitchen sinks. In that experiment, RKHS weightings appear to have an edge, achieving slightly better performance on average than random kitchen sinks.

In Section 7.4, we add other conventional algorithms (AdaBoost, SVM) to the comparison. All algorithms perform quite similarly.

In Section 7.5, we observe that RKHS weightings perform significantly better than random kitchen sinks when the number of random features is small, and the gap closes as the number of features increases.

Additional details about the experiments, such as the cross-validation parameters used, can be found in Appendix C.

## 7.1 Comparison of learning algorithms for RKHS weightings

Our first order of business is to compare the three learning algorithms that we presented in Section 5: the Stochastic Functional Gradient Descent (SFGD) (Algorithm 3), the least squares fit of the coefficients (Algorithm 4), and the Lasso fit of the coefficients (Algorithm 5). The goal is to choose the best algorithm for further experiments.

Figure 1 compares their training and test performance on a simple binary classification dataset (digits 1 and 7 of MNIST), as well as their training times. A fourth figure shows the number of nonzero coefficients of the model after training to evaluate the sparsity of the models produced by the Lasso fit.

Performance-wise, Figures 1a and 1b show that the SFGD appears to require more sampled features to converge to a low error than the least squares fit and Lasso fit. The least squares fit was also far faster than both the SFGD and the Lasso fit algorithm (Figure 1c). The Lasso fit was in fact the slowest algorithm, and also the one with the highest variance in training time. The reason is that the running time of the Lasso is highly dependent on the regularization parameter $\lambda_1$; larger values of $\lambda_1$ lead to sparser models and shorter training times, but potentially lower accuracy. The variability of the training times in Figure 1c is due to different values of $\lambda_1$ being chosen by cross-validation.

Figure 1d shows how the Lasso fit can produce much smaller models when using an explicit $\ell_1$-regularization on the coefficients. If the higher training cost is not deemed an issue in a given circumstance, then it is an advantageous alternative to the least squares fit, as it should produce a smaller model with similar performance.

Finally, Figure 1c appears to suggest that the SFGD has a higher computational complexity than the least squares fit, while the opposite is true. Figure 2 compares the training times of these two algorithms in more detail, with a breakdown of the main computational costs. For the SFGD, the main cost always comes from calculating the functional gradient approximation (see Table 5). For the least squares fit, calculating

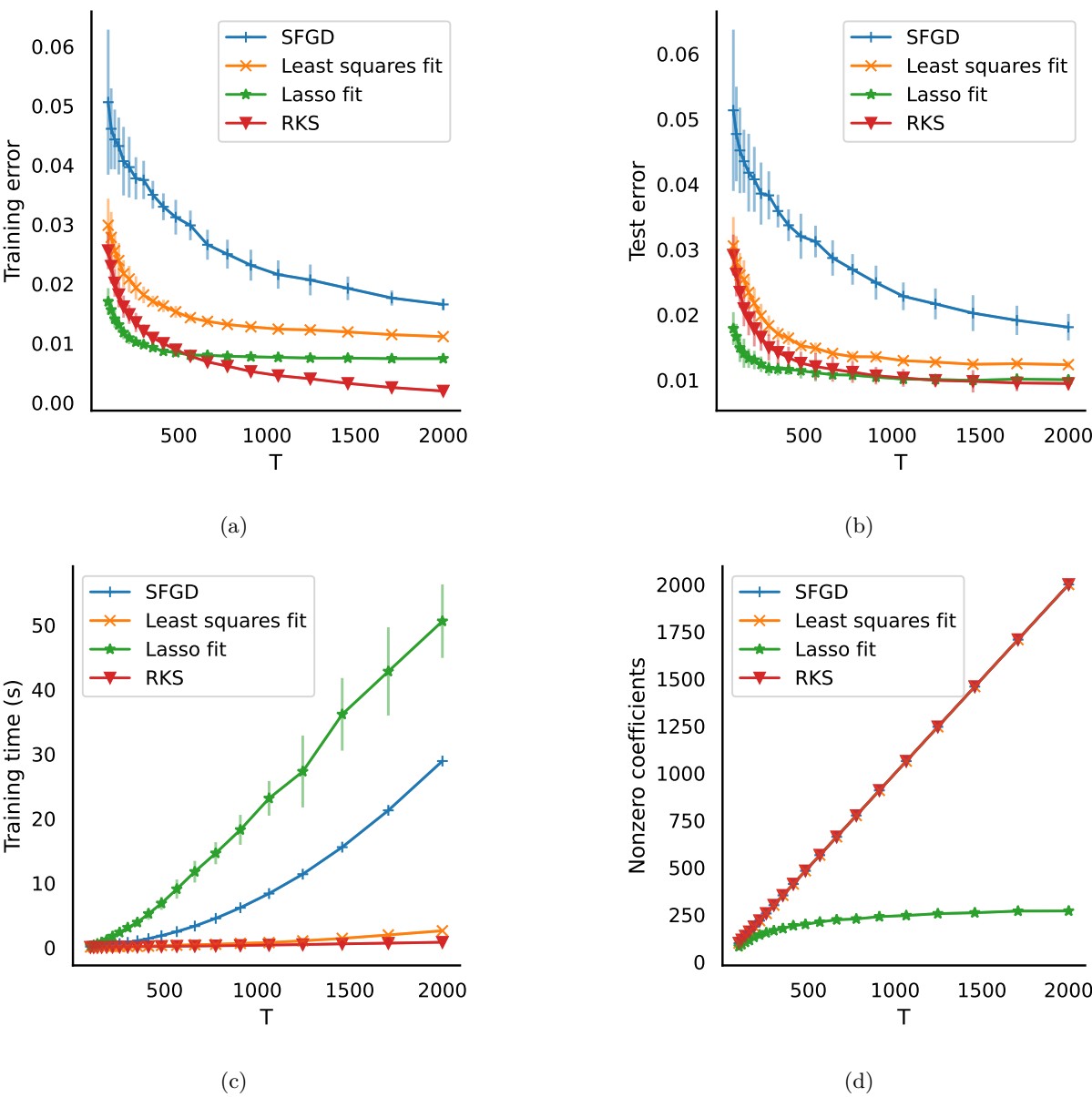

Figure 1: Comparison of three algorithms for learning RKHS weightings: **SFGD** (Algorithm 3, with a batch size $b = 100$), **least squares fit** of the coefficients (Algorithm 4), and **Lasso fit** of the coefficients (Algorithm 5). The random kitchen sinks algorithm (**RKS**) is included for completeness. The regularization parameters were set to $\lambda_{\mathcal{H}} = \lambda_2 = \lambda_1 = 10^{-6}$. Each algorithm was tasked with learning the model using the RWSign instantiation (simply using the sign base predictor for RKS) on **mnist17**. Every point is the average of 10 independently seeded runs. The error bars represent one standard deviation. The SFGD provides slightly poorer performance than the other three algorithms. The least squares fit and RKS are by far the fastest methods.

the various matrices which make up the linear problem will always be the main cost for low values of $T$, but solving the problem will eventually dominate the execution for high $T$, since it has $\mathcal{O}(T^3)$ complexity. However, reaching this regime requires a number of random features which is unnecessarily large for most applications.

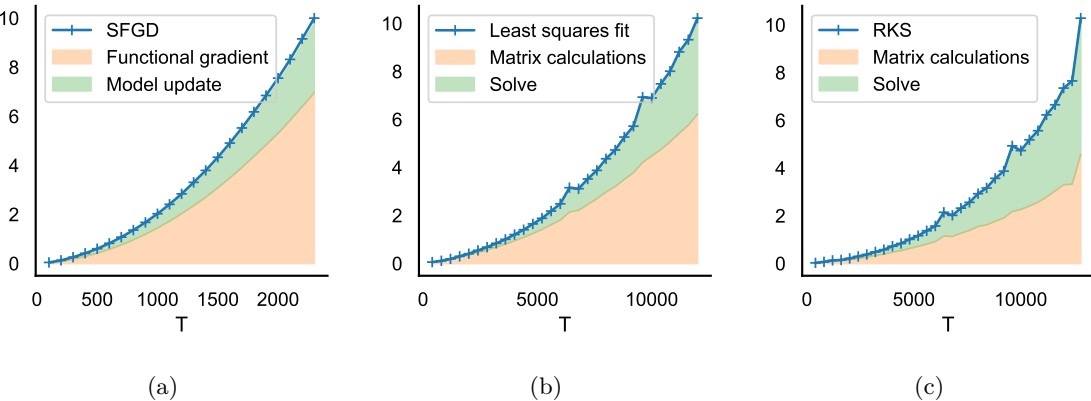

Figure 2: Breakdown of the training time (in seconds) of the **Stochastic Functional Gradient Descent** (Algorithm 3, with a batch size $b = 100$), **least squares fit** of the coefficients (Algorithm 4), and **RKS** (Algorithm 1). The regularization parameters were set to $\lambda_{\mathcal{H}} = \lambda_2 = 10^{-6}$. Each algorithm was tasked with learning the model using the RWSign instantiation on **abalone**. Every point is the average of 10 independently seeded runs. For SFGD, **Model update** includes the projection step.

To summarize, the Stochastic Functional Gradient Descent is theoretically interesting, but its practical performance is much worse than the alternative algorithms. The Lasso can generate sparse predictors, but at a high computational cost. Therefore, we will use the least squares fit (Algorithm 4) for the remaining experiments.

## 7.2 Curse of dimensionality

In Section 4.6, we described how to circumvent the curse of dimensionality that affects RKHS weightings by setting the distribution and kernel parameters judiciously. Figures 3 and 4 illustrate why such an approach is necessary.

We first notice that RWStumps does not suffer from the curse of dimensionality. This is due to its fixed two-dimensional space of random features. We see how a wide range of values of $\gamma$ can yield low training MSE, and the MSE varies smoothly with $\gamma$. For RWSign and RWRelu, however, the picture is different. If $\gamma$ is too low, no learning is possible, because the model outputs are vanishingly small. At some threshold value of $\gamma$, which depends on the dimensionality, the performance improves dramatically. The model then reaches low training MSE for a modest interval of values of $\gamma$. Finally, the MSE increases again, since the model becomes closer and closer to constant when $\gamma$ is too large. The method described in Section 4.6 allows automatically finding values of $\gamma$ which fall within the goldilock zone.

## 7.3 Comparison to random kitchen sinks

This second experiment is an in-depth performance comparison of RKHS weightings and random kitchen sinks. Both models were fitted on various datasets, using all the instantiations we presented in Table 2. (For a given RKHS weighting instantiation $(\mathcal{W}, \phi, \mathcal{K}, p)$, the random kitchen sinks inherits all but the kernel $\mathcal{K}$.) These results can be found in Tables 10 and 11. These tables also include the values of the bounds on the generalization gap found in Theorems 6.2 and 6.5. These latter bound values (Theorem 6.5) had to be approximated, since the $L^2(p)$-norm is not known exactly (as opposed to the RKHS norm required by Theorem 6.2). For an RKHS weighting $\Lambda\alpha$, the $L^2(p)$-norm squared of the weight function $\alpha$ was approximated as $\|\alpha\|_{L^2(p)}^2 \approx \frac{1}{T} \sum_{t=1}^T \alpha(u_t)^2$ for a newly sampled set of features $(u_1, \ldots, u_T)$. The constant

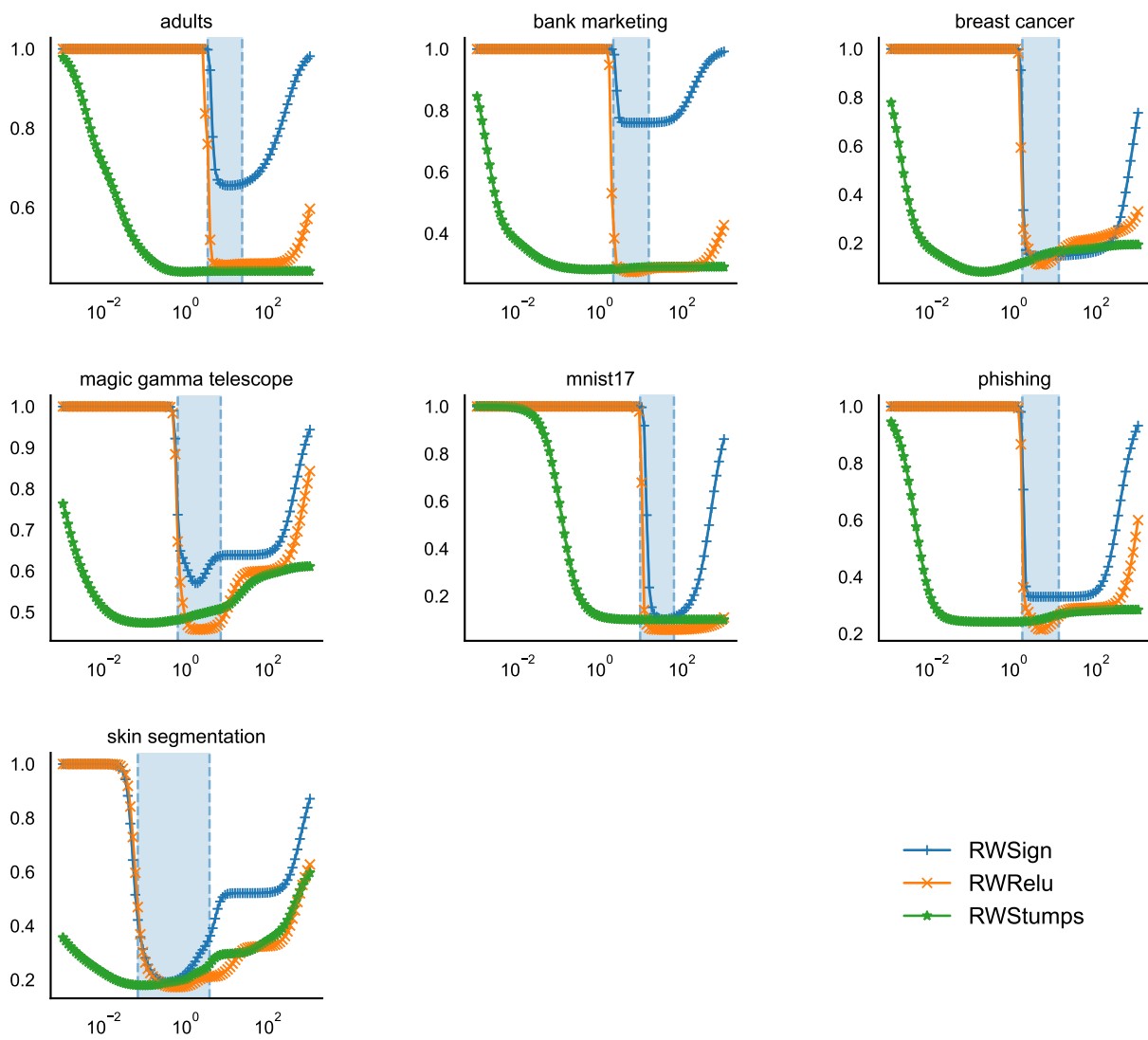

Figure 3: Training mean squared error of RKHS weightings with regard to the hyperparameter $\gamma$ of the Gaussian kernel. Values of $\gamma$ range from $10^{-3}$ to $10^3$. Models were trained by Algorithm 4 with other parameters fixed at $\sigma = 1$, $T = 1000$, $\lambda_{\mathcal{H}} = 10^{-6}$. The shaded area identifies the interval of values of $\gamma$ used in practice for RWSign and RWRelu in our other experiments using the method described in Section 4.6. Each point is the average of 10 independently seeded runs. Error bars represent one standard deviation.

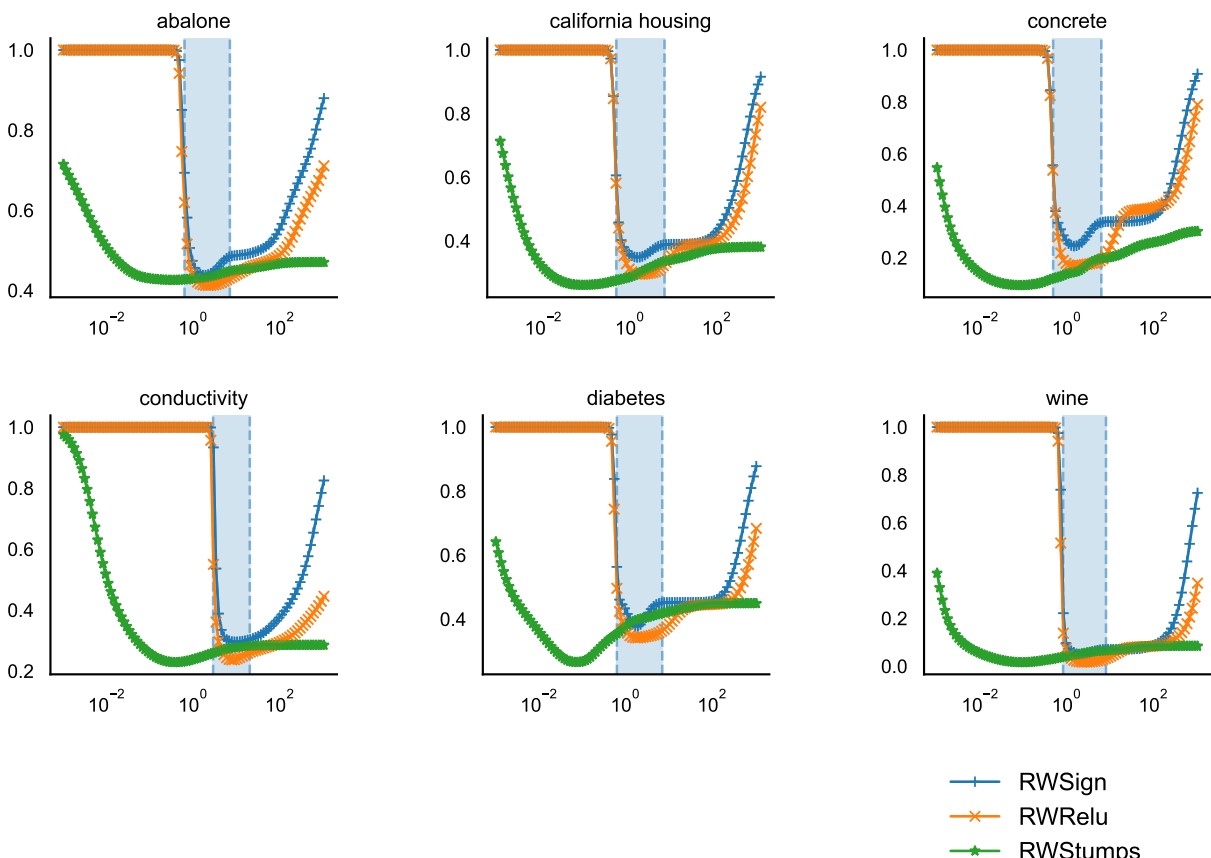

Figure 4: Training mean squared error of RKHS weightings with regard to the hyperparameter $\gamma$ of the Gaussian kernel. Values of $\gamma$ range from $10^{-3}$ to $10^3$. Models were trained by Algorithm 4 with other parameters fixed at $\sigma = 1$, $T = 1000$, $\lambda_{\mathcal{H}} = 10^{-6}$. The shaded area identifies the interval of values of $\gamma$ used in practice for RWSign and RWRelu in our other experiments using the method described in Section 4.6. Each point is the average of 10 independently seeded runs. Error bars represent one standard deviation.

$\tau := \sup_{x \in \mathcal{X}} \sqrt{\mathbb{E}_{w \sim p}\left[\phi(w, x)^2\right]}$ was approximated as the maximum over the test set of $\sqrt{\frac{1}{T} \sum_{t=1}^{T} \phi(u_t, x)^2}$, i.e. $\tau \approx \max_{(x,y) \in \mathcal{S}} \sqrt{\frac{1}{T} \sum_{t=1}^{T} \phi(u_t, x)^2}$.

Table 10: Classification performance comparison of RKHS weightings (RW) to random kitchen sinks (RKS). For each combination of dataset and instantiation, an RKHS weighting was learned by cross-validation and using Algorithm 4 (the least squares fit) in order to select the best model hyperparameters. The sampling size $T$ was 500, and each value in the table is the average of 10 independently seeded runs. Train error and Test error are the proportion of incorrectly classified examples on the training and test sets. Th. 6.2 is the 95% generalization gap (the difference between Test MSE and Train MSE) guaranteed by Theorem 6.2. Th. 6.5 is the 95% generalization gap guaranteed by Theorem 6.5.

| Dataset | Instantiation | Train error RKS | Train error RW | Test error RKS | Test error RW | Train MSE RKS | Train MSE RW | Test MSE RKS | Test MSE RW | Th. 6.2 RW | Th. 6.5 RW | Train time (s) RKS | Train time (s) RW |
|---|---|---|---|---|---|---|---|---|---|---|---|---|---|
| adults | RWExpRelu | 0.152 | 0.150 | 0.155 | 0.152 | 0.430 | 0.425 | 0.442 | 0.437 | $>10^4$ | 5.49 | 0.347 | 0.901 |
| | RWExpSign | 0.191 | 0.194 | 0.198 | 0.198 | 0.553 | 0.579 | 0.570 | 0.581 | $>10^{12}$ | $>10^7$ | 0.384 | 0.520 |
| | RWRelu | 0.152 | 0.149 | 0.155 | 0.152 | 0.430 | 0.422 | 0.442 | 0.434 | $>10^8$ | $>10^5$ | 0.348 | 0.900 |
| | RWSign | 0.191 | 0.216 | 0.198 | 0.220 | 0.553 | 0.649 | 0.570 | 0.649 | $>10^3$ | 52.6 | 0.389 | 0.536 |
| | RWStumps | 0.148 | 0.147 | 0.150 | **0.147** | 0.432 | 0.431 | 0.434 | 0.432 | 61.3 | 1.45 | 0.225 | 0.386 |
| cancer | RWExpRelu | 0.014 | 0.013 | 0.029 | 0.030 | 0.131 | 0.109 | 0.200 | 0.186 | $>10^2$ | 0.967 | 0.029 | 0.059 |
| | RWExpSign | 0.011 | 0.012 | 0.041 | 0.048 | 0.074 | 0.108 | 0.157 | 0.172 | $>10^9$ | $>10^5$ | 0.033 | 0.048 |
| | RWRelu | 0.014 | 0.015 | 0.029 | **0.026** | 0.131 | 0.136 | 0.200 | 0.198 | $>10^5$ | $>10^2$ | 0.031 | 0.063 |
| | RWSign | 0.011 | 0.010 | 0.041 | 0.040 | 0.074 | 0.078 | 0.157 | 0.152 | $>10^6$ | $>10^4$ | 0.033 | 0.051 |
| | RWStumps | 0.012 | 0.017 | 0.032 | 0.036 | 0.105 | 0.110 | 0.154 | 0.148 | 72.3 | 2.41 | 0.017 | 0.040 |
| marketing | RWExpRelu | 0.094 | 0.093 | 0.100 | 0.100 | 0.268 | 0.264 | 0.286 | 0.281 | $>10^4$ | 2.83 | 0.352 | 0.909 |
| | RWExpSign | 0.139 | 0.103 | 0.148 | 0.110 | 0.458 | 0.328 | 0.482 | 0.349 | $>10^{11}$ | $>10^7$ | 0.389 | 0.508 |
| | RWRelu | 0.094 | 0.092 | 0.100 | **0.099** | 0.268 | 0.262 | 0.286 | 0.280 | $>10^7$ | $>10^3$ | 0.347 | 0.893 |
| | RWSign | 0.139 | 0.132 | 0.148 | 0.138 | 0.458 | 0.516 | 0.482 | 0.528 | $>10^6$ | $>10^4$ | 0.388 | 0.537 |
| | RWStumps | 0.097 | 0.097 | 0.102 | 0.101 | 0.280 | 0.277 | 0.291 | 0.288 | 28.1 | 0.657 | 0.306 | 0.521 |
| mnist17 | RWExpRelu | 0.006 | 0.005 | 0.012 | **0.011** | 0.057 | 0.051 | 0.162 | 0.645 | $>10^3$ | 6.8 | 0.303 | 0.578 |
| | RWExpSign | 0.010 | 0.008 | 0.014 | 0.013 | 0.078 | 0.073 | 0.086 | 0.078 | $>10^8$ | $>10^3$ | 0.196 | 0.271 |
| | RWRelu | 0.006 | 0.007 | 0.012 | 0.012 | 0.057 | 0.069 | 0.162 | 0.164 | $>10^5$ | $>10^3$ | 0.303 | 0.632 |
| | RWSign | 0.010 | 0.008 | 0.014 | 0.013 | 0.078 | 0.074 | 0.086 | 0.078 | $>10^4$ | 11.2 | 0.221 | 0.364 |
| | RWStumps | 0.008 | 0.008 | 0.013 | 0.013 | 0.074 | 0.069 | 0.080 | 0.075 | $>10^2$ | 5.44 | 0.151 | 0.226 |
| phishing | RWExpRelu | 0.049 | 0.046 | 0.062 | 0.058 | 0.199 | 0.182 | 0.233 | 0.214 | $>10^3$ | 1.7 | 0.094 | 0.232 |
| | RWExpSign | 0.061 | 0.051 | 0.072 | 0.065 | 0.254 | 0.211 | 0.281 | 0.245 | $>10^9$ | $>10^6$ | 0.115 | 0.157 |
| | RWRelu | 0.049 | 0.043 | 0.062 | **0.055** | 0.199 | 0.172 | 0.233 | 0.205 | $>10^6$ | $>10^3$ | 0.095 | 0.225 |
| | RWSign | 0.061 | 0.056 | 0.072 | 0.067 | 0.254 | 0.228 | 0.281 | 0.251 | $>10^5$ | $>10^3$ | 0.112 | 0.165 |
| | RWStumps | 0.069 | 0.069 | 0.075 | 0.075 | 0.241 | 0.241 | 0.249 | 0.249 | 18.4 | 0.634 | 0.094 | 0.151 |
| skin | RWExpRelu | 0.015 | 0.023 | **0.016** | 0.023 | 0.088 | 0.121 | 0.090 | 0.121 | $>10^6$ | $>10^4$ | 1.601 | 4.641 |
| | RWExpSign | 0.039 | 0.059 | 0.040 | 0.060 | 0.128 | 0.206 | 0.132 | 0.209 | $>10^7$ | $>10^5$ | 1.867 | 2.642 |
| | RWRelu | 0.015 | 0.029 | **0.016** | 0.029 | 0.088 | 0.125 | 0.090 | 0.126 | $>10^5$ | $>10^3$ | 1.598 | 5.373 |
| | RWSign | 0.039 | 0.049 | 0.040 | 0.050 | 0.128 | 0.144 | 0.132 | 0.147 | $>10^3$ | 92.7 | 1.874 | 3.449 |
| | RWStumps | 0.040 | 0.040 | 0.040 | 0.040 | 0.172 | 0.168 | 0.173 | 0.169 | 50.0 | 1.44 | 1.321 | 2.743 |
| telescope | RWExpRelu | 0.131 | 0.125 | 0.137 | **0.135** | 0.415 | 0.397 | 0.442 | 0.428 | $>10^4$ | $>10^2$ | 0.121 | 0.348 |
| | RWExpSign | 0.162 | 0.139 | 0.172 | 0.146 | 0.492 | 0.442 | 0.524 | 0.468 | $>10^{10}$ | $>10^7$ | 0.169 | 0.243 |
| | RWRelu | 0.131 | 0.127 | 0.137 | **0.135** | 0.415 | 0.401 | 0.442 | 0.426 | $>10^7$ | $>10^5$ | 0.131 | 0.340 |
| | RWSign | 0.162 | 0.143 | 0.172 | 0.147 | 0.492 | 0.448 | 0.524 | 0.468 | $>10^4$ | $>10^2$ | 0.185 | 0.244 |
| | RWStumps | 0.139 | 0.139 | 0.149 | 0.146 | 0.468 | 0.470 | 0.487 | 0.483 | 46.8 | 1.0 | 0.140 | 0.257 |

On 6 of the 7 classification datasets (Table 10), one of the RKHS weighting instantiations has the lowest test error. However, Table 16 in the appendix shows that the difference with random kitchen sinks is meaningful on only one of those datasets (phishing).[13] Likewise, RKS has the best performance on one dataset (skin). On the regression datasets (Table 11), an RKHS weighting reaches the lowest mean squared error on 5 of the 6 datasets, but again with sometimes minimal margins. In fact, only one of those values (conductivity) is meaningfully better than what the random kitchen sinks achieve. Combined, these two experiments suggest that RKHS weightings have at most a slight edge over random kitchen sinks.

---

[13]We define a difference as meaningful or significant if the lower value plus one standard deviation is smaller than the larger value minus one standard deviation.

Table 11: Regression performance comparison of RKHS weightings (RW) to random kitchen sinks (RKS). For each combination of dataset and instantiation, an RKHS weighting was learned by cross-validation and using Algorithm 4 (the least squares fit) in order to select the best model hyperparameters. The sampling size $T$ was 500, and each value in the table is the average of 10 independently seeded runs. Th. 6.2 is the 95% generalization gap (the difference between Test MSE and Train MSE) guaranteed by Theorem 6.2. Th. 6.5 is the 95% generalization gap guaranteed by Theorem 6.5.

| Dataset | Instantiation | Train MSE RKS | Train MSE RW | Test MSE RKS | Test MSE RW | Th. 6.2 RW | Th. 6.5 RW | Train time (s) RKS | Train time (s) RW |
|---|---|---|---|---|---|---|---|---|---|
| abalone | RWExpRelu | 0.391 | 0.396 | 0.427 | **0.426** | $>10^3$ | 14.4 | 0.053 | 0.120 |
| | RWExpSign | 0.394 | 0.413 | 0.441 | 0.433 | $>10^6$ | $>10^3$ | 0.073 | 0.085 |
| | RWRelu | 0.391 | 0.398 | 0.427 | 0.430 | $>10^5$ | $>10^3$ | 0.052 | 0.118 |
| | RWSign | 0.394 | 0.406 | 0.441 | 0.431 | $>10^3$ | 22.8 | 0.066 | 0.088 |
| | RWStumps | 0.416 | 0.421 | 0.460 | 0.441 | $>10^3$ | 68.9 | 0.066 | 0.089 |
| california housing | RWExpRelu | 0.249 | 0.239 | 0.269 | 0.259 | $>10^5$ | $>10^3$ | 0.134 | 0.372 |
| | RWExpSign | 0.317 | 0.281 | 0.337 | 0.298 | $>10^9$ | $>10^7$ | 0.182 | 0.265 |
| | RWRelu | 0.249 | 0.243 | 0.269 | 0.261 | $>10^8$ | $>10^6$ | 0.135 | 0.363 |
| | RWSign | 0.317 | 0.280 | 0.337 | 0.296 | $>10^4$ | $>10^2$ | 0.179 | 0.267 |
| | RWStumps | 0.248 | 0.234 | 0.266 | **0.252** | $>10^2$ | 10.8 | 0.141 | 0.255 |
| concrete | RWExpRelu | 0.068 | 0.065 | 0.144 | 0.155 | $>10^4$ | 51.7 | 0.033 | 0.087 |
| | RWExpSign | 0.126 | 0.113 | 0.255 | 0.210 | $>10^{10}$ | $>10^7$ | 0.037 | 0.051 |
| | RWRelu | 0.068 | 0.087 | 0.144 | 0.153 | $>10^6$ | $>10^4$ | 0.034 | 0.082 |
| | RWSign | 0.126 | 0.083 | 0.255 | 0.189 | $>10^5$ | $>10^3$ | 0.039 | 0.057 |
| | RWStumps | 0.068 | 0.064 | **0.090** | 0.098 | $>10^3$ | 60.2 | 0.024 | 0.058 |
| conductivity | RWExpRelu | 0.176 | 0.167 | 0.193 | 0.185 | $>10^4$ | 6.01 | 0.190 | 0.462 |
| | RWExpSign | 0.207 | 0.217 | 0.224 | 0.222 | $>10^{11}$ | $>10^7$ | 0.172 | 0.207 |
| | RWRelu | 0.176 | 0.165 | 0.193 | **0.183** | $>10^7$ | $>10^4$ | 0.193 | 0.460 |
| | RWSign | 0.207 | 0.268 | 0.224 | 0.270 | $>10^4$ | $>10^2$ | 0.174 | 0.221 |
| | RWStumps | 0.187 | 0.182 | 0.198 | 0.193 | $>10^3$ | 16.9 | 0.172 | 0.309 |
| diabetes | RWExpRelu | 0.375 | 0.355 | 0.529 | 0.544 | 64.1 | 0.297 | 0.028 | 0.057 |
| | RWExpSign | 0.391 | 0.452 | 0.514 | 0.530 | $>10^2$ | 2.19 | 0.029 | 0.044 |
| | RWRelu | 0.375 | 0.382 | 0.529 | 0.529 | $>10^4$ | 26.1 | 0.027 | 0.055 |
| | RWSign | 0.391 | 0.453 | 0.514 | 0.514 | 10.5 | 0.331 | 0.030 | 0.045 |
| | RWStumps | 0.375 | 0.427 | 0.530 | **0.510** | $>10^2$ | 3.36 | 0.016 | 0.033 |
| wine | RWExpRelu | 0.032 | 0.032 | 0.113 | 0.101 | 52.2 | 0.305 | 0.028 | 0.038 |
| | RWExpSign | 0.009 | 0.026 | 0.091 | **0.088** | $>10^6$ | $>10^3$ | 0.027 | 0.038 |
| | RWRelu | 0.032 | 0.039 | 0.113 | 0.147 | $>10^4$ | 45.0 | 0.027 | 0.049 |
| | RWSign | 0.009 | 0.028 | 0.091 | 0.089 | $>10^5$ | $>10^2$ | 0.028 | 0.041 |
| | RWStumps | 0.042 | 0.066 | 0.108 | 0.136 | $>10^2$ | 5.46 | 0.012 | 0.031 |

We can also look at the performance of RKHS weightings and random kitchen sinks when using the same instantiation. In Table 10 we see that RKHS weightings have lower test error on 21 combinations of dataset and instantiation (8 are significant), as opposed to 8 for random kitchen sinks (3 are significant). In Table 11, RKHS weightings have lower test MSE on 19 combinations of dataset and instantiation (11 are significant), as opposed to 9 for random kitchen sinks (5 are significant). This is a slightly more convincing argument for RKHS weightings having an edge, on average, over random kitchen sinks.

Another aspect we can analyse in Tables 10 and 11 is the performance with regard to the instantiation. A throughline of both tables is that the ReLU appears to be a much better feature function than the sign. To point out only a single example, the test error on **adults** (Table 10) is 0.155 for random kitchen sinks and 0.152 for RKHS weightings when using the ReLU as the base predictor, but climbs to 0.198 (or even 0.220) when using the sign. This behavior is repeated, to various amplitudes, throughout both tables. The only major exception is the dataset wine, where the sign function leads to lower test MSE.

Yet another interesting point of comparison is between the bounded and unbounded versions of the same RKHS weighting instantiations: RWSign against RWExpSign, and RWRelu against RWExpRelu. RWExp-Sign outperforms RWSign on 4 of 7 datasets in Table 10 (2 are significant), and 2 of 6 datasets in Table 11 (1 is significant). RWSign instead outperforms RWExpSign on 1 of 7 datasets in Table 10 (significant), and 4 of 6 datasets in Table 11 (2 are significant). Similarly, RWExpRelu outperforms RWRelu on 2 of 7 datasets in Table 10 (1 is significant), and 3 of 6 datasets in Table 11 (2 are significant), while RWRelu outperforms RWExpRelu on 3 of 7 datasets in Table 10 (but the differences are not significant), and 3 of 6 datasets in Table 11 (1 is significant). Little conclusion can be reached from these observations.

As for the generalization bounds, it is clear that the Theorem 6.2 bound, based on the RKHS norm of the weight function, is always much larger than the Theorem 6.5 bound, which depends instead on the $L^2(p)$-norm of the weight function. This indicates that the latter bound is probably much closer to the true behavior of the model than the first. This means that the RKHS norm of the weight function can be quite large, as what really matters is its $L^2(p)$-norm. That said, the generalization gap is much smaller even than the Theorem 6.5 bound. Therefore, Theorem 6.5 does not seem to closely capture the behavior of the model either, at least in these experiments.

Finally, the training times for RKHS weightings are always a little slower than that of random kitchen sinks. However, both algorithms have virtually identical algorithmic complexities, so the differences are mostly due to implementation. Indeed, we used high level Python objects to represent the weight functions of RKHS weightings, which introduces overhead that is absent from our more straightforward implementation of the random kitchen sinks algorithm. A more optimized version of RKHS weightings would likely be as fast as random kitchen sinks.

### 7.4 Comparison to other models

From Tables 10 and 11, we can extract for each dataset the instantiations which lead to the best prediction performance. In Tables 12 and Tables 13, we compare the performance of RKHS weightings and random kitchen sinks, using those instantiations, to AdaBoost and SVM, two meaningful state of the art algorithms: AdaBoost is the algorithm to which Rahimi & Recht (2009) compare the random kitchen sinks, and SVM is a state of the art kernel method, ideal for a comparison with RKHS Weightings, which are implicitly a kernel method, and random kitchen sinks, which approximate a kernel method.

In Table 12, we compare all four algorithms on classification problems. AdaBoost has the best prediction accuracy on 3 of the 7 datasets; RKHS Weightings on 2; RKS on 1; SVM on 5. The random kitchen sinks and RKHS Weightings perform similarly on all datasets.

The story is much the same in Table 13, where we compare the four algorithms on regression problems. AdaBoost has the best performance on 1 dataset; RKHS Weightings on 3; RKS on 3;[14] SVM on 2. As in Table 12, differences between RKHS Weightings and random kitchen sinks are small.

---

[14]The perceptive reader might notice that we bolded both 0.358 and 0.338 in Table 13 (RKHS weightings with RWStumps, and RKS with RWStumps on diabetes) despite the seemingly large difference between the two values. However, we can see in Table 19 in the appendix that the standard deviation for the value 0.338 is high, making it statistically equivalent.

Table 12: Binary classification performance comparison of RKHS weightings to AdaBoost (**AB**), **SVM** and the random kitchen sinks (**RKS**) on various datasets. Instantiations were chosen based on their performance in Table 10. Algorithm 4 (the least squares fit of the coefficient) was used to learn RKHS weightings with $T = 2000$ random features. Train error and Test error are the misclassification rates on the training and test sets. Inference time is the computation time of the model on the training set and test sets combined. Every line (except SVM, which is deterministic) is the average of 10 independent runs.

| Dataset | Algorithm | Instantiation | Train error | Test error | Train time (s) | Inference time |
|---|---|---|---|---|---|---|
| adults | AdaBoost | | 0.141 | **0.140** | 15.225 | 3.663 |
| | RKHS Weighting | RWExpRelu | 0.137 | 0.148 | 3.967 | 4.473 |
| | | RWRelu | 0.138 | 0.147 | 3.984 | 4.357 |
| | | RWStumps | 0.146 | 0.145 | 2.163 | 1.824 |
| | RKS | RWRelu | 0.137 | 0.150 | 1.685 | 1.246 |
| | | RWStumps | 0.144 | 0.144 | **1.335** | **0.809** |
| | SVM | | 0.143 | 0.146 | 75.187 | 57.222 |
| cancer | AdaBoost | | 0.000 | **0.021** | 0.512 | 0.040 |
| | RKHS Weighting | RWExpSign | 0.011 | 0.045 | 0.259 | 0.032 |
| | | RWSign | 0.010 | 0.048 | 0.265 | 0.033 |
| | RKS | RWSign | 0.006 | 0.038 | 0.185 | 0.023 |
| | SVM | | 0.014 | **0.021** | **0.002** | **0.002** |
| marketing | AdaBoost | | 0.097 | 0.101 | 11.803 | 1.607 |
| | RKHS Weighting | RWExpRelu | 0.083 | **0.097** | 4.020 | 4.052 |
| | | RWRelu | 0.089 | **0.097** | 3.980 | 3.958 |
| | | RWStumps | 0.096 | 0.100 | 2.750 | 2.303 |
| | RKS | RWRelu | 0.085 | **0.097** | **1.700** | 1.113 |
| | | RWStumps | 0.096 | 0.100 | **1.692** | **1.069** |
| | SVM | | 0.077 | **0.097** | 28.927 | 17.778 |
| mnist17 | AdaBoost | | 0.000 | **0.004** | 90.048 | 9.455 |
| | RKHS Weighting | RWExpRelu | 0.002 | 0.008 | 2.066 | 1.686 |
| | | RWRelu | 0.003 | 0.010 | 3.470 | 1.642 |
| | | RWStumps | 0.006 | 0.009 | 1.085 | 0.711 |
| | RKS | RWRelu | 0.003 | 0.009 | 1.015 | 0.585 |
| | | RWStumps | 0.006 | 0.010 | **0.734** | **0.410** |
| | SVM | | 0.000 | 0.008 | 6.370 | 3.935 |
| phishing | AdaBoost | | 0.061 | 0.067 | 1.687 | **0.225** |
| | RKHS Weighting | RWExpRelu | 0.027 | 0.047 | 1.170 | 1.058 |
| | | RWRelu | 0.040 | 0.052 | 1.167 | 1.032 |
| | RKS | RWRelu | 0.025 | **0.045** | **0.519** | 0.276 |
| | SVM | | 0.024 | **0.043** | 0.709 | 0.824 |
| skin | AdaBoost | | 0.043 | 0.043 | 10.941 | **1.287** |
| | RKHS Weighting | RWExpRelu | 0.020 | 0.020 | 21.153 | 21.928 |
| | | RWRelu | 0.022 | 0.022 | 23.677 | 25.434 |
| | RKS | RWRelu | 0.012 | 0.013 | **8.424** | 5.763 |
| | SVM | | 0.000 | **0.000** | 533.843 | 53.736 |
| telescope | AdaBoost | | 0.144 | 0.157 | 10.140 | **0.234** |
| | RKHS Weighting | RWExpRelu | 0.119 | 0.133 | 1.790 | 1.726 |
| | | RWRelu | 0.120 | **0.131** | 1.768 | 1.689 |
| | RKS | RWRelu | 0.120 | 0.135 | **0.779** | 0.465 |
| | SVM | | 0.095 | **0.130** | 3.298 | 4.067 |

Table 13: Regression performance comparison of RKHS weightings to AdaBoost (**AB**), **SVM** and the random kitchen sinks (**RKS**) on various datasets. Instantiations were chosen based on their performance in Table 11. Algorithm 4 (the least squares fit of the coefficient) was used to learn RKHS weightings with $T = 2000$ random features. Inference time is the computation time of the model on the training set and test sets combined. Every line (except SVM, which is deterministic) is the average of 10 independent runs.

| Dataset | Algorithm | Instantiation | Train $R^2$ | Test $R^2$ | Train time (s) | Inference time |
|---|---|---|---|---|---|---|
| abalone | AdaBoost | | 0.440 | 0.444 | **0.067** | **0.003** |
| | RKHS Weighting | RWExpRelu | 0.604 | **0.583** | 0.601 | 0.420 |
| | | RWRelu | 0.603 | 0.576 | 0.574 | 0.403 |
| | | RWSign | 0.591 | 0.576 | 0.457 | 0.223 |
| | RKS | RWRelu | 0.618 | 0.581 | 0.314 | 0.100 |
| | | RWSign | 0.647 | 0.578 | 0.343 | 0.139 |
| | SVM | | 0.579 | 0.571 | 0.350 | 0.400 |
| concrete | AdaBoost | | 0.819 | 0.767 | 0.129 | **0.017** |
| | RKHS Weighting | RWExpRelu | 0.955 | 0.836 | 0.314 | 0.114 |
| | | RWRelu | 0.925 | 0.836 | 0.297 | 0.105 |
| | | RWStumps | 0.960 | **0.912** | 0.311 | 0.058 |
| | RKS | RWRelu | 0.959 | 0.850 | 0.193 | 0.024 |
| | | RWStumps | 0.945 | **0.910** | 0.148 | 0.025 |
| | SVM | | 0.957 | 0.851 | **0.025** | 0.020 |
| conductivity | AdaBoost | | 0.731 | 0.723 | 2.239 | **0.028** |
| | RKHS Weighting | RWExpRelu | 0.877 | 0.849 | 2.068 | 1.957 |
| | | RWRelu | 0.870 | 0.845 | 2.095 | 1.903 |
| | | RWStumps | 0.867 | 0.848 | 1.493 | 1.154 |
| | RKS | RWRelu | 0.878 | 0.849 | **0.896** | 0.517 |
| | | RWStumps | 0.876 | 0.852 | **0.882** | 0.533 |
| | SVM | | 0.903 | **0.871** | 13.384 | 11.280 |
| diabetes | AdaBoost | | 0.669 | 0.286 | 0.101 | 0.013 |
| | RKHS Weighting | RWExpRelu | 0.692 | 0.300 | 0.287 | 0.061 |
| | | RWRelu | 0.618 | 0.334 | 0.261 | 0.055 |
| | | RWStumps | 0.573 | **0.358** | 0.228 | 0.018 |
| | RKS | RWRelu | 0.628 | 0.335 | 0.181 | 0.017 |
| | | RWStumps | 0.626 | **0.338** | 0.140 | 0.016 |
| | SVM | | 0.609 | 0.343 | **0.008** | **0.010** |
| housing | AdaBoost | | 0.569 | 0.546 | **0.323** | **0.010** |
| | RKHS Weighting | RWExpRelu | 0.770 | 0.743 | 1.977 | 1.809 |
| | | RWRelu | 0.753 | 0.735 | 1.925 | 1.747 |
| | | RWStumps | 0.787 | 0.764 | 1.290 | 0.955 |
| | RKS | RWRelu | 0.779 | 0.741 | 0.846 | 0.437 |
| | | RWStumps | 0.788 | 0.759 | 0.725 | 0.401 |
| | SVM | | 0.815 | **0.775** | 9.278 | 8.181 |
| wine | AdaBoost | | 1.000 | **0.956** | 0.160 | 0.026 |
| | RKHS Weighting | RWExpSign | 0.979 | 0.893 | 0.259 | 0.019 |
| | | RWSign | 0.978 | 0.894 | 0.244 | 0.019 |
| | RKS | RWSign | 1.000 | 0.904 | 0.167 | 0.013 |
| | SVM | | 0.990 | 0.942 | **0.004** | **0.002** |

In short, the methods tested in these experiments are quite interchangeable in terms of prediction quality; each can beat the others on some datasets. Where there is meaningful difference, however, is in the execution time of the algorithms. The biggest offender is the SVM, which suffers greatly from its poor scaling with regard to the dataset size (e.g. 533 seconds on skin). The AdaBoost training time can also be high depending on the number of boosting rounds chosen by cross-validation (e.g. 90 seconds on mnist17). Algorithm 4 for learning RKHS Weightings and random kitchen sinks have consistent, moderate execution times, and can be made faster by lowering the number of random features $T$, at the cost of performance.

Note that we reproduced Tables 12 and 13 in the appendix with added standard deviations. These are Tables 18 and 19.

## 7.5   RKHS weightings with few random features

Tables 10 and 11 suggest that RKHS weightings have a performance advantage over random kitchen sinks. Tables 12 and 13 suggest that the two methods are quite interchangeable. The major difference between those two experiments, which can explain the apparent contradiction, is that the number of random features $T$ increased from 500 to 2000 between the two experiments. We surmise that RKHS weightings have better performance than random kitchen sinks when the number of random features is small. We test this hypothesis in this section.

In Figure 5, we test the classification performance of the models with the RWRelu instantiation while varying the number $T$ of random features. In Figure 6, we test the regression performance instead. We observe a clear pattern. RKHS weightings perform significantly better on average than random kitchen sinks when the number $T$ of random features is small. Another observation that we can make, especially in the regression experiments of Figure 6, is that the variance of the prediction quality of RKHS weightings often seems smaller than that of random kitchen sinks.

We conjecture that the explanation for these two phenomenons is the higher expressivity (and thus quality) of individual terms in an RKHS weighting compared to a random kitchen sinks. In random kitchen sinks and RKHS weightings, the model is a sum of the form:

$$f(x) = \sum_{t=1}^{T} a_t f_t(x). \tag{109}$$

For random kitchen sinks, we simply have $f_t(x) = \phi(w_t, x)$. For RKHS weighting, we instead have $f_t(x) := \mathbb{E}_{w \sim p} [\mathcal{K}(w_t, w)\phi(w, x)]$, with the expectation given for some instantiations in Table 3. All possible random features $w$ simultaneously contribute to this expectation (although to exceedingly small degree for most values of $w$). This likely makes each individual random feature more useful on average for RKHS weightings. However, as $T$ increases, the Monte Carlo approximation converges, and the random kitchen sinks model converges to the same random feature model expression $f(x) = \mathbb{E}_{w \sim p} [\alpha(w)\phi(w, x)]$ as the RKHS weighting, making the effect mostly disappear.

The better performance of RKHS weightings with few random features suggests that sparse RKHS weightings can outperform sparse random kitchen sinks. To test this, we ran a similar experiment as Figures 5 and 6, but using Algorithms 2 and 5 (the Lasso fits) to learn random kitchen sinks and RKHS weightings respectively. The results are found in Figures 9 and 10 of Appendix D. However, we do not observe the expected phenomenon; both models perform fairly similarly. The likely reason for the absence of a distinct advantage for either model in this experiment is that Algorithms 2 and 5 still sample a large number $T$ of features; the Lasso then chooses the most relevant ones. The advantage of RKHS weightings in Figures 5 and 6 comes (we conjecture) from the lower density of individually good base predictors being sampled in random kitchen sinks. This is counteracted by sampling many features, and then selecting the best ones.

It is therefore not clear how to leverage this advantage of RKHS weightings over random kitchen sinks in practice. After all, the performance of both methods will simply improve with a larger number of random features, at only a small cost in training and inference time.

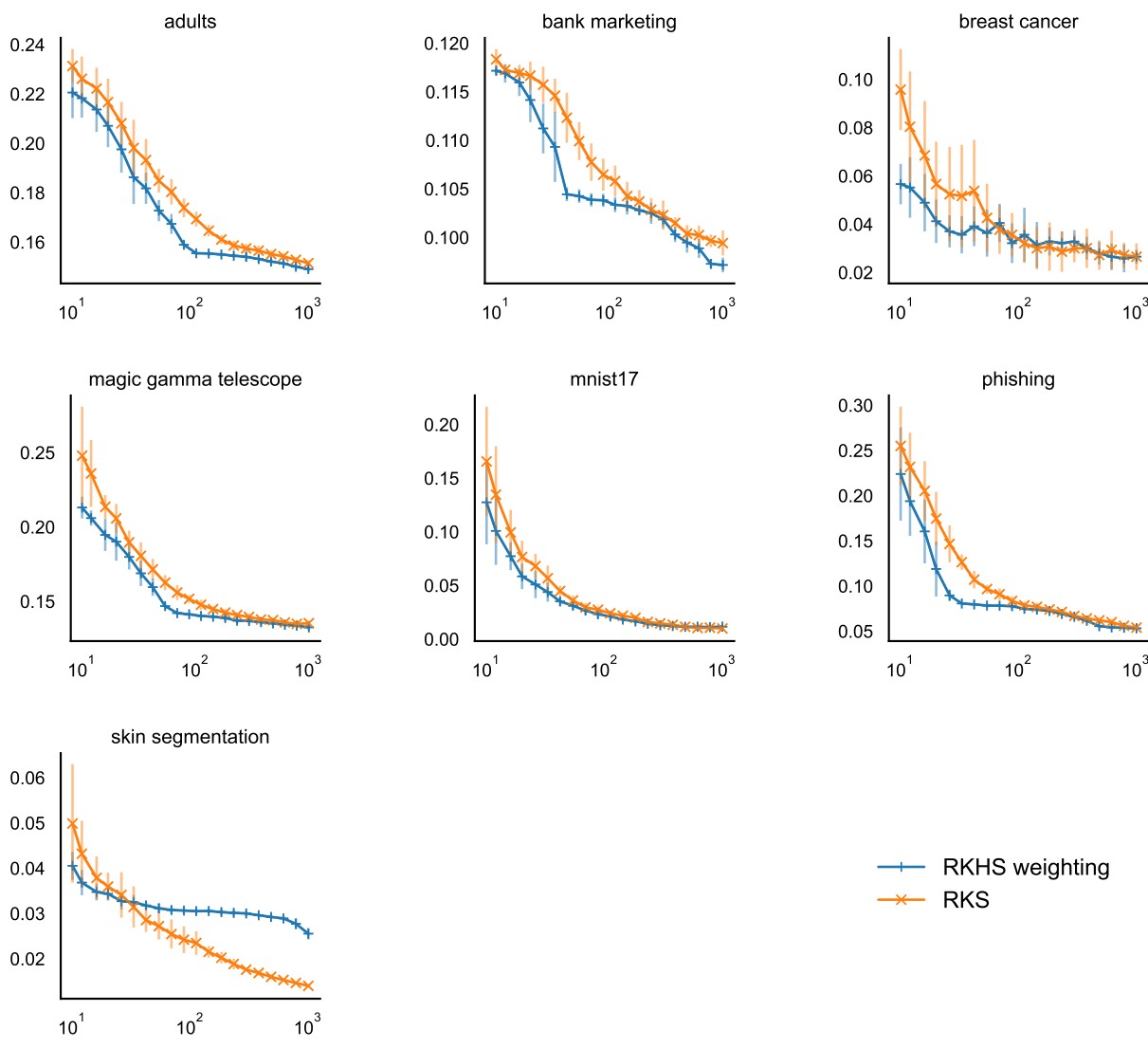

Figure 5: Comparison of the prediction performance of **RKHS weightings** (learned using Algorithm 4) and random kitchen sinks (**RKS**) with regard to the number of random features. The RKHS weightings instantiation is RWRelu. The random kitchen sinks used the same distribution (Gaussian) and base predictor (ReLU). Every point is the average of 10 independently seeded runs. The error bars represent one standard deviation.

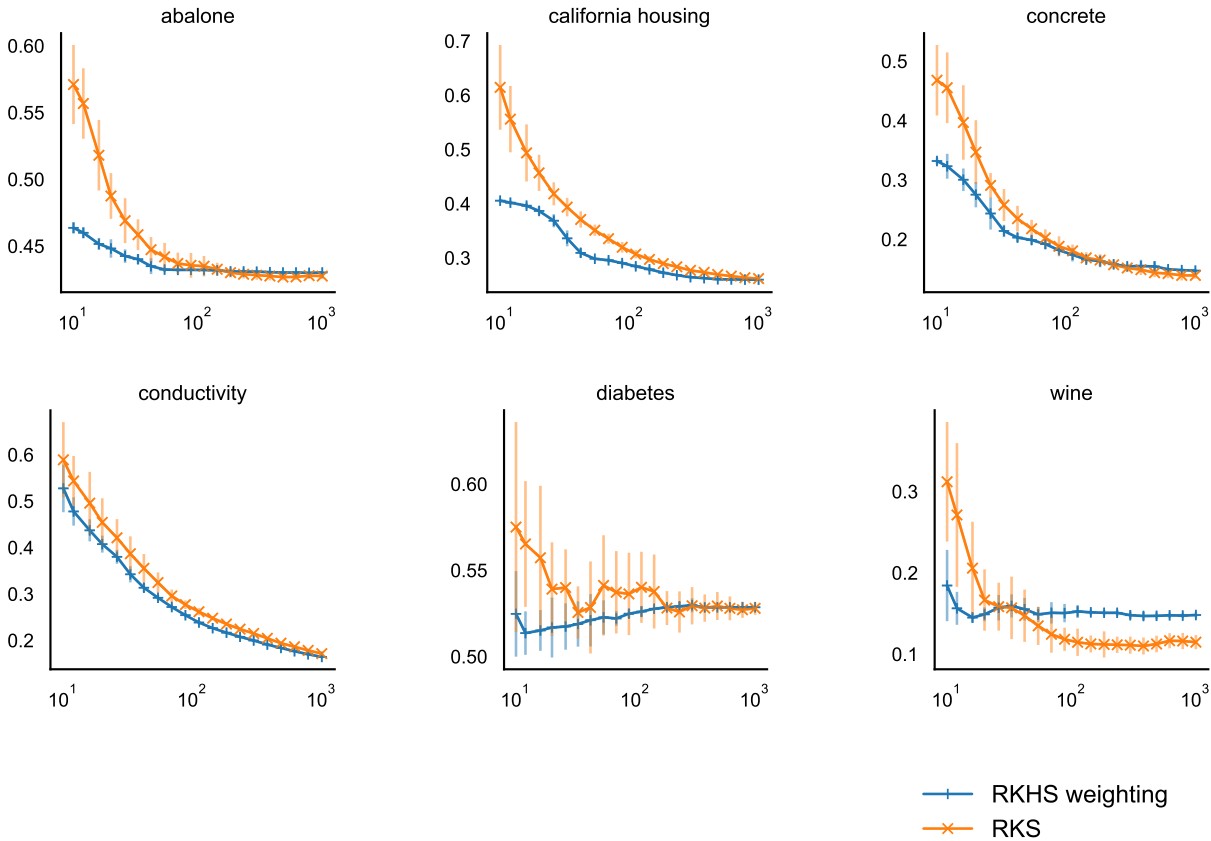

Figure 6: Comparison of the prediction performance of **RKHS weightings** (learned using Algorithm 4) and random kitchen sinks (**RKS**) with regard to the number of random features. The RKHS weightings instantiation is RWRelu. The random kitchen sinks used the same distribution (Gaussian) and base predictor (ReLU). Every point is the average of 5 independently seeded runs.

## 8 Future and limitations

This paper provides an initial exploration of the theory and practical aspects of RKHS weightings, especially in relation to random kitchen sinks (Rahimi & Recht, 2009), its closest relative in the space of random feature methods. Several directions merit further investigation. In particular, the following aspects appear to us to be the most important issues.

**Difficult integrals.** RKHS weightings are flexible in theory, allowing wide liberty in how to instantiate the model. However, the expectation $\mathbb{E}_{w \sim p}[\mathcal{K}(u, w)\phi(w, x)]$ must have an analytical form in order to calculate the output of the model via Equation (48). (We provided several examples in Table 2.) However, these integrals require hefty calculus to solve. The amount of work required to use a new instantiation is quite large. In fact, the analytical form for the expectation might not even exist for a given instantiation. In that case, the integrals can be approximated via Monte Carlo, at the cost of computation time and accuracy. We will explore this in a subsequent paper.

**Curse of dimensionality.** As we have addressed in Section 4.6, the model has a tendency to either vanish or explode (depending on the instantiation) when the dimensionality of the space $\mathcal{W}$ of random features increases. The model hyperparameters must therefore be chosen carefully, and we have presented a method for doing so. That method, however, requires knowing a meaningful upper bound on the constants $\theta$ and $\kappa$ of the model. This, again, requires calculating integrals. It would be useful to devise a simpler approach for selecting the model hyperparameters.

**Instantiating the model.** We saw in Tables 10 and 11 that different instantiations have different performances on different datasets. This means that the choice of instantiation is crucial for solving a given problem. How to choose or construct the best instantiation for the problem at hand is an important question to solve. Until this is better understood, trial and error is the only option, which is computationally costly. An avenue of future research is to look for combinations of problem types and instantiations (especially the choice of base predictor $\phi$ and kernel $\mathcal{K}$) that have interesting properties, such as high performance or interpretability. In fact, Dubé & Marchand (2025) have already laid some groundwork on constructing interpretable RKHS weightings.

**Unclear advantage over random kitchen sinks.** Our experiments uncovered an interesting phenomenon: RKHS weightings appear to perform better on average than random kitchen sinks when the number of random features is small. We were however unable at this point to find a practical application for this observation. Additionally, random kitchen sinks are simpler to use (requiring no calculus, and having one less moving part — the kernel — to validate), and provide similar performance in a slightly faster manner. There is therefore not yet a clear reason to use RKHS weightings over random kitchen sinks. While we think RKHS weightings are promising, their practical usefulness is not yet apparent. Further work on the points above is warranted to improve RKHS weightings, and potentially make them clearly advantageous over random kitchen sinks.

In summary, the model we have introduced in this paper appears capable in practice, similar to other random feature methods, and has multiple valid learning algorithms, but its full potential has not yet been fully realized. More work needs to be done in understanding how to choose or construct high performance instantiations, especially in the case of high-dimensional data.

## 9 Conclusion

In this paper, we examined a new random feature method, RKHS weightings, in the context of both binary classification and regression. We showed how the model can be learned using stochastic functional gradient descent or other algorithms, and proved convergence guarantees using the stability properties of the algorithm. We proved theoretical bounds on the generalization gap of the proposed class of predictors via Rademacher complexity theory. We ran experiments showing that the method compares well to the classical random kitchen sinks and other well-known machine learning algorithms.

This by no means constitutes a complete exploration of the applications and theoretical properties of RKHS weightings. Indeed, several challenges remain, such as understanding how to successfully instantiate the

model for a given problem, and how to leverage the flexibility of the model despite the requirement of solving difficult integrals. It is yet unclear what family of problems this model will excel at solving. We hope to use it in particular to build interpretable predictors, which is crucial for the safe and widespread adoption of artificial intelligence solutions.

**Acknowledgments**

This work is supported by the DEEL Project CRDPJ 537462-18 funded by the Natural Sciences and Engineering Research Council of Canada (NSERC) and the Consortium for Research and Innovation in Aerospace in Québec (CRIAQ), together with its industrial partners Thales Canada inc, Bell Textron Canada Limited, CAE inc and Bombardier inc.[15]

---

[15]https://deel.quebec

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

# A  Proofs

## A.1  Concentration inequalities

The first of three concentration inequalities we use in our proofs is the well-known Hoeffding's inequality. We reproduce it below from Theorem D.1 of Mohri et al. (2012).

**Theorem A.1.** *Let $X_1, \ldots, X_m$ be independent random variables with $X_i$ taking values in $[a_i, b_i]$ for all $i \in [1, m]$. Then for any $\epsilon > 0$, the following inequalities hold for $S_m = \sum_{i=1}^{m} X_i$:*

$$\Pr[S_m - \mathbb{E}[S_m] \geq \epsilon] \leq e^{-2\epsilon^2 / \sum_{i=1}^{m} (b_i - a_i)^2} \tag{110}$$

$$\Pr[S_m - \mathbb{E}[S_m] \leq -\epsilon] \leq e^{-2\epsilon^2 / \sum_{i=1}^{m} (b_i - a_i)^2} \tag{111}$$

The second inequality is McDiarmid's. We copy it here as is from Theorem D.3 of Mohri et al. (2012).

**Theorem A.2** (McDiarmid's inequality). *Let $X_1, \ldots, X_m \in \mathcal{X}^m$ be a set of $m \geq 1$ independent random variables and assume that there exist $c_1, \ldots, c_m > 0$ such that $f : \mathcal{X}^m \to \mathbb{R}$ satisfies the following conditions:*

$$|f(x_1, \ldots, x_i, \ldots, x_m) - f(x_1, \ldots, x_i', \ldots, x_m)| \leq c_i, \tag{112}$$

*for all $i \in [1, m]$ and any points $x_1, \ldots, x_m, x_i' \in \mathcal{X}$. Let $f(S)$ denote $f(X_1, \ldots, X_m)$, then, for all $\epsilon > 0$, the following inequalities hold:*

$$\Pr\left[f(S) - \mathbb{E}[f(S)] \geq \epsilon\right] \leq \exp\left(\frac{-2\epsilon^2}{\sum_{i=1}^{m} c_i^2}\right), \tag{113}$$

$$\Pr\left[f(S) - \mathbb{E}[f(S)] \leq -\epsilon\right] \leq \exp\left(\frac{-2\epsilon^2}{\sum_{i=1}^{m} c_i^2}\right). \tag{114}$$

The third concentration inequality, given in Exercise 6.1 of Boucheron et al. (2013), replaces the bounded differences condition of McDiarmid's inequality by the expected sum of squared differences. This condition can sometimes be tighter.

**Theorem A.3.** *Suppose that $Z = f(X_1, \ldots, X_n)$ is a real-valued function of $n$ independent random variables. (Note that the variables need not be identically distributed.) Denote $\mathbf{X} = (X_1, \ldots, X_n)$. For any $k \in \{1, \ldots, n\}$, denote $\mathbf{X}_k' := (X_1, \ldots, X_{k-1}, X_k', X_{k+1}, \ldots, X_n)$ the vector $\mathbf{X}$ with $X_k$ replaced by an identically distributed copy $X_k'$. Denote $Z_k' := f(\mathbf{X}_k')$. Additionally, denote $\mathbf{X}' := (X_1', \ldots, X_n')$ an identically distributed copy of the entire vector $\mathbf{X}$. Suppose that there exists some constant $v > 0$ such that:*

$$\mathbb{E}_{\mathbf{X}, \mathbf{X}'}\left[\sum_{k=1}^{n} (Z - Z_k')^2\right] \leq v. \tag{115}$$

*Then:*

$$\mathbb{P}_{\mathbf{X}}\left[Z > \mathbb{E}_{\mathbf{X}}[Z] + t\right] \leq e^{-t^2/(2v)}. \tag{116}$$

*Equivalently, for $\delta \in (0,1)$, we have:*

$$\mathbb{P}_{\mathbf{X}}\left[Z \leq \mathbb{E}_{\mathbf{X}}[Z] + \sqrt{2v \log \frac{1}{\delta}}\right] \geq 1 - \delta. \tag{117}$$

## A.2 Useful lemmas

The proof of Lemma A.4 uses the following simple lemma, which simply formalises the fact that if a result holds with high probability over the sampling of a random variable for any fixed value of a second variable, then it holds with high probability over the samplings of both variables. In our scenario, this means that we can prove a bound on either the sampling of $\mathcal{U} \sim p^T$ or $\mathcal{S} \sim \mathcal{D}^m$ (with the other fixed), and the bound can then trivially be extended to both samplings.

**Lemma A.1.** *Suppose we have two independent random variables $\mathcal{S}$ and $\mathcal{U}$, and a boolean valued function $Q(\mathcal{S},\mathcal{U})$ of those random variables. If we have $\mathbb{P}_{\mathcal{S}}[Q(\mathcal{S},\mathcal{U})|\mathcal{U}] \geq c$ for some $c \in [0,1]$ and for all $\mathcal{U}$, then $\mathbb{P}_{\mathcal{S},\mathcal{U}}[Q(\mathcal{S},\mathcal{U})] \geq c$ as well.*

*Proof.* We have:

$$\mathbb{P}_{\mathcal{S},\mathcal{U}}[Q(\mathcal{S},\mathcal{U})] = \mathbb{E}_{\mathcal{S},\mathcal{U}}\left[\mathbb{1}[Q(\mathcal{S},\mathcal{U}) \text{ is true}]\right]$$

$$= \mathbb{E}_{\mathcal{U}}\left[\mathbb{E}_{\mathcal{S}}\left[\mathbb{1}[Q(\mathcal{S},\mathcal{U}) \text{ is true}]\Big|\mathcal{U}\right]\right] \qquad \text{(Law of total expectations)}$$

$$= \mathbb{E}_{\mathcal{U}}\left[\mathbb{P}_{\mathcal{S}}\left[Q(\mathcal{S},\mathcal{U})\Big|\mathcal{U}\right]\right]$$

$$\geq \mathbb{E}_{\mathcal{U}}[c]$$

$$= c.$$

$\square$

## A.3 Proof of Equation (47)

We need to prove that

$$\Lambda\alpha(x) := \mathbb{E}_{w \sim p}[\alpha(w)\phi(w,x)]$$

$$= \mathbb{E}_{w \sim p}\left[\sum_{t=1}^{\infty} a_t \mathcal{K}(w_t, w)\phi(w,x)\right]$$

$$= \sum_{t=1}^{\infty} a_t \mathbb{E}_{w \sim p}[\mathcal{K}(w_t, w)\phi(w,x)]. \tag{118}$$

Assume A1. Let us write $\alpha_T := \sum_{t=1}^{T} a_t \mathcal{K}(w_t, \cdot)$. We will use the Dominated Convergence Theorem (see e.g. Evans (2018)). In order to prove that we can interchange the expectation and the infinite summation, we need to show that the partial series are dominated by an integrable function $g(w)$, i.e.

$$|\alpha_T(w)\phi(w,x)| \leq g(w). \tag{119}$$

Since the partial sums $\alpha_T := \sum_{t=1}^{T} a_t \mathcal{K}(w_t, \cdot)$ converge to $\alpha$ in the RKHS, there exists an $M \in \mathbb{R}$ such that $\|\alpha - \alpha_T\|_{\mathcal{H}} \leq M$ for all $T$. We assert that taking

$$g(w) := |\alpha(w)\phi(w,x)| + M\sqrt{\mathcal{K}(w,w)}|\phi(w,x)| \tag{120}$$

satisfies our needs. Indeed, we have Equation (119) if and only if:

$$|\alpha_T(w)\phi(w,x)| \leq |\alpha(w)\phi(w,x)| + M\sqrt{\mathcal{K}(w,w)}|\phi(w,x)|. \tag{121}$$

By the reverse triangle inequality, we have:

$$|\alpha_T(w)\phi(w,x)| - |\alpha(w)\phi(w,x)| \leq ||\alpha_T(w)\phi(w,x)| - |\alpha(w)\phi(w,x)||$$
$$\leq |\alpha_T(w)\phi(w,x) - \alpha(w)\phi(w,x)|.$$

Using the reproducing property of the RKHS and the Cauchy–Schwarz inequality, we obtain:

$$|\alpha_T(w)\phi(w,x) - \alpha(w)\phi(w,x)| = |\langle \alpha_T - \alpha, \mathcal{K}(w,\cdot)\rangle_{\mathcal{H}}\,\phi(w,x)|$$
$$\leq \|\alpha_T - \alpha\|_{\mathcal{H}}\|\mathcal{K}(w,\cdot)\|_{\mathcal{H}}|\phi(w,x)|$$
$$\leq M\sqrt{\mathcal{K}(w,w)}|\phi(w,x)|.$$

This shows that we indeed have Equation (121). Finally, the function on the right is integrable, since:

$$\mathop{\mathbb{E}}_{w\sim p}\left[|\alpha(w)\phi(w,x)| + M\sqrt{\mathcal{K}(w,w)}|\phi(w,x)|\right] = \mathop{\mathbb{E}}_{w\sim p}\left[|\langle\alpha,\mathcal{K}(w,\cdot)\rangle_{\mathcal{H}}\,\phi(w,x)| + M\sqrt{\mathcal{K}(w,w)}|\phi(w,x)|\right]$$
$$\leq \mathop{\mathbb{E}}_{w\sim p}\left[\|\alpha\|_{\mathcal{H}}\sqrt{\mathcal{K}(w,w)}\phi(w,x)| + M\sqrt{\mathcal{K}(w,w)}|\phi(w,x)|\right]$$
$$= (\|\alpha\|_{\mathcal{H}} + M)\mathop{\mathbb{E}}_{w\sim p}\left[\sqrt{\mathcal{K}(w,w)}|\phi(w,x)|\right]$$
$$\leq (\|\alpha\|_{\mathcal{H}} + M)\sqrt{\mathop{\mathbb{E}}_{w\sim p}[\mathcal{K}(w,w)\phi(w,x)^2]} \qquad \text{(Jensen)}$$
$$\leq (\|\alpha\|_{\mathcal{H}} + M)\kappa.$$

Therefore the dominated convergence theorem is applicable, giving us the result.

### A.4 Proof of Theorem 6.1

Starting from the definition of the sample Rademacher complexity of $\Lambda\mathcal{H}_B$, we have:

$$m\widehat{\mathcal{R}}_S(\Lambda\mathcal{H}_B) := \mathop{\mathbb{E}}_{\sigma\sim\{\pm 1\}^m}\left[\sup_{\Lambda\alpha\in\Lambda\mathcal{H}_B}\sum_{i=1}^m \sigma_i\Lambda\alpha(x_i)\right]$$
$$= \mathop{\mathbb{E}}_{\sigma}\left[\sup_{\alpha\in\mathcal{H}_B}\sum_{i=1}^m \sigma_i\langle\alpha,\psi(x_i)\rangle_{\mathcal{H}}\right] \qquad \text{(Equation (53))}$$
$$= \mathop{\mathbb{E}}_{\sigma}\left[\sup_{\alpha\in\mathcal{H}_B}\left\langle\alpha,\sum_{i=1}^m \sigma_i\psi(x_i)\right\rangle_{\mathcal{H}}\right]$$
$$\leq \mathop{\mathbb{E}}_{\sigma}\left[\sup_{\alpha\in\mathcal{H}_B}\|\alpha\|_{\mathcal{H}}\left\|\sum_{i=1}^m \sigma_i\psi(x_i)\right\|_{\mathcal{H}}\right] \qquad \text{(Cauchy–Schwarz)}$$
$$= B_{\mathcal{H}}\mathop{\mathbb{E}}_{\sigma}\left[\left\|\sum_{i=1}^m \sigma_i\psi(x_i)\right\|_{\mathcal{H}}\right].$$

We can then apply Jensen's inequality to get:

$$m\widehat{\mathcal{R}}_{\mathcal{S}}(\Lambda\mathcal{H}_B) \leq B_{\mathcal{H}} \underset{\sigma}{\mathbb{E}} \left[\left\|\sum_{i=1}^{m} \sigma_i \psi(x_i)\right\|_{\mathcal{H}}\right]$$

$$\leq B_{\mathcal{H}} \sqrt{\underset{\sigma}{\mathbb{E}} \left[\left\|\sum_{i=1}^{m} \sigma_i \psi(x_i)\right\|_{\mathcal{H}}^2\right]} \qquad \text{(Jensen)}$$

$$= B_{\mathcal{H}} \sqrt{\underset{\sigma}{\mathbb{E}} \left[\left\langle \sum_{i=1}^{m} \sigma_i \psi(x_i), \sum_{i=1}^{m} \sigma_i \psi(x_i) \right\rangle_{\mathcal{H}}^2\right]}$$

$$= B_{\mathcal{H}} \sqrt{\sum_{i=1}^{m} \sum_{j=1}^{m} \langle \psi(x_i), \psi(x_j)\rangle_{\mathcal{H}} \underset{\sigma}{\mathbb{E}}[\sigma_i \sigma_j]}$$

$$= B_{\mathcal{H}} \sqrt{\sum_{i=1}^{m} \|\psi(x_i)\|_{\mathcal{H}}^2} \qquad (\mathbb{E}_{\sigma}[\sigma_i \sigma_j] = \mathbb{1}[i=j])$$

$$\leq \sqrt{m}\theta B_{\mathcal{H}}.$$

We get the expected Rademacher complexity of $\Lambda\mathcal{H}_B$ by taking the expectation over the choice of sample. (Note that Lemma 26.10 of Shalev-Shwartz & Ben-David (2014) is a more general result, valid for any linear class in a Hilbert space.)

### A.5 Proof of Theorem 6.2

The proof of this theorem (adapted from the proof of Theorem 3.1 of Mohri et al. (2012)) consists in applying McDiarmid's inequality to the random variable:

$$f(\mathcal{S}) := \sup_{\alpha \in \mathcal{H}_B} [\mathcal{L}_{\mathcal{D}}(\Lambda\alpha) - \mathcal{L}_{\mathcal{S}}(\Lambda\alpha)]. \tag{122}$$

First, Lemma 26.2 of Shalev-Shwartz & Ben-David (2014) tells us that:

$$\underset{\mathcal{S}\sim\mathcal{D}^m}{\mathbb{E}}[f(\mathcal{S})] \leq 2\mathcal{R}_m(\ell \circ \Lambda\mathcal{H}_B), \tag{123}$$

where $\ell \circ \Lambda\mathcal{H}_B := \{(x,y) \mapsto \ell(\Lambda\alpha(x), y) \mid \alpha \in \mathcal{H}_B\}$. By Theorem 6.1 and the contraction lemma (Lemma 26.9 of Shalev-Shwartz & Ben-David (2014)), we get:

$$\mathcal{R}_m(\ell \circ \Lambda\mathcal{H}_B) = \underset{\mathcal{S}\sim\mathcal{D}^m}{\mathbb{E}} \left[\widehat{\mathcal{R}}_{\mathcal{S}}(\ell \circ \Lambda\mathcal{H}_B)\right]$$

$$\leq \rho \underset{\mathcal{S}\sim\mathcal{D}^m}{\mathbb{E}} \left[\widehat{\mathcal{R}}_{\mathcal{S}}(\Lambda\mathcal{H}_B)\right]$$

$$\leq \rho \underset{\mathcal{S}\sim\mathcal{D}^m}{\mathbb{E}} \left[\frac{\theta B_{\mathcal{H}}}{\sqrt{m}}\right]$$

$$\leq \frac{\rho\theta B_{\mathcal{H}}}{\sqrt{m}}. \tag{124}$$

Therefore, we have:

$$\underset{\mathcal{S}\sim\mathcal{D}^m}{\mathbb{E}}[f(\mathcal{S})] \leq \frac{2\rho\theta B_{\mathcal{H}}}{\sqrt{m}}. \tag{125}$$

To apply McDiarmid's inequality, we need to bound the change in the value of $f(\mathcal{S})$ resulting from exchanging a single example in the sample $\mathcal{S}$. Consider any $(x', y') \in \mathcal{X} \times \mathcal{Y}$, and denote $\mathcal{S}^{(i)} := (\mathcal{S}\setminus\{(x_i, y_i)\})\cup\{(x', y')\}$.

We have:

$$
\begin{aligned}
\left| f(\mathcal{S}) - f(\mathcal{S}^{(i)}) \right| &= \left| \sup_{\alpha \in \mathcal{H}_B} \left[ \mathcal{L}_\mathcal{D}(\Lambda\alpha) - \mathcal{L}_\mathcal{S}(\Lambda\alpha) \right] - \sup_{\alpha \in \mathcal{H}_B} \left[ \mathcal{L}_\mathcal{D}(\Lambda\alpha) - \mathcal{L}_{\mathcal{S}^{(i)}}(\Lambda\alpha) \right] \right| \\
&\leq \sup_{\alpha \in \mathcal{H}_B} \left[ \left| \mathcal{L}_\mathcal{D}(\Lambda\alpha) - \mathcal{L}_\mathcal{S}(\Lambda\alpha) - \mathcal{L}_\mathcal{D}(\Lambda\alpha) + \mathcal{L}_{\mathcal{S}^{(i)}}(\Lambda\alpha) \right| \right] \\
&= \sup_{\alpha \in \mathcal{H}_B} \left[ \left| \mathcal{L}_\mathcal{S}(\Lambda\alpha) - \mathcal{L}_{\mathcal{S}^{(i)}}(\Lambda\alpha) \right| \right] \\
&= \frac{1}{m} \sup_{\alpha \in \mathcal{H}_B} \left[ \left| \sum_{i=1}^m \ell(\Lambda\alpha(x_i), y) - \sum_{i=1}^m \ell(\Lambda\alpha(x_i), y) + \ell(\Lambda\alpha(x_i), y_i) - \ell(\Lambda\alpha(x'), y') \right| \right] \\
&= \frac{1}{m} \sup_{\alpha \in \mathcal{H}_B} \left[ \left| \ell(\Lambda\alpha(x_i), y_i) - \ell(\Lambda\alpha(x'), y') \right| \right] \\
&\leq \frac{\rho}{m} \sup_{\alpha \in \mathcal{H}_B} \left[ \left| \Lambda\alpha(x_i) - \Lambda\alpha(x') \right| \right] \\
&= \frac{2\rho\theta B_\mathcal{H}}{m}.
\end{aligned}
\tag{126}
$$

McDiarmid's inequality gives us:

$$
\mathbb{P}_{\mathcal{S} \sim \mathcal{D}^m} \left[ f(\mathcal{S}) - \mathbb{E}_{\mathcal{S} \sim \mathcal{D}^m}[f(\mathcal{S})] \geq \epsilon \right] \leq \exp\left( \frac{-2\epsilon^2 m}{(2\rho\theta B_\mathcal{H})^2} \right).
\tag{127}
$$

Setting:

$$
\delta := \exp\left( \frac{-2\epsilon^2 m}{(2\rho\theta B_\mathcal{H})^2} \right)
\tag{128}
$$

and rearranging, we get:

$$
\epsilon = 2\rho\theta B_\mathcal{H} \sqrt{\frac{\log\frac{1}{\delta}}{2m}}.
\tag{129}
$$

We obtain:

$$
\mathbb{P}_{\mathcal{S} \sim \mathcal{D}^m} \left[ f(\mathcal{S}) - \mathbb{E}_{\mathcal{S} \sim \mathcal{D}^m}[f(\mathcal{S})] \geq 2\rho\theta B_\mathcal{H} \sqrt{\frac{\log\frac{1}{\delta}}{2m}} \right] \leq \delta.
\tag{130}
$$

We can rewrite the previous probability as:

$$
\mathbb{P}_{\mathcal{S} \sim \mathcal{D}^m} \left[ f(\mathcal{S}) - \mathbb{E}_{\mathcal{S} \sim \mathcal{D}^m}[f(\mathcal{S})] \leq 2\rho\theta B_\mathcal{H} \sqrt{\frac{\log\frac{1}{\delta}}{2m}} \right] \geq 1 - \delta.
\tag{131}
$$

A final rearranging, and using Equation (125) to upper bound $\mathbb{E}_{\mathcal{S} \sim \mathcal{D}^m}[f(\mathcal{S})]$, we obtain:

$$
\mathbb{P}_{\mathcal{S} \sim \mathcal{D}^m} \left[ \sup_{\alpha \in \mathcal{H}_B} \left[ \mathcal{L}_\mathcal{D}(\Lambda\alpha) - \mathcal{L}_\mathcal{S}(\Lambda\alpha) \right] \leq \frac{2\rho\theta B_\mathcal{H}}{\sqrt{m}} \left( 1 + \sqrt{\frac{\log\frac{1}{\delta}}{2}} \right) \right] \geq 1 - \delta.
\tag{132}
$$

### A.6 Proof of Corollary 6.2.1

The proof is highly similar to that of Theorem 6.2. We only need to define:

$$
f(\mathcal{S}) := \sup_{\alpha \in \mathcal{H}_B} \left[ \mathcal{L}_\mathcal{D}(\Lambda\alpha) - \mathcal{L}_\mathcal{S}(\Lambda\alpha) \right]
\tag{133}
$$

rather than:

$$
f(\mathcal{S}) := \sup_{\alpha \in \mathcal{H}_B} \left[ \mathcal{L}_\mathcal{S}(\Lambda\alpha) - \mathcal{L}_\mathcal{D}(\Lambda\alpha) \right].
\tag{134}
$$

The proof of Lemma 26.2 of Shalev-Shwartz & Ben-David (2014) under this new definition will be virtually identical, and yield the same result.

## A.7 Proof of Theorem 6.4

We have:

$$
\begin{aligned}
m\widehat{\mathcal{R}}_{\mathcal{S}}(\Lambda\mathcal{F}_2) &:= \mathop{\mathbb{E}}_{\sigma \sim \{\pm 1\}^m}\left[\sup_{\alpha \in \mathcal{F}_2}\sum_{i=1}^m \sigma_i \Lambda\alpha(x_i)\right] \\
&= \mathop{\mathbb{E}}_{\sigma}\left[\sup_{\alpha \in \mathcal{F}_2}\sum_{i=1}^m \sigma_i \left\langle \alpha, \phi(\cdot, x_i)\right\rangle_{L^2(p)}\right] \\
&= \mathop{\mathbb{E}}_{\sigma}\left[\sup_{\alpha \in \mathcal{F}_2}\left\langle \alpha, \sum_{i=1}^m \sigma_i \phi(\cdot, x_i)\right\rangle_{L^2(p)}\right] \\
&\leq \mathop{\mathbb{E}}_{\sigma}\left[\sup_{\alpha \in \mathcal{F}_2}\|\alpha\|_{L^2(p)}\left\|\sum_{i=1}^m \sigma_i \phi(\cdot, x_i)\right\|_{L^2(p)}\right] &\text{(Cauchy--Schwarz)} \\
&= B_2 \mathop{\mathbb{E}}_{\sigma}\left[\left\|\sum_{i=1}^m \sigma_i \phi(\cdot, x_i)\right\|_{L^2(p)}\right].
\end{aligned}
$$

As we did in the proof of Theorem 6.1, we can use Jensen to greatly simplify the expression. We will obtain:

$$
\begin{aligned}
m\widehat{\mathcal{R}}_{\mathcal{S}}(\Lambda\mathcal{F}_2) &\leq B_2 \mathop{\mathbb{E}}_{\sigma}\left[\left\|\sum_{i=1}^m \sigma_i \phi(\cdot, x_i)\right\|_{L^2(p)}\right] \\
&\leq B_2 \sqrt{\sum_{i=1}^m \|\phi(\cdot, x_i)\|_{L^2(p)}^2} \\
&\leq B_2 \tau \sqrt{m}.
\end{aligned}
$$

This gives us the desired empirical Rademacher complexity. The Rademacher complexity is obtained by taking the expectation of $\mathcal{S}$, which does not affect the expression.

## A.8 Proof of Theorem 6.7

We first need to prove a simple lemma.

**Lemma A.2.** *Consider some independent identically distributed variables $(z_1, \ldots, z_m)$ taken from a Hilbert space. Then:*

$$
\mathbb{E}\left[\left\|\frac{1}{m}\sum_{i=1}^m z_i\right\|^2\right] \leq \mathbb{E}\left[\|z_1\|^2\right].
$$

*Proof.* We have:

$$
\mathbb{E}\left[\left\|\frac{1}{m}\sum_{i=1}^{m}z_i\right\|^2\right] = \frac{1}{m^2}\mathbb{E}\left[\left\langle\sum_{i=1}^{m}z_i, \sum_{j=1}^{m}z_j\right\rangle\right]
$$

$$
= \frac{1}{m^2}\sum_{i=1}^{m}\sum_{j=1}^{m}\mathbb{E}\left[\langle z_i, z_j\rangle\right] \qquad \text{(Linearity)}
$$

$$
\leq \frac{1}{m^2}\sum_{i=1}^{m}\sum_{j=1}^{m}\mathbb{E}\left[\|z_i\|\|z_j\|\right] \qquad \text{(Cauchy–Schwarz)}
$$

$$
= \frac{1}{m^2}\sum_{i=1}^{m}\left[\mathbb{E}\left[\|z_i\|^2\right] + \sum_{j\neq i}\mathbb{E}\left[\|z_i\|\right]\mathbb{E}\left[\|z_j\|\right]\right] \qquad \text{(Independence)}
$$

$$
= \frac{1}{m^2}\sum_{i=1}^{m}\left[\mathbb{E}\left[\|z_1\|^2\right] + (m-1)\left(\mathbb{E}\left[\|z_1\|\right]\right)^2\right] \qquad \text{(Identical distribution)}
$$

$$
\leq \frac{1}{m^2}\sum_{i=1}^{m}\left[\mathbb{E}\left[\|z_1\|^2\right] + (m-1)\mathbb{E}\left[\|z_1\|^2\right]\right] \qquad \text{(Jensen)}
$$

$$
= \mathbb{E}\left[\|z_1\|^2\right].
$$

$\square$

Now, consider any sample $\mathcal{S}$ and denote:

$$
A_\alpha(\mathcal{S}) := \underset{\alpha\in\mathcal{H}_B}{\mathrm{argmin}}\left(\mathcal{L}_\mathcal{S}(\Lambda\alpha) + \lambda_\mathcal{H}\|\alpha\|_\mathcal{H}^2\right), \tag{135}
$$

and:

$$
A(\mathcal{S}) := \Lambda A_\alpha(\mathcal{S}) = \underset{\Lambda\alpha\in\Lambda\mathcal{H}_B}{\mathrm{argmin}}\left(\mathcal{L}_\mathcal{S}(\Lambda\alpha) + \lambda_\mathcal{H}\|\alpha\|_\mathcal{H}^2\right), \tag{136}
$$

the regularized empirical risk minimizer. Then, write:

$$
\begin{aligned}
\mathbb{E}\left[\mathcal{L}_\mathcal{D}(\Lambda\bar{\alpha})\right] = \ &\mathbb{E}\left[\mathcal{L}_\mathcal{D}(\Lambda\bar{\alpha}) - \mathcal{L}_\mathcal{S}(\Lambda\bar{\alpha})\right] \\
&+ \mathbb{E}\left[\mathcal{L}_\mathcal{S}(\Lambda\bar{\alpha}) - \mathcal{L}_\mathcal{S}(A(\mathcal{S}))\right] \\
&+ \mathbb{E}\left[\mathcal{L}_\mathcal{S}(A(\mathcal{S})) - \mathcal{L}_\mathcal{D}(A(\mathcal{S}))\right] \\
&+ \mathbb{E}\left[\mathcal{L}_\mathcal{D}(A(\mathcal{S}))\right].
\end{aligned} \tag{137}
$$

(Every expectation in this proof is over the sample and all sampled parameters and batches, unless specified otherwise.) We can bound each of these terms separately. First, Lemma 26.2 of Shalev-Shwartz & Ben-David (2014) applied to Equation (124) gives us:

$$
\underset{\mathcal{S}\sim\mathcal{D}^m}{\mathbb{E}}\left[\sup_{\Lambda\alpha\in\Lambda\mathcal{H}_B}\left(\mathcal{L}_\mathcal{D}(\Lambda\alpha) - \mathcal{L}_\mathcal{S}(\Lambda\alpha)\right)\right] \leq \frac{2\rho\theta B_\mathcal{H}}{\sqrt{m}}. \tag{138}
$$

Being a bound on the supremum, it is also valid for $\Lambda\bar{\alpha}$, independently of the sampled parameters and batches, which gives us:

$$
\mathbb{E}\left[\mathcal{L}_\mathcal{D}(\Lambda\bar{\alpha}) - \mathcal{L}_\mathcal{S}(\Lambda\bar{\alpha})\right] \leq \frac{2\rho\theta B_\mathcal{H}}{\sqrt{m}}. \tag{139}
$$

Next, the proof of Corollary 13.6 of Shalev-Shwartz & Ben-David (2014) can be modified by swapping the two terms on the left-hand side of equation 13.11, which allows us to get:

$$
\mathbb{E}\left[\mathcal{L}_\mathcal{S}(A(\mathcal{S})) - \mathcal{L}_\mathcal{D}(A(\mathcal{S}))\right] = \underset{\mathcal{S}\sim\mathcal{D}^m}{\mathbb{E}}\left[\mathcal{L}_\mathcal{S}(A(\mathcal{S})) - \mathcal{L}_\mathcal{D}(A(\mathcal{S}))\right] \leq \frac{4\rho^2}{\lambda_\mathcal{H}m}. \tag{140}
$$

Also, Corollary 13.8 of Shalev-Shwartz & Ben-David (2014) directly gives us:

$$\mathbb{E}\left[\mathcal{L}_{\mathcal{D}}(A(\mathcal{S}))\right] = \underset{\mathcal{S} \sim \mathcal{D}^m}{\mathbb{E}}\left[\mathcal{L}_{\mathcal{D}}(A(\mathcal{S}))\right] \leq \min_{\Lambda\alpha \in \Lambda\mathcal{H}_B} \mathcal{L}_{\mathcal{D}}(\Lambda\alpha) + \lambda_{\mathcal{H}} B_{\mathcal{H}}^2 + \frac{4\rho^2}{\lambda_{\mathcal{H}} m}. \tag{141}$$

Bounding the term $\mathbb{E}\left[\mathcal{L}_{\mathcal{S}}(\Lambda\bar{\alpha}) - \mathcal{L}_{\mathcal{S}}(A(\mathcal{S}))\right]$ requires more work. We seek to apply Theorem 14.11 of Shalev-Shwartz & Ben-David (2014) to the regularized empirical risk $\mathcal{L}_{\mathcal{S}}(\Lambda\alpha) + \lambda_{\mathcal{H}}\|\alpha\|_{\mathcal{H}}^2$, which is $\lambda_{\mathcal{H}}$-strongly convex (by convexity of $\ell$ and linearity in $\alpha$ of $\Lambda\alpha$). At iteration $t$ of Algorithm 3, the subgradient (unbiased gradient approximation) is given by Equation (71):

$$v_t := v(\alpha^{(t-1)}, w_t, \mathcal{B}_t) := \left(\frac{1}{b} \sum_{(x,y)\in\mathcal{B}_t} \ell'(\Lambda\alpha^{(t-1)}(x), y)\phi(w_t, x)\right)\mathcal{K}(w_t, \cdot) + 2\lambda_{\mathcal{H}}\alpha^{(t-1)}.$$

We need to upper bound $\mathbb{E}\left[\|v_t\|^2\right]$. First, write:

$$u_t := v_t - 2\lambda_{\mathcal{H}}\alpha^{(t-1)} = \frac{1}{b} \sum_{(x,y)\in\mathcal{B}_t} \ell'(\Lambda\alpha^{(t-1)}(x), y)\phi(w_t, x)\mathcal{K}(w_t, \cdot).$$

For the purpose of finding an upper bound, we can assume that the batch size is 1, using Lemma A.2. Writing $\mathcal{B}_t = \{(x_t, y_t)\}$, we have:

$$\begin{aligned}
\mathbb{E}\left[\|u_t\|^2\right] &= \mathbb{E}\left[\left\|\frac{1}{b}\sum_{(x,y)\in\mathcal{B}_t}\ell'(\Lambda\alpha^{(t-1)}(x), y)\phi(w_t, x)\mathcal{K}(w_t, \cdot)\right\|_{\mathcal{H}}^2\right] \\
&= \mathbb{E}\left[\left\|\ell'(\Lambda\alpha^{(t-1)}(x_t), y_t)\phi(w_t, x_t)\mathcal{K}(w_t, \cdot)\right\|_{\mathcal{H}}^2\right] \\
&\leq \mathbb{E}\left[\rho^2\|\mathcal{K}(w_t, \cdot)\phi(w_t, x_t)\|_{\mathcal{H}}^2\right] \\
&= \rho^2\,\mathbb{E}\left[\mathcal{K}(w_t, w_t)\phi(w_t, x_t)^2\right] \\
&\leq \rho^2\kappa^2. \tag{142}
\end{aligned}$$

Next, because of the projection step in Algorithm 3, we have $\alpha^{(t-1)} \in \Lambda\mathcal{H}_B$ for every $t$, i.e. $\|\alpha^{(t-1)}\|_{\mathcal{H}} \leq B$. This allows us to write:

$$\begin{aligned}
\mathbb{E}\left[\|v_t\|^2\right] &= \mathbb{E}\left[\left\|u_t + 2\lambda_{\mathcal{H}}\alpha^{(t-1)}\right\|_{\mathcal{H}}^2\right] \\
&\leq \mathbb{E}\left[\left(\|u_t\|_{\mathcal{H}} + 2\lambda_{\mathcal{H}}\left\|\alpha^{(t-1)}\right\|_{\mathcal{H}}\right)^2\right] && \text{(Triangle inequality)} \\
&= \mathbb{E}\left[\|u_t\|_{\mathcal{H}}^2 + 4\lambda_{\mathcal{H}}\|u_t\|_{\mathcal{H}}\left\|\alpha^{(t-1)}\right\|_{\mathcal{H}} + 4\lambda_{\mathcal{H}}^2\left\|\alpha^{(t-1)}\right\|_{\mathcal{H}}^2\right] \\
&\leq \mathbb{E}\left[\|u_t\|_{\mathcal{H}}^2\right] + 4\lambda_{\mathcal{H}}B\,\mathbb{E}\left[\|u_t\|_{\mathcal{H}}\right] + 4\lambda_{\mathcal{H}}^2 B_{\mathcal{H}}^2 \\
&\leq \rho^2\kappa^2 + 4\lambda_{\mathcal{H}}B\sqrt{\mathbb{E}\left[\|u_t\|_{\mathcal{H}}^2\right]} + 4\lambda_{\mathcal{H}}^2 B_{\mathcal{H}}^2 && \text{(Jensen)} \\
&\leq \rho^2\kappa^2 + 4\lambda_{\mathcal{H}}B_{\mathcal{H}}\rho\kappa + 4\lambda_{\mathcal{H}}^2 B_{\mathcal{H}}^2 \\
&= (\rho\kappa + 2\lambda_{\mathcal{H}}B_{\mathcal{H}})^2.
\end{aligned}$$

Applying Theorem 14.11 of Shalev-Shwartz & Ben-David (2014), we get:

$$\mathbb{E}\left[\mathcal{L}_{\mathcal{S}}(\Lambda\bar{\alpha}) - \mathcal{L}_{\mathcal{S}}(A(\mathcal{S}))\right] \leq \mathbb{E}\left[\lambda_{\mathcal{H}}\|\bar{\alpha}\|_{\mathcal{H}}^2 - \lambda_{\mathcal{H}}\|A_{\alpha}(\mathcal{S})\|_{\mathcal{H}}^2\right] + \frac{(\rho\kappa + 2\lambda_{\mathcal{H}}B_{\mathcal{H}})^2}{2\lambda_{\mathcal{H}}T}(1 + \log(T)). \tag{143}$$

We can simplify this expression by noticing that $\lambda_{\mathcal{H}}\|\bar{\alpha}\|_{\mathcal{H}}^2 - \lambda_{\mathcal{H}}\|A_\alpha(\mathcal{S})\|_{\mathcal{H}}^2$ is at most $\lambda_{\mathcal{H}}B_{\mathcal{H}}^2$:

$$\mathbb{E}\left[\mathcal{L}_{\mathcal{S}}(\Lambda\bar{\alpha}) - \mathcal{L}_{\mathcal{S}}(A(\mathcal{S}))\right] \leq \lambda_{\mathcal{H}}B_{\mathcal{H}}^2 + \frac{(\rho\kappa + 2\lambda_{\mathcal{H}}B_{\mathcal{H}})^2}{2\lambda_{\mathcal{H}}T}(1 + \log(T)). \tag{144}$$

Inserting Equations (139), (140), (141) and (144) into Equation (137), we obtain the first part of the theorem:

$$\mathbb{E}\left[\mathcal{L}_{\mathcal{D}}(\Lambda\bar{\alpha})\right] \leq \min_{\Lambda\alpha\in\Lambda\mathcal{H}_B}\mathcal{L}_{\mathcal{D}}(\Lambda\alpha) + \frac{2\rho\theta B_{\mathcal{H}}}{\sqrt{m}} + \lambda_{\mathcal{H}}B_{\mathcal{H}}^2 + \frac{8\rho^2}{\lambda_{\mathcal{H}}m} + \frac{(\rho\kappa + 2\lambda_{\mathcal{H}}B_{\mathcal{H}})^2}{2\lambda_{\mathcal{H}}T}(1 + \log(T)). \tag{145}$$

Taking $\lambda_{\mathcal{H}} = \sqrt{\frac{8\rho^2}{B_{\mathcal{H}}^2 m}}$, we get the second part of the theorem:

$$
\begin{aligned}
\mathbb{E}\left[\mathcal{L}_{\mathcal{D}}(\Lambda\bar{\alpha})\right] &\leq \min_{\Lambda\alpha\in\Lambda\mathcal{H}_B}\mathcal{L}_{\mathcal{D}}(\Lambda\alpha) + \frac{2\rho\theta B_{\mathcal{H}}}{\sqrt{m}} + \lambda_{\mathcal{H}}B_{\mathcal{H}}^2 + \frac{8\rho^2}{\lambda_{\mathcal{H}}m} + \frac{(\rho\kappa + 2\lambda_{\mathcal{H}}B_{\mathcal{H}})^2}{2\lambda_{\mathcal{H}}T}(1 + \log(T)) \\
&= \min_{\Lambda\alpha\in\Lambda\mathcal{H}_B}\mathcal{L}_{\mathcal{D}}(\Lambda\alpha) + \frac{2\rho\theta B_{\mathcal{H}}}{\sqrt{m}} + 2\sqrt{\frac{8\rho^2 B_{\mathcal{H}}^2}{m}} + \sqrt{\frac{B_{\mathcal{H}}^2 m}{32\rho^2 T^2}}\left(\rho\kappa + 2\sqrt{\frac{8\rho^2}{m}}\right)^2(1 + \log(T)) \\
&= \min_{\Lambda\alpha\in\Lambda\mathcal{H}_B}\mathcal{L}_{\mathcal{D}}(\Lambda\alpha) + \frac{2\rho\theta B_{\mathcal{H}}}{\sqrt{m}} + \sqrt{\frac{32\rho^2 B_{\mathcal{H}}^2}{m}} + \sqrt{\frac{\rho^2 B_{\mathcal{H}}^2}{m}}\left(\frac{m}{\sqrt{32}T}\left(\kappa + \sqrt{\frac{32}{m}}\right)^2(1 + \log(T))\right) \\
&= \min_{\Lambda\alpha\in\Lambda\mathcal{H}_B}\mathcal{L}_{\mathcal{D}}(\Lambda\alpha) + \frac{\rho B_{\mathcal{H}}}{\sqrt{m}}\left(\sqrt{32} + 2\theta + \frac{m}{\sqrt{32}T}\left(\kappa + \sqrt{\frac{32}{m}}\right)^2(1 + \log(T))\right).
\end{aligned}
$$

## A.9 Proof of Theorem 6.8

Theorem 6.8 depends on the $\ell_1$-norm of the vector of coefficients of the optimal weight function. Instead of considering weight functions of RKHS norm at most $B_{\mathcal{H}}$, as in Theorem 6.2, Theorem 6.8 looks at weight functions with this $\ell_1$-norm at most $B_1$. To justify this replacement, we prove the following lemma, which shows that a version of the representer theorem still applies in this case; the optimal weight function still belongs to $\mathcal{H}_{\mathcal{S}}$.

**Lemma A.3.** *Assume A1 and A2. Define $h_i := \psi(x_i)$ for $i \in \{1, \ldots, m\}$, and suppose that $\{h_{m+1}, h_{m+2}, \ldots\}$ is some orthonormal basis for $\mathcal{H}_{\mathcal{S}}^{\perp}$, so that we can write $\mathcal{H} = \mathcal{H}_{\mathcal{S}} + \mathcal{H}_{\mathcal{S}}^{\perp}$. For some $\alpha = \sum_{i=1}^{\infty} a_i h_i \in \mathcal{H}$, define $\mathbf{a} := (a_1, a_2, \ldots)$ the (infinite) vector of coefficients, and $[\![\alpha]\!] := \|\mathbf{a}\|_1 = \sum_{i=1}^{\infty}|a_i|$.[16] For any $B_1 > 0$, there exists $\alpha_{\mathcal{S}} \in \mathcal{H}_{\mathcal{S}}$ with $[\![\alpha_{\mathcal{S}}]\!] \leq B_1$ which minimizes the empirical risk:*

$$\mathcal{L}_{\mathcal{S}}(\Lambda\alpha_{\mathcal{S}}) = \min_{\alpha\in\mathcal{H}:[\![\alpha]\!]\leq B_1}\mathcal{L}_{\mathcal{S}}(\Lambda\alpha). \tag{146}$$

*Furthermore, for any $\alpha \in \mathcal{H}_{\mathcal{S}}$, we have $\|\alpha\|_{\mathcal{H}} \leq \theta[\![\alpha]\!] \leq \theta B_1$.*

*Proof.* The optimal weight function $\alpha$ can be written $\alpha = \alpha_{\mathcal{S}} + \alpha_{\mathcal{S}}^{\perp}$, with $\alpha_{\mathcal{S}} \in \mathcal{H}_{\mathcal{S}}$, and $\alpha_{\mathcal{S}}^{\perp} \in \mathcal{H}_{\mathcal{S}}^{\perp}$. Since $\psi(x_i) \in \mathcal{H}_{\mathcal{S}}$, we have $\langle \alpha_{\mathcal{S}}^{\perp}, \psi(x_i)\rangle_{\mathcal{H}} = 0$ for all $i$. This allows us to write:

$$
\begin{aligned}
\Lambda\alpha(x_i) &= \langle\alpha, \psi(x_i)\rangle_{\mathcal{H}} \\
&= \langle\alpha_{\mathcal{S}} + \alpha_{\mathcal{S}}^{\perp}, \psi(x_i)\rangle_{\mathcal{H}} \\
&= \langle\alpha_{\mathcal{S}}, \psi(x_i)\rangle_{\mathcal{H}} + \langle\alpha_{\mathcal{S}}^{\perp}, \psi(x_i)\rangle_{\mathcal{H}} \\
&= \langle\alpha_{\mathcal{S}}, \psi(x_i)\rangle_{\mathcal{H}} \\
&= \Lambda\alpha_{\mathcal{S}}(x_i).
\end{aligned}
$$

---

[16]If an $\alpha \in \mathcal{H}_{\mathcal{S}}$ has multiple writings, i.e. $\alpha = \sum_{i=1}^{m} a_i\psi(x_i) = \sum_{i=1}^{m} b_i\psi(x_i)$, we choose the one with the lowest $\ell_1$-norm of the vector of coefficients. Also, note that $[\![\cdot]\!]$ is not a norm, thus the altered notation.

In other words, $\alpha_{\mathcal{S}}^\perp$ does not contribute to the predictions on the training set $\mathcal{S}$. This means that $\mathcal{L}_{\mathcal{S}}(\Lambda\alpha) = \mathcal{L}_{\mathcal{S}}(\Lambda\alpha_{\mathcal{S}})$. Furthermore, any nonzero coefficient $a_i$, for $i > m$, adds to $[\![\alpha]\!]$ (without improving the empirical risk). Therefore, $[\![\alpha_{\mathcal{S}}]\!] \leq [\![\alpha]\!]$.

To upper bound $\|\alpha_{\mathcal{S}}\|_{\mathcal{H}}$, we can notice that $\langle \psi(x_i), \psi(x_j) \rangle_{\mathcal{H}} \leq \|\psi(x_i)\|_{\mathcal{H}} \|\psi(x_j)\|_{\mathcal{H}} \leq \theta^2$ (by the Cauchy–Schwarz inequality), which means that:

$$
\begin{aligned}
\|\alpha_{\mathcal{S}}\|_{\mathcal{H}}^2 &= \left\langle \sum_{i=1}^m a_i \psi(x_i), \sum_{i=1}^m a_i \psi(x_i) \right\rangle_{\mathcal{H}} \\
&= \sum_{i=1}^m \sum_{j=1}^m a_i a_j \langle \psi(x_i), \psi(x_j) \rangle_{\mathcal{H}} = \|\Lambda\alpha_{\mathcal{S}}\|_{\mathcal{H}_{\mathcal{X}}}^2 \\
&\leq \sum_{i=1}^m \sum_{j=1}^m |a_i a_j| |\langle \psi(x_i), \psi(x_j) \rangle_{\mathcal{H}}| \\
&\leq \theta^2 \sum_{i=1}^m \sum_{j=1}^m |a_i a_j| \\
&= \theta^2 \sum_{i=1}^m |a_i| \sum_{j=1}^m |a_j| \\
&\leq \theta^2 \|a\|_1^2 \\
&\leq \theta^2 B_1^2.
\end{aligned}
$$

$\square$

The second lemma that we require to prove Theorem 6.8 is the following. It shows that, with high probability, the approximation class $\mathcal{H}_{\mathcal{U}}$ contains a weight function which is close, in both RKHS norm and risk, to the optimal $\alpha_{\mathcal{S}}$ guaranteed to exist by Lemma A.3.

**Lemma A.4** (Approximation error)**.** *Assume A1, A2. For any $\mathcal{S} \in (\mathcal{X} \times \mathcal{Y})^m$, take:*

$$
\alpha_{\mathcal{S}} \in \mathcal{H}_{\mathcal{S}} \cap \operatorname*{argmin}_{\alpha \in \mathcal{H}:[\![\alpha]\!] \leq B_1} \mathcal{L}_{\mathcal{S}}(\Lambda\alpha), \tag{147}
$$

*where $\mathcal{H}_{\mathcal{S}}$, defined in Equation (60), is the span of $\{\psi(x_i)\}_{i=1}^m$. With probability at least $1 - \delta$ over $\mathcal{S} \sim \mathcal{D}^m$ and $\mathcal{U} \sim p^T$, there exists some $\alpha_{\mathcal{S},\mathcal{U}} \in \mathcal{H}_{\mathcal{U}}$ with $\|\alpha_{\mathcal{S},\mathcal{U}}\|_{\mathcal{H}} \leq \theta B_1 + \frac{B_1 \kappa}{\sqrt{T}}\left(1 + 2\sqrt{2\log\frac{1}{\delta}}\right)$ such that:*

$$
\|\alpha_{\mathcal{S},\mathcal{U}} - \alpha_{\mathcal{S}}\|_{\mathcal{H}} \leq \frac{B_1 \kappa}{\sqrt{T}}\left(1 + 2\sqrt{2\log\frac{1}{\delta}}\right), \tag{148}
$$

*and:*

$$
|\mathcal{L}_{\mathcal{D}}(\Lambda\alpha_{\mathcal{S},\mathcal{U}}) - \mathcal{L}_{\mathcal{D}}(\Lambda\alpha_{\mathcal{S}})| \leq \frac{B_1 \rho \theta \kappa}{\sqrt{T}}\left(1 + 2\sqrt{2\log\frac{1}{\delta}}\right), \tag{149}
$$

*and:*

$$
|\mathcal{L}_{\mathcal{S}}(\Lambda\alpha_{\mathcal{S},\mathcal{U}}) - \mathcal{L}_{\mathcal{S}}(\Lambda\alpha_{\mathcal{S}})| \leq \frac{B_1 \rho \theta \kappa}{\sqrt{T}}\left(1 + 2\sqrt{2\log\frac{1}{\delta}}\right). \tag{150}
$$

*Proof.* Given $\mathcal{S} \sim \mathcal{D}^m$ and $\mathcal{U} \sim p^T$, we take:

$$
\alpha_{\mathcal{S},\mathcal{U}} = \frac{1}{T} \sum_{t=1}^T \alpha_{\mathcal{S},u_t}, \tag{151}
$$

where

$$\alpha_{\mathcal{S},u_t} := \sum_{i=1}^{m} a_i \phi(w_t, x_i) \mathcal{K}(w_t, \cdot). \tag{152}$$

First, we can show that the difference in risk between two predictors is upper bounded as a function of the RKHS norm of the difference of their weight function:

$$
\begin{aligned}
|\mathcal{L}_{\mathcal{D}}(\Lambda\alpha) - \mathcal{L}_{\mathcal{D}}(\Lambda\beta)| &= \left| \mathop{\mathbb{E}}_{(x,y)\sim\mathcal{D}} [\ell(\Lambda\alpha(x), y) - \ell(\Lambda\beta(x), y)] \right| \\
&\leq \mathop{\mathbb{E}}_{(x,y)\sim\mathcal{D}} \left[ |\ell(\Lambda\alpha(x), y) - \ell(\Lambda\beta(x), y)| \right] \\
&\leq \rho \mathop{\mathbb{E}}_{(x,y)\sim\mathcal{D}} \left[ |\Lambda\alpha(x) - \Lambda\beta(x)| \right] && (\ell \text{ is } \rho\text{-Lipschitz}) \\
&\leq \rho \mathop{\mathbb{E}}_{(x,y)\sim\mathcal{D}} \left[ \|\Lambda(\alpha - \beta)\|_{L_\infty(\mathcal{X})} \right] \\
&\leq \rho \|\Lambda\| \|\alpha - \beta\|_{\mathcal{H}} && (\text{Definition of } \|\Lambda\|) \\
&\leq \rho\theta \|\alpha - \beta\|_{\mathcal{H}}. && (\text{Theorem 4.1})
\end{aligned}
$$

Applied to $\alpha_{\mathcal{S},\mathcal{U}}$ and $\alpha_{\mathcal{S}}$, this means that:

$$|\mathcal{L}_{\mathcal{D}}(\Lambda\alpha_{\mathcal{S},\mathcal{U}}) - \mathcal{L}_{\mathcal{D}}(\Lambda\alpha_{\mathcal{S}})| \leq \rho\theta \|\alpha_{\mathcal{S},\mathcal{U}} - \alpha_{\mathcal{S}}\|_{\mathcal{H}}. \tag{153}$$

The same result also applies to the empirical risk:

$$|\mathcal{L}_{\mathcal{S}}(\Lambda\alpha_{\mathcal{S},\mathcal{U}}) - \mathcal{L}_{\mathcal{S}}(\Lambda\alpha_{\mathcal{S}})| \leq \rho\theta \|\alpha_{\mathcal{S},\mathcal{U}} - \alpha_{\mathcal{S}}\|_{\mathcal{H}}. \tag{154}$$

Therefore, $\Lambda\alpha_{\mathcal{S},\mathcal{U}}$ and $\Lambda\alpha_{\mathcal{S}}$ have similar predictions on the data distribution or training set if the weight functions $\alpha_{\mathcal{S},\mathcal{U}}$ and $\alpha_{\mathcal{S}}$ are close in RKHS space. We can also see that, for a fixed sample $\mathcal{S}$, $\alpha_{\mathcal{S}}$ is the expectation of $\alpha_{\mathcal{S},\mathcal{U}}$ with regard to the sampling of $\mathcal{U} \sim p^T$:

$$
\begin{aligned}
\mathop{\mathbb{E}}_{\mathcal{U}\sim p^T} [\alpha_{\mathcal{S},\mathcal{U}}] &= \mathop{\mathbb{E}}_{\mathcal{U}\sim p^T} \left[ \frac{1}{T} \sum_{t=1}^{T} \sum_{i=1}^{m} a_i \phi(w_t, x_i) \mathcal{K}(w_t, \cdot) \right] \\
&= \frac{1}{T} \sum_{t=1}^{T} \sum_{i=1}^{m} a_i \mathop{\mathbb{E}}_{\mathcal{U}\sim p^T} [\phi(w_t, x_i) \mathcal{K}(w_t, \cdot)] \\
&= \sum_{i=1}^{m} a_i \mathop{\mathbb{E}}_{w\sim p} [\phi(w, x_i) \mathcal{K}(w, \cdot)] \\
&= \sum_{i=1}^{m} a_i \psi(x_i) \\
&= \alpha_{\mathcal{S}}.
\end{aligned}
$$

We can derive an upper bound in high probability for $\|\alpha_{\mathcal{S},\mathcal{U}} - \alpha_{\mathcal{S}}\|_{\mathcal{H}}$ using Theorem A.3, which assures us that:

$$\mathop{\mathbb{P}}_{\mathcal{U}\sim p^T} \left[ \|\alpha_{\mathcal{S},\mathcal{U}} - \alpha_{\mathcal{S}}\|_{\mathcal{H}} \leq \mathop{\mathbb{E}}_{\mathcal{U}} [\|\alpha_{\mathcal{S},\mathcal{U}} - \alpha_{\mathcal{S}}\|_{\mathcal{H}}] + \sqrt{2v \log\frac{1}{\delta}} \right] \geq 1 - \delta, \tag{155}$$

where $v$ is the expected sum of squared differences. Specifically, if $\mathcal{U}' = \{u'_1, \ldots, u'_N\}$ is an identically and independently sampled copy of $\mathcal{U}$, and denoting $\mathcal{U}'_k := (\mathcal{U} \setminus \{u_k\}) \cup \{u'_k\}$, we have:

$$v = \mathop{\mathbb{E}}_{\mathcal{U},\mathcal{U}'} \left[ \sum_{k}^{N} (f(\mathcal{U}) - f(\mathcal{U}'_k))^2 \right], \tag{156}$$

where:

$$f(\mathcal{U}) := \|\alpha_{\mathcal{S},\mathcal{U}} - \alpha_{\mathcal{S}}\|_{\mathcal{H}}. \tag{157}$$

For some $k$, we have:

$$
\begin{aligned}
\mathbb{E}_{\mathcal{U},\mathcal{U}'}\left[(f(\mathcal{U}) - f(\mathcal{U}'_k))^2\right] &= \mathbb{E}_{\mathcal{U},\mathcal{U}'}\left[\left(\left\|\alpha_{\mathcal{S},\mathcal{U}} - \alpha_{\mathcal{S}}\right\|_{\mathcal{H}} - \left\|\alpha_{\mathcal{S},\mathcal{U}'_k} - \alpha_{\mathcal{S}}\right\|_{\mathcal{H}}\right)^2\right] \\
&\leq \mathbb{E}_{\mathcal{U},\mathcal{U}'}\left[\left\|\alpha_{\mathcal{S},\mathcal{U}} - \alpha_{\mathcal{S},\mathcal{U}'_k}\right\|_{\mathcal{H}}^2\right] \qquad\qquad\qquad \text{(Triangle inequality)} \\
&= \mathbb{E}_{\mathcal{U},\mathcal{U}'}\left[\left\|\frac{1}{T}\sum_{i=1}^{m} a_i(\phi(u_k, x_i)\mathcal{K}(u_k, \cdot) - \phi(u'_k, x_i)\mathcal{K}(u'_k, \cdot))\right\|_{\mathcal{H}}^2\right] \\
&= \frac{1}{T^2}\sum_{i=1}^{m}\sum_{j=1}^{m} a_i a_j \mathbb{E}_{\mathcal{U},\mathcal{U}'}\left[\langle\phi(u_k, x_i)\mathcal{K}(u_k, \cdot) - \phi(u'_k, x_i)\mathcal{K}(u'_k, \cdot),\right. \\
&\qquad\qquad\qquad\qquad\qquad \left. \phi(u_k, x_j)\mathcal{K}(u_k, \cdot) - \phi(u'_k, x_j)\mathcal{K}(u'_k, \cdot)\rangle\right] \\
&\leq \frac{1}{T^2}\sum_{i=1}^{m}\sum_{j=1}^{m} a_i a_j \mathbb{E}_{\mathcal{U},\mathcal{U}'}\left[\|\phi(u_k, x_i)\mathcal{K}(u_k, \cdot) - \phi(u'_k, x_i)\mathcal{K}(u'_k, \cdot)\|_{\mathcal{H}}\right. \\
&\qquad\qquad\qquad\qquad\qquad \left. \times \|\phi(u_k, x_j)\mathcal{K}(u_k, \cdot) - \phi(u'_k, x_j)\mathcal{K}(u'_k, \cdot)\|_{\mathcal{H}}\right] \\
&\leq \frac{1}{T^2}\sum_{i=1}^{m}\sum_{j=1}^{m} a_i a_j \mathbb{E}_{\mathcal{U},\mathcal{U}'}\left[(\|\phi(u_k, x_i)\mathcal{K}(u_k, \cdot)\|_{\mathcal{H}} + \|\phi(u'_k, x_i)\mathcal{K}(u'_k, \cdot)\|_{\mathcal{H}})\right. \\
&\qquad\qquad\qquad\qquad\qquad \left. \times \left(\|\phi(u_k, x_j)\mathcal{K}(u_k, \cdot)\|_{\mathcal{H}} - \|\phi(u'_k, x_j)\mathcal{K}(u'_k, \cdot)\|_{\mathcal{H}}\right)\right].
\end{aligned}
$$

Expanding the previous expression will yield four similar terms. The first one is:

$$
\mathbb{E}_{\mathcal{U},\mathcal{U}'}\left[\|\phi(u_k, x_i)\mathcal{K}(u_k, \cdot)\|_{\mathcal{H}}\|\phi(u_k, x_j)\mathcal{K}(u_k, \cdot)\|_{\mathcal{H}}\right]. \tag{158}
$$

This expectation in fact only depends on the sampling of $u_k$, and can therefore be seen as an inner product in $L^2(p)$. By Cauchy–Schwarz and the definition of $\kappa$ in Equation (50), this expectation is upper bounded by:

$$
\sqrt{\mathbb{E}_{u_k \sim p}\left[\|\phi(u_k, x_i)\mathcal{K}(u_k, \cdot)\|_{\mathcal{H}}^2\right]}\sqrt{\mathbb{E}_{u_k \sim p}\left[\|\phi(u_k, x_j)\mathcal{K}(u_k, \cdot)\|_{\mathcal{H}}^2\right]} \leq \kappa^2. \tag{159}
$$

The other three terms are upper bounded similarly, yielding:

$$
\mathbb{E}_{\mathcal{U},\mathcal{U}'}\left[(f(\mathcal{U}) - f(\mathcal{U}'_k))^2\right] \leq \frac{4\kappa^2}{T^2}\sum_{i=1}^{m}\sum_{j=1}^{m} a_i a_j = \frac{4\kappa^2\|a\|_1^2}{T^2} \leq \frac{4\kappa^2 B_1^2}{T^2}. \tag{160}
$$

Summing over all values of $k$, we get that:

$$
v \leq \frac{4\kappa^2 B_1^2}{T}. \tag{161}
$$

The expectation of $\|\alpha_{\mathcal{S},\mathcal{U}} - \alpha_{\mathcal{S}}\|_{\mathcal{H}}$ can be upper bounded as:

$$
\mathop{\mathbb{E}}_{\mathcal{U}} \left[ \|\alpha_{\mathcal{S},\mathcal{U}} - \alpha_{\mathcal{S}}\|_{\mathcal{H}} \right] \leq \sqrt{\mathop{\mathbb{E}}_{\mathcal{U}} \left[ \|\alpha_{\mathcal{S},\mathcal{U}} - \alpha_{\mathcal{S}}\|_{\mathcal{H}}^2 \right]} \tag{Jensen}
$$

$$
= \sqrt{\mathop{\mathbb{E}}_{\mathcal{U}} \left[ \left\| \frac{1}{T} \sum_{t=1}^{T} \alpha_{\mathcal{S},u_t} - \alpha_{\mathcal{S}} \right\|_{\mathcal{H}}^2 \right]}
$$

$$
= \sqrt{\frac{1}{T} \mathop{\mathbb{E}}_{\mathcal{U}} \left[ \|\alpha_{\mathcal{S},u_t}\|_{\mathcal{H}}^2 - \|\alpha_{\mathcal{S}}\|_{\mathcal{H}}^2 \right]}
$$

$$
\leq \sqrt{\frac{1}{T} \mathop{\mathbb{E}}_{\mathcal{U}} \left[ \|\alpha_{\mathcal{S},u_t}\|_{\mathcal{H}}^2 \right]}
$$

$$
= \sqrt{\frac{1}{T} \mathop{\mathbb{E}}_{u} \left[ \left\| \sum_{i=1}^{m} a_i \phi(u, x_i) \mathcal{K}(u, \cdot) \right\|_{\mathcal{H}}^2 \right]}
$$

$$
\leq \sqrt{\frac{1}{T} \|a\|_1^2 \kappa^2}
$$

$$
\leq \frac{\kappa B_1}{\sqrt{T}}.
$$

We get the result by applying Theorem A.3 with these upper bounds on $v$ and the expectation of our random variable $\|\alpha_{\mathcal{S},\mathcal{U}} - \alpha_{\mathcal{S}}\|_{\mathcal{H}}$, namely that with probability at least $1 - \delta$ over the sampling of $\mathcal{U}$, we have:

$$
\|\alpha_{\mathcal{S},\mathcal{U}} - \alpha_{\mathcal{S}}\|_{\mathcal{H}} \leq \frac{\kappa B_1}{\sqrt{T}} \left( 1 + 2\sqrt{2 \log \frac{1}{\delta}} \right), \tag{162}
$$

which implies the result. The fact that $\|\alpha_{\mathcal{S},\mathcal{U}}\|_{\mathcal{H}} \leq \theta B_1 + \frac{\kappa B_1}{\sqrt{T}} \left( 1 + 2\sqrt{2 \log \frac{1}{\delta}} \right)$ immediately follows from the triangle inequality ($\|\alpha_{\mathcal{S},\mathcal{U}}\|_{\mathcal{H}} \leq \|\alpha_{\mathcal{S}}\|_{\mathcal{H}} + \|\alpha_{\mathcal{S},\mathcal{U}} - \alpha_{\mathcal{S}}\|_{\mathcal{H}}$) and Lemma A.3, and holds whenever Equation (162) does. Finally, we can use Lemma A.1 to extend the result to also include the samplings of $\mathcal{S}$. $\qquad\square$

We can finally prove the main result.

*Proof of Theorem 6.8.* The existence of $\alpha_{\mathcal{S}}$ is guaranteed by Lemma A.3.

We seek a bound on $\mathcal{L}_{\mathcal{D}}(\Lambda\bar{\alpha}_{\mathcal{S},\mathcal{U}}) - \mathcal{L}_{\mathcal{S}}(\Lambda\alpha_{\mathcal{S}})$. Using the weight function $\alpha_{\mathcal{S},\mathcal{U}} \in \mathcal{H}_{\mathcal{U}}$ given by Lemma A.4, we break down the expression into three terms:

$$
\mathcal{L}_{\mathcal{D}}(\Lambda\bar{\alpha}_{\mathcal{S},\mathcal{U}}) - \mathcal{L}_{\mathcal{S}}(\Lambda\alpha_{\mathcal{S}}) = \mathcal{L}_{\mathcal{D}}(\Lambda\bar{\alpha}_{\mathcal{S},\mathcal{U}}) - \mathcal{L}_{\mathcal{S}}(\Lambda\bar{\alpha}_{\mathcal{S},\mathcal{U}})
$$
$$
+ \mathcal{L}_{\mathcal{S}}(\Lambda\bar{\alpha}_{\mathcal{S},\mathcal{U}}) - \mathcal{L}_{\mathcal{S}}(\Lambda\alpha_{\mathcal{S},\mathcal{U}})
$$
$$
+ \mathcal{L}_{\mathcal{S}}(\Lambda\alpha_{\mathcal{S},\mathcal{U}}) - \mathcal{L}_{\mathcal{S}}(\Lambda\alpha_{\mathcal{S}}).
$$

**Bounding $\mathcal{L}_{\mathcal{S}}(\Lambda\alpha_{\mathcal{S},\mathcal{U}}) - \mathcal{L}_{\mathcal{S}}(\Lambda\alpha_{\mathcal{S}})$.** We know from Lemma A.4 that:

$$
\mathop{\mathbb{P}}_{\mathcal{S}\sim\mathcal{D}^m, \mathcal{U}\sim p^T} \left[ |\mathcal{L}_{\mathcal{S}}(\Lambda\alpha_{\mathcal{S},\mathcal{U}}) - \mathcal{L}_{\mathcal{S}}(\Lambda\alpha_{\mathcal{S}})| \leq \frac{\rho\theta\kappa B_1}{\sqrt{T}} \left( 1 + 2\sqrt{2 \log \frac{1}{\delta}} \right) \right] \geq 1 - \delta. \tag{163}
$$

The inequality $\|\alpha_{\mathcal{S},\mathcal{U}}\|_{\mathcal{H}} \leq B' := \theta B_1 + \frac{\kappa B_1}{\sqrt{T}} \left( 1 + 2\sqrt{2 \log \frac{1}{\delta}} \right)$ holds simultaneously.

**Bounding** $\mathcal{L}_{\mathcal{D}}(\Lambda\bar{\alpha}_{\mathcal{S},\mathcal{U}}) - \mathcal{L}_{\mathcal{S}}(\Lambda\bar{\alpha}_{\mathcal{S},\mathcal{U}})$. Theorem 6.2 gives us that the following holds simultaneously for all $\alpha \in \mathcal{H}_{B'}$ with probability at least $1 - \delta$ over the choice of sample $\mathcal{S}$:

$$
\begin{aligned}
\mathcal{L}_{\mathcal{D}}(\Lambda\alpha) - \mathcal{L}_{\mathcal{S}}(\Lambda\alpha) &\leq \frac{2\rho\theta B'}{\sqrt{m}}\left(1 + \sqrt{\frac{\log\frac{1}{\delta}}{2}}\right) \\
&\leq \frac{2\rho\theta}{\sqrt{m}}\left(\theta B_1 + \frac{\kappa B_1}{\sqrt{T}}\left(1 + 2\sqrt{2\log\frac{1}{\delta}}\right)\right)\left(1 + \sqrt{\frac{\log\frac{1}{\delta}}{2}}\right) \\
&= \frac{2\rho\theta B_1}{\sqrt{m}}\left(\theta + \frac{\kappa}{\sqrt{T}}\left(1 + 2\sqrt{2\log\frac{1}{\delta}}\right)\right)\left(1 + \sqrt{\frac{\log\frac{1}{\delta}}{2}}\right).
\end{aligned}
\tag{164}
$$

This result applies in particular to $\Lambda\bar{\alpha}_{\mathcal{S},\mathcal{U}}$.

**Bounding** $\mathcal{L}_{\mathcal{S}}(\Lambda\bar{\alpha}_{\mathcal{S},\mathcal{U}}) - \mathcal{L}_{\mathcal{S}}(\Lambda\alpha_{\mathcal{S},\mathcal{U}})$. Since $\Lambda\bar{\alpha}_{\mathcal{S},\mathcal{U}}$ minimizes the empirical risk, the term $\mathcal{L}_{\mathcal{S}}(\Lambda\bar{\alpha}_{\mathcal{S},\mathcal{U}}) - \mathcal{L}_{\mathcal{S}}(\Lambda\alpha_{\mathcal{S},\mathcal{U}})$ is smaller than 0, and can simply be ignored.

**Assembling the ingredients.** Equations (163) and (164) hold independently each with probability at least $1 - \delta$. Using Lemma A.1, we also know that they hold with probability at least $1 - \delta$ over the samplings of both $\mathcal{S}$ and $\mathcal{U}$. By the union bound, they hold together with probability at least $1 - 2\delta$. Equation (104) is obtained by replacing $\delta$ by $\frac{\delta}{2}$. $\qquad\square$

## B  Calculus

In this appendix, we present all the calculus required to calculate the expectation $\mathbb{E}_{w \sim p}\left[\mathcal{K}(u, w)\phi(w, x)\right]$, as well as constants $\theta$ and $\kappa$, for all instantiations. We begin with a few simple lemmas.

**Lemma B.1.** *Consider a Hilbert space $\mathcal{W}$. Let $u, w \in \mathcal{W}$ and $a, b > 0$. Then:*

$$
a\|w - u\|^2 + b\|w\|^2 = (a + b)\left\|w - \frac{a}{a + b}u\right\|^2 + \frac{ab}{a + b}\|u\|^2.
$$

*Proof.* We have:

$$
\begin{aligned}
a\|w - u\|^2 + b\|w\|^2 &= a\|w\|^2 - 2a\langle w, u\rangle + a\|u\|^2 + b\|w\|^2 \\
&= (a + b)\|w\|^2 - 2a\langle w, u\rangle + a\|u\|^2 \\
&= (a + b)\|w\|^2 - 2a\langle w, u\rangle + \frac{a^2}{a + b}\|u\|^2 - \frac{a^2}{a + b}\|u\|^2 + a\|u\|^2 \\
&= (a + b)\left[\|w\|^2 - 2\frac{a}{a + b}\langle w, u\rangle + \frac{a^2}{(a + b)^2}\|u\|^2\right] - \frac{a^2}{a + b}\|u\|^2 + a\|u\|^2 \\
&= (a + b)\left\|w - \frac{a}{a + b}u\right\|^2 + \frac{ab}{a + b}\|u\|^2.
\end{aligned}
$$

$\qquad\square$

**Lemma B.2.** *Consider a Hilbert space $\mathcal{W}$. Let $u, w \in \mathcal{W}$ and $a, b > 0$. Then:*

$$
a\|w - u\|^2 + b\|w - w_0\|^2 = (a + b)\left\|w - \frac{au + bw_0}{a + b}\right\|^2 + a\|u\|^2 + b\|w_0\|^2 - \frac{1}{a + b}\|au + bw_0\|^2.
$$

*Proof.* Same process as above. $\qquad\square$

**Lemma B.3.** *Consider a Hilbert space $\mathcal{W}$. Let $u, w \in \mathcal{W}$ and $a, b > 0$. Then:*

$$\frac{\|w - u\|^2}{a} + \frac{\|w\|^2}{b} = \left(\frac{1}{a} + \frac{1}{b}\right)\left\|w - \frac{1}{1 + \frac{a}{b}}u\right\|^2 + \frac{1}{a + b}\|u\|^2.$$

*Proof.* We have:

$$\frac{\|w - u\|^2}{a} + \frac{\|w\|^2}{b} = \left(\frac{1}{a} + \frac{1}{b}\right)\left\|w - \frac{1}{a\left(\frac{1}{a} + \frac{1}{b}\right)}u\right\|^2 + \frac{1}{ab\left(\frac{1}{a} + \frac{1}{b}\right)}\|u\|^2 \qquad \text{(Lemma B.1)}$$

$$= \left(\frac{1}{a} + \frac{1}{b}\right)\left\|w - \frac{1}{1 + \frac{a}{b}}u\right\|^2 + \frac{1}{a + b}\|u\|^2.$$

$\square$

**Lemma B.4.** *We have:*

$$\int_{\mathbb{R}^n} e^{-a\|w - u\|^2}\mathrm{d}w = \left(\frac{\pi}{a}\right)^{\frac{n}{2}}.$$

*Proof.* This is the unnormalized integral of a Gaussian density with variance $\frac{1}{2a}I$ and mean $u$. $\square$

## B.1 Calculus for RWSign

Throughout this section, we assume that $(\mathcal{W}, \phi, \mathcal{K}, p)$ are those of instantiation RWSign. See Table 2 for details. We work up to the full integral in $\mathbb{R}^n$ through a series of lemmas.

**Lemma B.5.** *We have:*

$$\int_{-\infty}^{\infty} e^{-(w-u)^2/2\gamma^2} \operatorname{sign}(wx)\mathrm{d}w = \sqrt{2\pi}\gamma \operatorname{sign}(x) \operatorname{erf}\left(\frac{u}{\sqrt{2}\gamma}\right).$$

*Proof.* Applying an adequate change of variable causes the error function to appear:

$$\int_{-\infty}^{\infty} e^{-(w-u)^2/2\gamma^2} \operatorname{sign}(wx)\mathrm{d}w$$

$$= \operatorname{sign}(x) \int_{-\infty}^{\infty} e^{-(w-u)^2/2\gamma^2} \operatorname{sign}(w)\mathrm{d}w$$

$$= \operatorname{sign}(x)\left(\int_0^{\infty} e^{-(w-u)^2/2\gamma^2}\mathrm{d}w - \int_{-\infty}^0 e^{-(w-u)^2/2\gamma^2}\mathrm{d}w\right)$$

$$= \sqrt{2}\gamma \operatorname{sign}(x)\left(\int_{\frac{-u}{\sqrt{2}\gamma}}^{\infty} e^{-t^2}\mathrm{d}w - \int_{-\infty}^{\frac{-u}{\sqrt{2}\gamma}} e^{-t^2}\mathrm{d}w\right) \qquad (t := \tfrac{w-u}{\sqrt{2}\gamma},\ \mathrm{d}w = \tfrac{\mathrm{d}w}{\sqrt{2}\gamma})$$

$$= \sqrt{2}\gamma \operatorname{sign}(x)\left(\int_0^{\infty} e^{-t^2}\mathrm{d}w + \int_{\frac{-u}{\sqrt{2}\gamma}}^0 e^{-t^2}\mathrm{d}w - \int_{-\infty}^0 e^{-t^2}\mathrm{d}w - \int_0^{\frac{-u}{\sqrt{2}\gamma}} e^{-t^2}\mathrm{d}w\right)$$

$$= \sqrt{2}\gamma \operatorname{sign}(x)\left(\int_{\frac{-u}{\sqrt{2}\gamma}}^0 e^{-t^2}\mathrm{d}w - \int_0^{\frac{-u}{\sqrt{2}\gamma}} e^{-t^2}\mathrm{d}w\right)$$

$$= \sqrt{2}\gamma \operatorname{sign}(x)\left(\int_0^{\frac{u}{\sqrt{2}\gamma}} e^{-t^2}\mathrm{d}w + \int_0^{\frac{u}{\sqrt{2}\gamma}} e^{-t^2}\mathrm{d}w\right)$$

$$= 2\sqrt{2}\gamma \operatorname{sign}(x) \int_0^{\frac{u}{\sqrt{2}\gamma}} e^{-t^2}\mathrm{d}w$$

$$= 2\sqrt{2}\gamma \operatorname{sign}(x) \frac{\sqrt{\pi}}{2} \operatorname{erf}\left(\frac{u}{\sqrt{2}\gamma}\right)$$

$$= \sqrt{2\pi}\gamma \operatorname{sign}(x) \operatorname{erf}\left(\frac{u}{\sqrt{2}\gamma}\right).$$

$\square$

**Lemma B.6.** *We have:*

$$\int_{\mathbb{R}^n} e^{-\|w-u\|^2/2\gamma^2} \operatorname{sign}(\langle w, x\rangle)\mathrm{d}w = \left(\sqrt{2\pi}\gamma\right)^n \operatorname{erf}\left(\frac{\langle u, x\rangle}{\sqrt{2}\gamma\|x\|}\right).$$

*Proof.* Calculate the integral using an orthonormal basis $\{v_1, \ldots, v_n\}$ of $\mathbb{R}^n$ such that $v_n := \frac{x}{\|x\|}$. Write $w = (w_1, \ldots, w_n)$ in this new basis (i.e. $w_i := \langle w, v_i\rangle$ for all $i$), and similarly $(u_1, \ldots, u_n)$ for $u$. Under this change of coordinates, the integral becomes:

$$\int_{\mathbb{R}^n} e^{-\|w-u\|^2/2\gamma^2} \operatorname{sign}(\langle w, x\rangle)\mathrm{d}w$$

$$= \int_{\mathbb{R}}\int_{\mathbb{R}^{n-1}} e^{-\left[\sum_{i=1}^{n-1}(w_i-u_i)^2 + (w_n-u_n)^2\right]/2\gamma^2} \operatorname{sign}(w_n\|x\|)\mathrm{d}w_1, \ldots, \mathrm{d}w_{n-1}\mathrm{d}w_n.$$

We are left with a product of $n$ independent integrals:

$$\int_{\mathbb{R}^n} e^{-\|w-u\|^2/2\gamma^2} \operatorname{sign}(\langle w, x\rangle)\mathrm{d}w$$

$$= \int_{\mathbb{R}^{n-1}} e^{-\sum_{i=1}^{n-1}(w_i-u_i)^2/2\gamma^2}\mathrm{d}w_1 \ldots \mathrm{d}w_{n-1} \int_{\mathbb{R}} e^{-(w_n-u_n)^2/2\gamma^2} \operatorname{sign}(w_n\|x\|)\mathrm{d}w_n$$

$$= \int_{\mathbb{R}^{n-1}} \prod_{i=1}^{n-1} e^{-(w_i-u_i)^2/2\gamma^2}\mathrm{d}w_1 \ldots \mathrm{d}w_{n-1} \int_{\mathbb{R}} e^{-(w_n-u_n)^2/2\gamma^2} \operatorname{sign}(w_n\|x\|)\mathrm{d}w_n$$

$$= \prod_{i=1}^{n-1} \int_{\mathbb{R}} e^{-(w_i-u_i)^2/2\gamma^2}\mathrm{d}w_i \int_{\mathbb{R}} e^{-(w_n-u_n)^2/2\gamma^2} \operatorname{sign}(w_n\|x\|)\mathrm{d}w_n.$$

For each $i$, Lemma B.4 gives us:

$$\int_{\mathbb{R}} e^{-(w_i - u_i)^2/2\gamma^2} \mathrm{d}w_i = \sqrt{2\pi}\gamma.$$

Also, Lemma B.5 gives us:

$$\int_{\mathbb{R}} e^{-(w_n - u_n)^2/2\gamma^2} \mathrm{sign}(w_n\|x\|)\mathrm{d}w_n = \sqrt{2\pi}\gamma\,\mathrm{erf}\left(\frac{u_n}{\sqrt{2}\gamma}\right).$$

Finally, since $u_n = \frac{\langle u, x\rangle}{\|x\|}$, we have the result. $\qquad\square$

**Lemma B.7.** *We have:*

$$\mathbb{E}_{w\sim p}[\mathcal{K}(u,w)\phi(w,x)] = \left(1 + \tfrac{\sigma^2}{\gamma^2}\right)^{-n/2} e^{\frac{-\|u\|_2^2}{2\sigma^2 + 2\gamma^2}}\,\mathrm{erf}\left(\frac{\langle u', x\rangle}{\sqrt{2}\zeta\|x\|_2}\right),$$

*where $\zeta$ is defined by the relationship:*

$$\frac{1}{2\zeta^2} = \frac{1}{2\gamma^2} + \frac{1}{2\sigma^2},$$

*and:*

$$u' := \left(1 + \tfrac{\gamma^2}{\sigma^2}\right)^{-1} u.$$

*Proof.* The proof is simply completing the square at the exponent and applying Lemma B.6. We have:

$$\mathbb{E}_{w\sim p}[\mathcal{K}(u,w)\phi(w,x)] = \left(\frac{1}{\sqrt{2\pi\sigma^2}}\right)^n \int_{\mathbb{R}^n} e^{-\|w-u\|^2/2\gamma^2} e^{-\|w\|^2/2\sigma^2}\,\mathrm{sign}(\langle w, x\rangle)\mathrm{d}w.$$

Lemma B.3 gives us:

$$-\|w - u\|^2/2\gamma^2 - \|w\|^2/2\sigma^2 = -\frac{1}{2\zeta^2}\left\|w - \left(1 + \frac{\gamma^2}{\sigma^2}\right)^{-1} u\right\|^2 - \left(\frac{1}{2\sigma^2 + 2\gamma^2}\right)\|u\|^2$$

$$= -\frac{1}{2\zeta^2}\|w - u'\|^2 - \left(\frac{1}{2\sigma^2 + 2\gamma^2}\right)\|u\|^2.$$

Therefore, we have:

$$\mathbb{E}_{w\sim p}[\mathcal{K}(u,w)\phi(w,x)] = \left(\frac{1}{\sqrt{2\pi\sigma^2}}\right)^n e^{-\|u\|^2/(2\gamma^2 + 2\sigma^2)} \int_{\mathbb{R}^n} e^{-\|w-u'\|^2/2\zeta^2}\,\mathrm{sign}(\langle w, x\rangle)\mathrm{d}w.$$

Applying Lemma B.6, we get:

$$\mathbb{E}_{w\sim p}[\mathcal{K}(u,w)\phi(w,x)] = \left(\frac{\zeta}{\sigma}\right)^n e^{-\|u\|^2/(2\gamma^2 + 2\sigma^2)}\,\mathrm{erf}\left(\frac{\langle u', x\rangle}{\sqrt{2}\zeta\|x\|}\right).$$

We obtain the final result by noticing that.

$$\left(\frac{\zeta}{\sigma}\right)^n = \left(1 + \tfrac{\sigma^2}{\gamma^2}\right)^{-n/2}.$$

$\qquad\square$

To calculate $\theta$, we need a few more lemmas.

**Lemma B.8.** *Consider $\sigma > 0$ and $\gamma > 0$. Then:*

$$\int_{\mathbb{R}^n} \int_{\mathbb{R}^n} e^{-\|u-w\|^2/2\gamma^2} e^{-\langle u,w \rangle/\sigma^2} \mathrm{d}u\mathrm{d}w = (2\pi)^n \left(\frac{\gamma^2 \sigma^4}{2\sigma^2 - \gamma^2}\right)^{n/2}. \tag{165}$$

*Proof.*

$$\begin{aligned}
\int_{\mathbb{R}^n} \int_{\mathbb{R}^n} e^{-\frac{\|u-w\|^2}{2\gamma^2}} e^{-\frac{\langle u,w \rangle}{\sigma^2}} \mathrm{d}u\mathrm{d}w &= \int_{\mathbb{R}^n} \int_{\mathbb{R}^n} e^{-\frac{\|t\|^2}{2\gamma^2}} e^{-\frac{\langle t+w,w \rangle}{\sigma^2}} \mathrm{d}t\mathrm{d}w && (t := u - w, \ \mathrm{d}t = \mathrm{d}u) \\
&= \int_{\mathbb{R}^n} \int_{\mathbb{R}^n} e^{-\frac{\|t\|^2}{2\gamma^2}} e^{-\frac{\|w\|^2}{\sigma^2}} e^{-\frac{\langle t,w \rangle}{\sigma^2}} \mathrm{d}t\mathrm{d}w \\
&= \int_{\mathbb{R}^n} e^{-\frac{\|w\|^2}{\sigma^2}} \left[\int_{\mathbb{R}^n} e^{-\frac{\|t\|^2}{2\gamma^2}} e^{-\frac{\langle t,w \rangle}{\sigma^2}} \mathrm{d}t\right] \mathrm{d}w \\
&= \int_{\mathbb{R}^n} e^{-\frac{\|w\|^2}{\sigma^2}} \left[\int_{\mathbb{R}^n} e^{-\frac{\left\|t+\frac{\gamma^2 w}{\sigma^2}\right\|^2}{2\gamma^2}} e^{\frac{\left\|\frac{\gamma^2 w}{\sigma^2}\right\|^2}{2\gamma^2}} \mathrm{d}t\right] \mathrm{d}w \\
&= \int_{\mathbb{R}^n} e^{-\frac{\|w\|^2}{\sigma^2}} e^{\frac{\left\|\frac{\gamma^2 w}{\sigma^2}\right\|^2}{2\gamma^2}} \left[\int_{\mathbb{R}^n} e^{-\frac{\left\|t+\frac{\gamma^2 w}{\sigma^2}\right\|^2}{2\gamma^2}} \mathrm{d}t\right] \mathrm{d}w \\
&= \left(\sqrt{2\pi\gamma^2}\right)^n \int_{\mathbb{R}^n} e^{-\frac{\|w\|^2}{\sigma^2}} e^{\frac{\left\|\frac{\gamma^2 w}{\sigma^2}\right\|^2}{2\gamma^2}} \mathrm{d}w.
\end{aligned}$$

Then, simplifying the exponent:

$$\begin{aligned}
-\frac{\|w\|^2}{\sigma^2} + \frac{\left\|\frac{\gamma^2 w}{\sigma^2}\right\|^2}{2\gamma^2} &= -\frac{\|w\|^2}{2\sigma^2}\left(2 - \frac{\gamma^2}{\sigma^2}\right) \\
&= -\frac{\|w\|^2}{2\sigma^2}\left(\frac{2\sigma^2 - \gamma^2}{\sigma^2}\right) \\
&= -\frac{\|w\|^2}{2\sigma^4}(2\sigma^2 - \gamma^2),
\end{aligned}$$

we get:

$$\begin{aligned}
\int_{\mathbb{R}^n} \int_{\mathbb{R}^n} e^{-\frac{\|u-w\|^2}{2\gamma^2}} e^{-\frac{\langle u,w \rangle}{\sigma^2}} \mathrm{d}u\mathrm{d}w &= \left(\sqrt{2\pi\gamma^2}\right)^n \int_{\mathbb{R}^n} e^{-\frac{\|w\|^2}{2\sigma^4}\left(2\sigma^2 - \gamma^2\right)} \mathrm{d}w \\
&= \left(\sqrt{2\pi\gamma^2}\right)^n \left(\sqrt{2\pi \frac{\sigma^4}{2\sigma^2 - \gamma^2}}\right)^n \\
&= (2\pi)^n \left(\frac{\gamma^2 \sigma^4}{2\sigma^2 - \gamma^2}\right)^{n/2}.
\end{aligned}$$

$\square$

**Lemma B.9.** *Consider $\sigma > 0$ and $\gamma > 0$. Denote $I_n$ the identity matrix in $\mathbb{R}^n$. Then:*

$$\mathop{\mathbb{E}}_{w \sim \mathcal{N}(0,\sigma^2 I_n)} \mathop{\mathbb{E}}_{u \sim \mathcal{N}(0,\sigma^2 I_n)} \left[e^{-\|u-w\|^2/2\gamma^2}\right] = \left(1 + \frac{2\sigma^2}{\gamma^2}\right)^{-n/2}. \tag{166}$$

*Proof.* The expectation is a straightforward integral:

$$
\begin{aligned}
\mathop{\mathbb{E}}_{w\sim p}\mathop{\mathbb{E}}_{u\sim p}[\mathcal{K}(u,w)] &= \left(\frac{1}{\sqrt{2\pi\sigma^2}}\right)^{2n}\int_{\mathbb{R}^n}\int_{\mathbb{R}^n}e^{-\frac{\|u-w\|^2}{2\gamma^2}}e^{-\frac{\|u\|^2}{2\sigma^2}}e^{-\frac{\|w\|^2}{2\sigma^2}}\,\mathrm{d}u\mathrm{d}w\\
&= \left(\frac{1}{\sqrt{2\pi\sigma^2}}\right)^{2n}\int_{\mathbb{R}^n}\int_{\mathbb{R}^n}e^{-\frac{\|u-w\|^2}{2\gamma^2}}e^{-\frac{\|u\|^2}{2\sigma^2}}e^{\frac{\langle u,w\rangle}{\sigma^2}}e^{-\frac{\|w\|^2}{2\sigma^2}}e^{-\frac{\langle u,w\rangle}{\sigma^2}}\,\mathrm{d}u\mathrm{d}w\\
&= \left(\frac{1}{\sqrt{2\pi\sigma^2}}\right)^{2n}\int_{\mathbb{R}^n}\int_{\mathbb{R}^n}e^{-\frac{\|u-w\|^2}{2\gamma^2}}e^{-\frac{\|u-w\|^2}{2\sigma^2}}e^{-\frac{\langle u,w\rangle}{\sigma^2}}\,\mathrm{d}u\mathrm{d}w\\
&= \left(\frac{1}{\sqrt{2\pi\sigma^2}}\right)^{2n}\int_{\mathbb{R}^n}\int_{\mathbb{R}^n}e^{-\frac{\|u-w\|^2}{2\zeta^2}}e^{-\frac{\langle u,w\rangle}{\sigma^2}}\,\mathrm{d}u\mathrm{d}w \qquad \left(\tfrac{1}{2\zeta^2}=\tfrac{1}{2\gamma^2}+\tfrac{1}{2\sigma^2}=\tfrac{\sigma^2+\gamma^2}{2\sigma^2\gamma^2}\right)\\
&= \left(\frac{1}{\sqrt{2\pi\sigma^2}}\right)^{2n}(2\pi)^n\left(\frac{\zeta^2\sigma^4}{2\sigma^2-\zeta^2}\right)^{n/2} \qquad\qquad\text{(Lemma B.8)}\\
&= \left(\frac{\zeta^2}{2\sigma^2-\zeta^2}\right)^{n/2}\\
&= \left(\frac{2\sigma^2}{\zeta^2}-1\right)^{-n/2}\\
&= \left(\frac{2\sigma^2}{\sigma^2}+\frac{2\sigma^2}{\gamma^2}-1\right)^{-n/2}\\
&= \left(1+\frac{2\sigma^2}{\gamma^2}\right)^{-n/2}.
\end{aligned}
$$

$\square$

**Lemma B.10.** *We have:*

$$
\theta \le \left(1+\tfrac{2\sigma^2}{\gamma^2}\right)^{-n/4}. \tag{167}
$$

*Proof.* We have:

$$
\begin{aligned}
\theta^2 &= \sup_{x\in\mathcal{X}}\|\psi(x)\|_{\mathcal{H}}^2\\
&= \sup_{x\in\mathcal{X}}\mathop{\mathbb{E}}_{w\sim p}\mathop{\mathbb{E}}_{u\sim p}[\mathcal{K}(u,w)\phi(u,x)\phi(w,x)]\\
&\le \sup_{x\in\mathcal{X}}\mathop{\mathbb{E}}_{w\sim p}\mathop{\mathbb{E}}_{u\sim p}[|\mathcal{K}(u,w)\phi(u,x)\phi(w,x)|]\\
&= \mathop{\mathbb{E}}_{w\sim p}\mathop{\mathbb{E}}_{u\sim p}[\mathcal{K}(u,w)]. \qquad\qquad (|\phi(w,x)|=1\text{ for all }w\text{ and }x)
\end{aligned}
$$

The result is given by Lemma B.9 (and taking the square root). $\square$

### B.2 Calculus for RWExpSign

Throughout this section, we assume that $(\mathcal{W},\phi,\mathcal{K},p)$ are those of instantiation RWExpSign. See Table 2 for details.

**Lemma B.11.** *We have:*

$$
\mathop{\mathbb{E}}_{w\sim p}[\mathcal{K}(u,w)\phi(w,x)] = e^{\frac{\sigma^2}{8\gamma^4}\|u\|^2}\operatorname{erf}\left(\frac{\sigma}{\sqrt{8\pi}\gamma^2}\frac{\langle u,x\rangle}{\|x\|}\right). \tag{168}
$$

*Proof.* We have:

$$
\mathop{\mathbb{E}}_{w\sim p}[\mathcal{K}(u,w)\phi(w,x)] = \left(\frac{1}{\sqrt{2\pi\sigma^2}}\right)^n\int_{\mathbb{R}^n}e^{\langle u,w\rangle/2\gamma^2}e^{-\|w\|^2/2\sigma^2}\operatorname{sign}(\langle w,x\rangle)\,\mathrm{d}w. \tag{169}
$$

Completing the square, we get:

$$\underset{w\sim p}{\mathbb{E}}[\mathcal{K}(u,w)\phi(w,x)] = \left(\frac{1}{\sqrt{2\pi\sigma^2}}\right)^n \int_{\mathbb{R}^n} e^{-\left\|w-\frac{\sigma^2}{2\gamma^2}u\right\|^2/2\sigma^2 + \left\|\frac{\sigma^2}{2\gamma^2}u\right\|^2/2\sigma^2} \operatorname{sign}(\langle w,x\rangle)\mathrm{d}w. \tag{170}$$

Defining $u' := \frac{\sigma^2}{2\gamma^2}u$ and rearranging the terms, we get:

$$\underset{w\sim p}{\mathbb{E}}[\mathcal{K}(u,w)\phi(w,x)] = \left(\frac{1}{\sqrt{2\pi\sigma^2}}\right)^n e^{\frac{\sigma^2}{8\gamma^4}\|u\|^2} \int_{\mathbb{R}^n} e^{-\left\|w-u'\right\|^2/2\sigma^2} \operatorname{sign}(\langle w,x\rangle)\mathrm{d}w. \tag{171}$$

Applying Lemma B.6, we get the final results:

$$\underset{w\sim p}{\mathbb{E}}[\mathcal{K}(u,w)\phi(w,x)] = \left(\frac{1}{\sqrt{2\pi\sigma^2}}\right)^n e^{\frac{\sigma^2}{8\gamma^4}\|u\|^2} \left(\sqrt{2\pi\sigma^2}\right)^n \operatorname{erf}\left(\frac{1}{\sqrt{2\pi\sigma^2}}\frac{\langle u',x\rangle}{\|x\|}\right)$$

$$= e^{\frac{\sigma^2}{8\gamma^4}\|u\|^2} \operatorname{erf}\left(\frac{\sigma}{\sqrt{8\pi\gamma^2}}\frac{\langle u,x\rangle}{\|x\|}\right). \tag{172}$$

$\square$

**Lemma B.12.** *We have:*

$$\theta \le \left(\frac{1}{1-\frac{\sigma^2}{2\gamma^2}}\right)^{n/2}. \tag{173}$$

*Proof.* First, notice that we have:

$$\theta := \sup_{x\in\mathcal{X}} \left\|\underset{w\sim p}{\mathbb{E}}[\phi(w,x)\mathcal{K}(w,\cdot)]\right\|_{\mathcal{H}}$$

$$\le \sup_{x\in\mathcal{X}} \underset{w\sim p}{\mathbb{E}}\left[|\phi(w,x)|\sqrt{\mathcal{K}(w,w)}\right] =: \iota.$$

We have:

$$\iota := \sup_{x\in\mathcal{X}} \underset{w\sim p}{\mathbb{E}}\left[\sqrt{\mathcal{K}(w,w)}|\phi(w,x)|\right]$$

$$= \underset{w\sim p}{\mathbb{E}}\left[\sqrt{\mathcal{K}(w,w)}\right] \qquad\qquad (|\phi(w,x)| = |\operatorname{sign}(\langle w,x\rangle)| = 1)$$

$$= \left(\frac{1}{\sqrt{2\pi\sigma^2}}\right)^n \int_{\mathbb{R}^n} e^{\|w\|^2/4\gamma^2} e^{-\|w\|^2/2\sigma^2}\mathrm{d}w$$

$$= \left(\frac{1}{\sqrt{2\pi\sigma^2}}\right)^n \int_{\mathbb{R}^n} e^{-\|w\|^2\left(1/2\sigma^2 - 1/4\gamma^2\right)}\mathrm{d}w$$

$$= \left(\frac{1}{\sqrt{2\pi\sigma^2}}\right)^n \left(\sqrt{2\pi\zeta^2}\right)^n \qquad\qquad (\tfrac{1}{2\zeta^2} := \tfrac{1}{2\sigma^2} - \tfrac{1}{4\gamma^2})$$

$$= \left(\frac{\zeta}{\sigma}\right)^n$$

$$= \left(\frac{1}{\sigma}\sqrt{\frac{1}{\frac{1}{\sigma^2}-\frac{1}{2\gamma^2}}}\right)^n$$

$$= \left(\frac{1}{1-\frac{\sigma^2}{2\gamma^2}}\right)^{n/2}.$$

$\square$

**Lemma B.13.** *We have:*

$$\kappa = \left(\frac{1}{1-\frac{\sigma^2}{\gamma^2}}\right)^{n/4}. \tag{174}$$

*Proof.* We have:

$$
\begin{aligned}
\kappa^2 &:= \sup_{x \in \mathcal{X}} \mathop{\mathbb{E}}_{w \sim p} \left[ \mathcal{K}(w, w) \phi(w, x)^2 \right] \\
&= \mathop{\mathbb{E}}_{w \sim p} \left[ \mathcal{K}(w, w) \right] && (\phi(w,x)^2 = \text{sign}(\langle w, x \rangle)^2 = 1) \\
&= \left( \frac{1}{\sqrt{2\pi\sigma^2}} \right)^n \int_{\mathbb{R}^n} e^{\|w\|^2/2\gamma^2} e^{-\|w\|^2/2\sigma^2} \mathrm{d}w \\
&= \left( \frac{1}{\sqrt{2\pi\sigma^2}} \right)^n \int_{\mathbb{R}^n} e^{-\|w\|^2 \left( 1/2\sigma^2 - 1/2\gamma^2 \right)} \mathrm{d}w \\
&= \left( \frac{1}{\sqrt{2\pi\sigma^2}} \right)^n \left( \sqrt{2\pi\zeta^2} \right)^n && (\tfrac{1}{2\zeta^2} := \tfrac{1}{2\sigma^2} - \tfrac{1}{2\gamma^2}) \\
&= \left( \frac{\zeta}{\sigma} \right)^n \\
&= \left( \frac{1}{\sigma} \sqrt{\frac{1}{\frac{1}{\sigma^2} - \frac{1}{\gamma^2}}} \right)^n \\
&= \left( \frac{1}{1 - \frac{\sigma^2}{\gamma^2}} \right)^{n/2}.
\end{aligned}
$$

We get the result by taking the square root. $\qquad\square$

### B.3  Calculus for RWExpRelu

Throughout this section, we assume that $(\mathcal{W}, \phi, \mathcal{K}, p)$ are those of instantiation RWExpRelu. See Table 2 for details. The calculus for RWExpRelu relies on the calculus of Dubé & Marchand (2025) for instantiation RWRelu.

**Lemma B.14.** *We have:*

$$
\mathop{\mathbb{E}}_{w \sim p} \left[ \mathcal{K}(u, w) \phi(w, x) \right] = \left( \frac{1}{\sqrt{2\pi\sigma^2}} \right) e^{\frac{\sigma^2}{8\gamma^4} \|u\|_2^2} \left[ \frac{\sigma \|x\|}{\sqrt{2}} \left( \sqrt{2}\sigma e^{-\frac{\langle u, x \rangle^2}{2\sigma^2 \|x\|^2}} + \sqrt{\pi} \frac{\langle u, x \rangle}{\|x\|} \left[ 1 + \text{erf} \left( \frac{\langle u, x \rangle}{\sqrt{2}\sigma\|x\|} \right) \right] \right) \right].
\tag{175}
$$

*Proof.* We have:

$$
\mathop{\mathbb{E}}_{w \sim p} \left[ \mathcal{K}(u, w) \phi(w, x) \right] = \left( \frac{1}{\sqrt{2\pi\sigma^2}} \right)^n \int_{\mathbb{R}^n} e^{\langle u, w \rangle/2\gamma^2} e^{-\|w\|^2/2\sigma^2} \max(0, \langle w, x \rangle) \mathrm{d}w.
\tag{176}
$$

Completing the square, we get:

$$
\mathop{\mathbb{E}}_{w \sim p} \left[ \mathcal{K}(u, w) \phi(w, x) \right] = \left( \frac{1}{\sqrt{2\pi\sigma^2}} \right)^n \int_{\mathbb{R}^n} e^{-\left\| w - \frac{\sigma^2}{2\gamma^2} u \right\|^2/2\sigma^2 + \left\| \frac{\sigma^2}{2\gamma^2} u \right\|^2/2\sigma^2} \max(0, \langle w, x \rangle) \mathrm{d}w.
\tag{177}
$$

Defining $u' := \frac{\sigma^2}{2\gamma^2} u$ and rearranging the terms, we get:

$$
\mathop{\mathbb{E}}_{w \sim p} \left[ \mathcal{K}(u, w) \phi(w, x) \right] = \left( \frac{1}{\sqrt{2\pi\sigma^2}} \right)^n e^{\frac{\sigma^2}{8\gamma^4} \|u\|^2} \int_{\mathbb{R}^n} e^{-\left\| w - u' \right\|^2/2\sigma^2} \max(0, \langle w, x \rangle) \mathrm{d}w.
\tag{178}
$$

We get the result by applying Lemma B.4 of Dubé & Marchand (2025), which solves the integral in the previous equation. $\qquad\square$

**Lemma B.15.** *We have:*

$$
\theta \leq \frac{\sigma \sup_x \|x\|}{\sqrt{2\pi}} \left( 1 - \frac{\sigma^2}{2\gamma^2} \right)^{-\frac{(n-1)}{2}}.
\tag{179}
$$

*Proof.* First, notice that we have:

$$\theta := \sup_{x \in \mathcal{X}} \left\| \mathop{\mathbb{E}}_{w \sim p} \left[ \phi(w, x) \mathcal{K}(w, \cdot) \right] \right\|_{\mathcal{H}}$$

$$\leq \sup_{x \in \mathcal{X}} \mathop{\mathbb{E}}_{w \sim p} \left[ |\phi(w, x)| \sqrt{\mathcal{K}(w, w)} \right] =: \iota.$$

We have:

$$\iota := \sup_{x \in \mathcal{X}} \mathop{\mathbb{E}}_{w \sim p} \left[ \sqrt{\mathcal{K}(w, w)} |\phi(w, x)| \right]$$

$$= \mathop{\mathbb{E}}_{w \sim p} \left[ e^{\|w\|^2 / 4\gamma^2} \max(0, \langle w, x \rangle) \right]$$

$$= \left( \frac{1}{\sqrt{2\pi\sigma^2}} \right)^n \int_{\mathbb{R}^n} e^{\|w\|^2 / 4\gamma^2} e^{-\|w\|^2 / 2\sigma^2} \max(0, \langle w, x \rangle) \mathrm{d}w$$

$$= \left( \frac{1}{\sqrt{2\pi\sigma^2}} \right)^n \int_{\mathbb{R}^n} e^{-\|w\|^2 \left( 1/2\sigma^2 - 1/4\gamma^2 \right)} \max(0, \langle w, x \rangle) \mathrm{d}w$$

$$\leq \left( \frac{1}{\sqrt{2\pi\sigma^2}} \right)^n \left( \sqrt{\frac{\pi}{\frac{1}{2\sigma^2} - \frac{1}{4\gamma^2}}} \right)^{n-1} \frac{\sup_x \|x\|}{2 \left( \frac{1}{2\sigma^2} - \frac{1}{4\gamma^2} \right)}. \qquad \text{(Lemma B.4 of Dubé \& Marchand (2025))}$$

Rearranging this expression gives the result. $\qquad \square$

**Lemma B.16.** *We have:*

$$\kappa^2 = \frac{\sigma^2 \sup_x \|x\|^2}{2} \left( 1 - \frac{\sigma^2}{\gamma^2} \right)^{-\frac{n}{2} - 1}. \tag{180}$$

*Proof.* We have:

$$\kappa^2 := \sup_{x \in \mathcal{X}} \mathop{\mathbb{E}}_{w \sim p} \left[ \mathcal{K}(w, w) \phi(w, x)^2 \right]$$

$$= \mathop{\mathbb{E}}_{w \sim p} \left[ e^{\|w\|^2 / 2\gamma^2} \max(0, \langle w, x \rangle)^2 \right]$$

$$= \left( \frac{1}{\sqrt{2\pi\sigma^2}} \right)^n \int_{\mathbb{R}^n} e^{\|w\|^2 / 2\gamma^2} e^{-\|w\|^2 / 2\sigma^2} \max(0, \langle w, x \rangle)^2 \mathrm{d}w$$

$$= \left( \frac{1}{\sqrt{2\pi\sigma^2}} \right)^n \int_{\mathbb{R}^n} e^{-\|w\|^2 \left( 1/2\sigma^2 - 1/2\gamma^2 \right)} \max(0, \langle w, x \rangle)^2 \mathrm{d}w$$

$$= \left( \frac{1}{\sqrt{2\pi\sigma^2}} \right)^n \left( \sqrt{\frac{\pi}{\frac{1}{2\sigma^2} - \frac{1}{2\gamma^2}}} \right)^n \frac{\sup_x \|x\|^2}{2 \left( \frac{1}{\sigma^2} - \frac{1}{\gamma^2} \right)} \qquad \text{(Proof of Lemma C.3 of Dubé \& Marchand (2025))}$$

$$= \left( \frac{1}{\sqrt{2\sigma^2}} \right)^n \left( \frac{1}{2\sigma^2} - \frac{1}{2\gamma^2} \right)^{-\frac{n}{2}} \frac{\sup_x \|x\|^2}{4 \left( \frac{1}{2\sigma^2} - \frac{1}{2\gamma^2} \right)}$$

$$= \frac{\sup_x \|x\|^2}{4} (2\sigma^2)^{-\frac{n}{2}} \left( \frac{1}{2\sigma^2} - \frac{1}{2\gamma^2} \right)^{-\frac{n}{2}} \left( \frac{1}{2\sigma^2} - \frac{1}{2\gamma^2} \right)^{-1}$$

$$= \frac{\sup_x \|x\|^2}{4} \left( 1 - \frac{\sigma^2}{\gamma^2} \right)^{-\frac{n}{2}} \left( \frac{1}{2\sigma^2} - \frac{1}{2\gamma^2} \right)^{-1}$$

$$= \frac{\sigma^2 \sup_x \|x\|^2}{2} \left( 1 - \frac{\sigma^2}{\gamma^2} \right)^{-\frac{n}{2}} \left( 1 - \frac{\sigma^2}{\gamma^2} \right)^{-1}$$

$$= \frac{\sigma^2 \sup_x \|x\|^2}{2} \left( 1 - \frac{\sigma^2}{\gamma^2} \right)^{-\frac{n}{2} - 1}.$$

$\qquad \square$

### B.4 Calculus for RWStumps

Throughout this section, we assume that $(\mathcal{W}, \phi, \mathcal{K}, p)$ are those of instantiation RWStumps. See Table 2 for details.

**Lemma B.17.** *We have:*

$$\mathbb{E}_{w \sim p}[\mathcal{K}(u, w)\phi(w, x)] = \frac{\zeta}{\sigma n} e^{\frac{-u_2^2}{2\sigma^2 + 2\gamma^2}} \operatorname{erf}\left(\frac{x_{u_1} - u_2'}{\sqrt{2}\zeta}\right),$$

*where $\zeta$ is defined by the relationship:*

$$\frac{1}{2\zeta^2} = \frac{1}{2\gamma^2} + \frac{1}{2\sigma^2},$$

*and:*

$$u_2' := \left(1 + \frac{\gamma^2}{\sigma^2}\right)^{-1} u_2.$$

*Proof.* We have:

$$
\mathbb{E}_{w \sim p}[\mathcal{K}(u, w)\phi(w, x)] = \mathbb{E}_{w_1 \sim \mathcal{U}(\{1, \ldots, n\})} \mathbb{E}_{w_2 \sim \mathcal{N}(0, \sigma^2)}[\mathcal{K}(u, w)\phi(w, x)]
$$

$$
= \frac{1}{n} \sum_{i=1}^{n} \mathbb{E}_{w_2 \sim \mathcal{N}(0, \sigma^2)}\left[\mathbb{1}[i = u_1] e^{-(w_2 - u_2)^2/2\gamma^2} \operatorname{sign}(x_i - w_2)\right]
$$

$$
= \frac{1}{n} \mathbb{E}_{w_2 \sim \mathcal{N}(0, \sigma^2)}\left[e^{-(w_2 - u_2)^2/2\gamma^2} \operatorname{sign}(x_{u_1} - w_2)\right]
$$

$$
= \frac{1}{n} \frac{1}{\sqrt{2\pi\sigma^2}} \int_{-\infty}^{\infty} e^{-(w_2 - u_2)^2/2\gamma^2} e^{-w^2/2\sigma^2} \operatorname{sign}(x_{u_1} - w_2) \mathrm{d}w_2
$$

$$
= \frac{1}{n\sqrt{2\pi\sigma^2}} e^{\frac{-u_2^2}{2\sigma^2 + 2\gamma^2}} \int_{-\infty}^{\infty} e^{-(w_2 - u_2')^2/2\zeta^2} \operatorname{sign}(x_{u_1} - w_2) \mathrm{d}w_2 \qquad \text{(Lemma B.3)}
$$

$$
= \frac{e^{\frac{-u_2^2}{2\sigma^2 + 2\gamma^2}}}{n\sqrt{2\pi\sigma^2}} \left[\int_{-\infty}^{x_{u_1}} e^{-(w_2 - u_2')^2/2\zeta^2} \mathrm{d}w_2 - \int_{x_{u_1}}^{\infty} e^{-(w_2 - u_2')^2/2\zeta^2} \mathrm{d}w_2\right]
$$

$$
= \frac{e^{\frac{-u_2^2}{2\sigma^2 + 2\gamma^2}}}{n\sqrt{2\pi\sigma^2}} \sqrt{2}\zeta \left[\int_{-\infty}^{\frac{x_{u_1} - u_2'}{\sqrt{2}\zeta}} e^{-t^2} \mathrm{d}w - \int_{\frac{x_{u_1} - u_2'}{\sqrt{2}\zeta}}^{\infty} e^{-t^2} \mathrm{d}w\right] \qquad (t := \tfrac{w_2 - u_2'}{\sqrt{2}\zeta}, \, \mathrm{d}w = \tfrac{\mathrm{d}w_2}{\sqrt{2}\zeta})
$$

$$
= \frac{\zeta}{\sigma} \frac{e^{\frac{-u_2^2}{2\sigma^2 + 2\gamma^2}}}{n\sqrt{\pi}} \left[\int_{-\infty}^{0} e^{-t^2} \mathrm{d}w + \int_{0}^{\frac{x_{u_1} - u_2'}{\sqrt{2}\zeta}} e^{-t^2} \mathrm{d}w - \int_{\frac{x_{u_1} - u_2'}{\sqrt{2}\zeta}}^{0} e^{-t^2} \mathrm{d}w - \int_{0}^{\infty} e^{-t^2} \mathrm{d}w\right]
$$

$$
= \frac{\zeta}{\sigma} \frac{e^{\frac{-u_2^2}{2\sigma^2 + 2\gamma^2}}}{n\sqrt{\pi}} \left[\int_{0}^{\frac{x_{u_1} - u_2'}{\sqrt{2}\zeta}} e^{-t^2} \mathrm{d}w - \int_{\frac{x_{u_1} - u_2'}{\sqrt{2}\zeta}}^{0} e^{-t^2} \mathrm{d}w\right].
$$

Finally, we have:

$$
\mathbb{E}_{w \sim p}[\mathcal{K}(u, w)\phi(w, x)] = \frac{\zeta}{\sigma} \frac{e^{\frac{-u_2^2}{2\sigma^2 + 2\gamma^2}}}{n\sqrt{\pi}} \left[\int_{0}^{\frac{x_{u_1} - u_2'}{\sqrt{2}\zeta}} e^{-t^2} \mathrm{d}w - \int_{\frac{x_{u_1} - u_2'}{\sqrt{2}\zeta}}^{0} e^{-t^2} \mathrm{d}w\right]
$$

$$
= \frac{\zeta}{\sigma} \frac{e^{\frac{-u_2^2}{2\sigma^2 + 2\gamma^2}}}{n} \operatorname{erf}\left(\frac{x_{u_1} - u_2'}{\sqrt{2}\zeta}\right).
$$

$\square$

**Lemma B.18.** *We have:*

$$\theta \leq \frac{1}{\sqrt{n}} \left(1 + \frac{2\sigma^2}{\gamma^2}\right)^{-1/4}. \tag{181}$$

*Proof.* We have:

$$
\begin{aligned}
\theta^2 &= \sup_{x \in \mathcal{X}} \|\psi(x)\|_{\mathcal{H}}^2 \\
&= \sup_{x \in \mathcal{X}} \mathop{\mathbb{E}}_{w \sim p} \mathop{\mathbb{E}}_{u \sim p} \left[\mathcal{K}(u,w)\phi(u,x)\phi(w,x)\right] \\
&\leq \sup_{x \in \mathcal{X}} \mathop{\mathbb{E}}_{w \sim p} \mathop{\mathbb{E}}_{u \sim p} \left[|\mathcal{K}(u,w)\phi(u,x)\phi(w,x)|\right] \\
&= \mathop{\mathbb{E}}_{w \sim p} \mathop{\mathbb{E}}_{u \sim p} \left[\mathcal{K}(u,w)\right] && (|\phi(w,x)| = 1 \text{ for all } w \text{ and } x) \\
&= \frac{1}{n^2} \sum_{i=1}^{n} \sum_{j=1}^{n} \mathop{\mathbb{E}}_{w \sim \mathcal{N}(0,\sigma^2)} \mathop{\mathbb{E}}_{u \sim \mathcal{N}(0,\sigma^2)} \left[\mathbb{1}[i=j]e^{-(u-w)/2\gamma^2}\right] \\
&= \frac{1}{n^2} \sum_{i=1}^{n} \mathop{\mathbb{E}}_{w \sim \mathcal{N}(0,\sigma^2)} \mathop{\mathbb{E}}_{u \sim \mathcal{N}(0,\sigma^2)} \left[e^{-(u-w)/2\gamma^2}\right] \\
&= \frac{1}{n^2} \sum_{i=1}^{n} \left(1 + \frac{2\sigma^2}{\gamma^2}\right)^{-1/2} && (\text{See proof of Lemma B.10}) \\
&= \frac{1}{n} \left(1 + \frac{2\sigma^2}{\gamma^2}\right)^{-1/2}.
\end{aligned}
$$

□

## C  Details of experimentation

**Preprocessing of the datasets**

All datasets have been scaled to have mean 0 and standard deviation 1 on all variables, including the target labels for regression datasets. Means and standard deviations were calculated on the training data, then the transformation applied to both training and test datasets.

**Hyperparameter selection**

Table 14 contains the hyperparameters of all the algorithms and models used in this paper. All parameters were crossvalidated using randomized search (rather than grid search) using 50 random combinations of parameters.

**Reproducing the results**

The code for this paper can be found at `https://github.com/gadub44/rkhs-weightings`. The repository contains a `requirements.txt` file containing the particular Python packages that were used in our experiments. They can be installed from the command line with the command `pip install -r requirements.txt`. Table 15 lists the commands to run the experiments of Section 7.

## D  Additional results

We present below a few additional tables and figures. Tables 16, 17, 18 and 19 are respectively Tables 10, 11, 12 and 13 with added standard deviations. Figures 9 and 10 are respectively Figures 5 and 6 with RKHS weightings trained using Algorithm 5 instead of Algorithm 4.

|  | Cross-validation parameters | Source code (clickable) |
|---|---|---|
| AdaBoostClassifier | number of estimators $\in \{10, 25, 50, 100, 150, 200, 250, 500\}$ | Scikit-learn |
| AdaBoostRegressor | number of estimators $\in \{10, 25, 50, 100, 150, 200, 250, 500\}$ | Scikit-learn |
| SVR | $C \sim \text{loguniform}(0.001, 1000)$
gamma $\sim \text{loguniform}(0.001, 1000)$ | Scikit-learn |
| SVC | $C \sim \text{loguniform}(0.001, 1000)$
gamma $\sim \text{loguniform}(0.001, 1000)$ | Scikit-learn |
| Random kitchen sinks | $\lambda_2 \sim \text{loguniform}(10^{-5}, 10^{-3})$ | This paper |
| All RKHS weightings | $\sigma \sim \text{loguniform}(0.01, 10)$
$\lambda_{\mathcal{H}} \sim \text{loguniform}(10^{-12}, 10^{-4})$ |  |
| RWSign | max theta $\sim \text{uniform}\{0.01, 0.9\}$ | This paper |
| RWRelu | max theta $\sim \text{uniform}\{0.01, 0.9\}$ | Dubé & Marchand (2025) |
| RWExpSign | max theta $\sim \text{uniform}\{1.5, 50\}$ | This paper |
| RWExpRelu | max kappa $\sim \text{uniform}\{1.5, 100\}$ | Dubé & Marchand (2025) |
| RWStumps | $\gamma \sim \text{loguniform}(0.01, 10)$ | This paper |

Table 14: Algorithms and models used in this paper and their cross-validation hyperparameters.

| Experiment | Command | Execution time |
|---|---|---|
| Figure 1 | `python -m experiments.algo_time_compar --final` | 5.8 hours |
| Figure 2 | `python -m experiments.algo_time_compar_breakdown` | 1.6 hours |
| Figures 3 and 4 | `python -m experiments.effect_of_gamma --final` | 34.8 hours |
| Figures 5 and 6 | `python -m experiments.few_features --final` | 85.4 hours = 3.56 days |
| Figures 7 and 8 | `python -m experiments.multiple_rfs_per_iteration --final` | 1.0 hour |
| Figures 9 and 10 | `python -m experiments.few_features_lasso --final` | 197.3 hours = 8.22 days |
| Tables 10 and 11 | `python -m experiments.rkhs_vs_rks --final` | 35.5 hours |
| Tables 12 and 13 | `python -m experiments.sota --final` | 90.2 hours = 3.76 days |
| All (Linux, macOS) | `./experiments/run.sh` | 451.6 hours = 18.82 days |

Table 15: Commands to run the various experiments in this paper, from the root folder of the supplementary material. The experiments were run on a machine equipped with an AMD Ryzen Threadripper 1900X 8-Core Processor (2 threads per core), running Debian GNU/Linux 12 (bookworm) with 32GB of system memory.

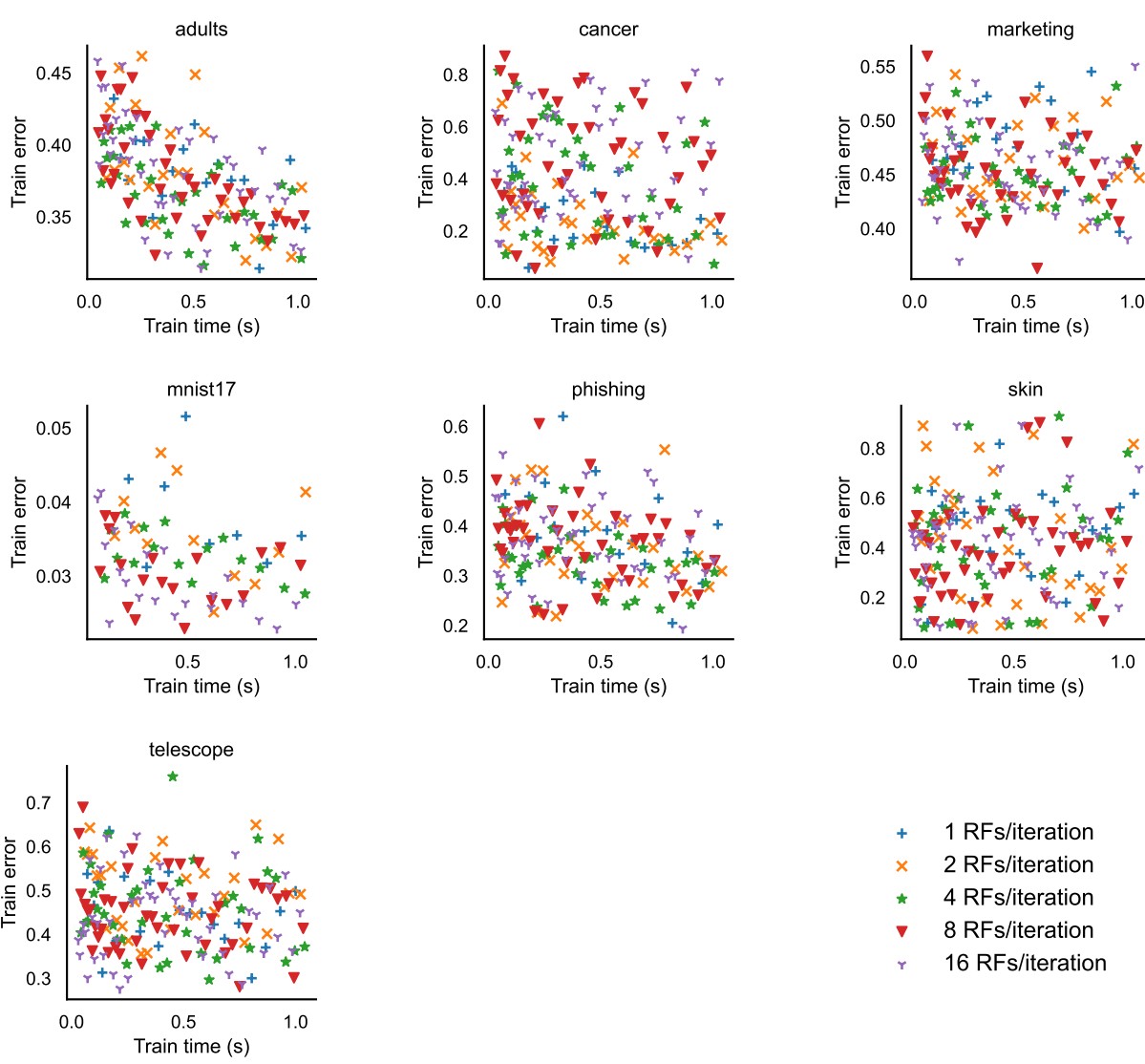

Figure 7: Training error of RKHS weightings using the RWSign instantiation and learned using SFGD (Algorithm 3), with $B = 1000$, $b = 100$ and $\lambda_{\mathcal{H}} = 10^{-6}$, with regard to the training time and for different random feature batch sizes (**RFs/iteration**). See Section 5.1.1. All values of RFs/iteration perform similarly.

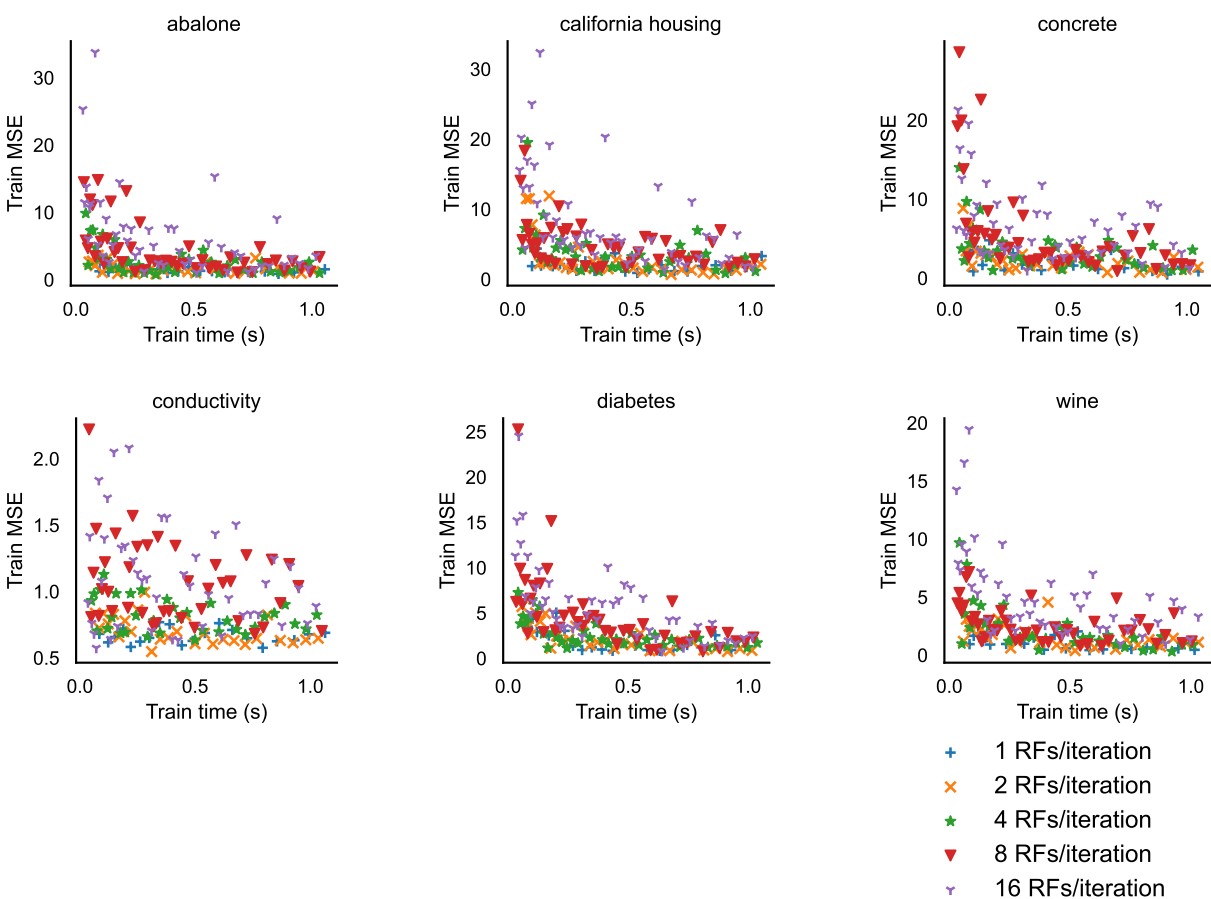

Figure 8: Training MSE of RKHS weightings using the RWSign instantiation and learned using SFGD (Algorithm 3), with $B = 1000$, $b = 100$ and $\lambda_{\mathcal{H}} = 10^{-6}$, with regard to the training time and for different random feature batch sizes (**RFs/iteration**). See Section 5.1.1. Higher values of RFs/iteration perform worse.

Table 16: (Train and test error columns of Table 10 with standard deviations.) Classification performance comparison of RKHS weightings (RW) to random kitchen sinks (RKS). For each combination of dataset and instantiation, an RKHS weighting was learned by cross-validation and using Algorithm 4 (the least squares fit) in order to select the best model hyperparameters. The sampling size $T$ was 500, and each value in the table is the average of 10 independently seeded runs. Train error and Test error are the proportion of incorrectly classified examples on the training and test sets.

| | | Train error | | Test error | |
| | | RKS | RW | RKS | RW |
| Dataset | Instantiation | | | | |
|---|---|---|---|---|---|
| adults | RWExpRelu | $0.152 \pm 0.001$ | $0.150 \pm 0.001$ | $0.155 \pm 0.001$ | $\mathbf{0.152 \pm 0.001}$ |
| | RWExpSign | $0.191 \pm 0.002$ | $0.194 \pm 0.002$ | $0.198 \pm 0.004$ | $0.198 \pm 0.002$ |
| | RWRelu | $0.152 \pm 0.001$ | $0.149 \pm 0.001$ | $0.155 \pm 0.001$ | $\mathbf{0.152 \pm 0.001}$ |
| | RWSign | $0.191 \pm 0.002$ | $0.216 \pm 0.000$ | $0.198 \pm 0.004$ | $0.220 \pm 0.000$ |
| | RWStumps | $0.148 \pm 0.005$ | $0.147 \pm 0.005$ | $\mathbf{0.150 \pm 0.004}$ | $\mathbf{0.147 \pm 0.004}$ |
| cancer | RWExpRelu | $0.014 \pm 0.001$ | $0.013 \pm 0.002$ | $\mathbf{0.029 \pm 0.006}$ | $\mathbf{0.030 \pm 0.006}$ |
| | RWExpSign | $0.011 \pm 0.005$ | $0.012 \pm 0.002$ | $\mathbf{0.041 \pm 0.010}$ | $0.048 \pm 0.008$ |
| | RWRelu | $0.014 \pm 0.001$ | $0.015 \pm 0.002$ | $\mathbf{0.029 \pm 0.006}$ | $\mathbf{0.026 \pm 0.006}$ |
| | RWSign | $0.011 \pm 0.005$ | $0.010 \pm 0.002$ | $\mathbf{0.041 \pm 0.010}$ | $\mathbf{0.040 \pm 0.009}$ |
| | RWStumps | $0.012 \pm 0.004$ | $0.017 \pm 0.003$ | $\mathbf{0.032 \pm 0.011}$ | $0.036 \pm 0.003$ |
| marketing | RWExpRelu | $0.094 \pm 0.001$ | $0.093 \pm 0.001$ | $\mathbf{0.100 \pm 0.001}$ | $\mathbf{0.100 \pm 0.001}$ |
| | RWExpSign | $0.139 \pm 0.003$ | $0.103 \pm 0.001$ | $0.148 \pm 0.004$ | $0.110 \pm 0.002$ |
| | RWRelu | $0.094 \pm 0.001$ | $0.092 \pm 0.001$ | $\mathbf{0.100 \pm 0.001}$ | $\mathbf{0.099 \pm 0.001}$ |
| | RWSign | $0.139 \pm 0.003$ | $0.132 \pm 0.002$ | $0.148 \pm 0.004$ | $0.138 \pm 0.003$ |
| | RWStumps | $0.097 \pm 0.001$ | $0.097 \pm 0.000$ | $0.102 \pm 0.001$ | $\mathbf{0.101 \pm 0.001}$ |
| mnist17 | RWExpRelu | $0.006 \pm 0.000$ | $0.005 \pm 0.000$ | $\mathbf{0.012 \pm 0.002}$ | $\mathbf{0.011 \pm 0.001}$ |
| | RWExpSign | $0.010 \pm 0.001$ | $0.008 \pm 0.001$ | $\mathbf{0.014 \pm 0.002}$ | $\mathbf{0.013 \pm 0.001}$ |
| | RWRelu | $0.006 \pm 0.000$ | $0.007 \pm 0.000$ | $\mathbf{0.012 \pm 0.002}$ | $\mathbf{0.012 \pm 0.001}$ |
| | RWSign | $0.010 \pm 0.001$ | $0.008 \pm 0.000$ | $\mathbf{0.014 \pm 0.002}$ | $\mathbf{0.013 \pm 0.001}$ |
| | RWStumps | $0.008 \pm 0.001$ | $0.008 \pm 0.000$ | $\mathbf{0.013 \pm 0.002}$ | $\mathbf{0.013 \pm 0.002}$ |
| phishing | RWExpRelu | $0.049 \pm 0.002$ | $0.046 \pm 0.002$ | $0.062 \pm 0.003$ | $\mathbf{0.058 \pm 0.002}$ |
| | RWExpSign | $0.061 \pm 0.003$ | $0.051 \pm 0.002$ | $0.072 \pm 0.003$ | $0.065 \pm 0.002$ |
| | RWRelu | $0.049 \pm 0.002$ | $0.043 \pm 0.001$ | $0.062 \pm 0.003$ | $\mathbf{0.055 \pm 0.001}$ |
| | RWSign | $0.061 \pm 0.003$ | $0.056 \pm 0.001$ | $0.072 \pm 0.003$ | $0.067 \pm 0.002$ |
| | RWStumps | $0.069 \pm 0.000$ | $0.069 \pm 0.001$ | $0.075 \pm 0.001$ | $0.075 \pm 0.001$ |
| skin | RWExpRelu | $0.015 \pm 0.000$ | $0.023 \pm 0.001$ | $\mathbf{0.016 \pm 0.001}$ | $0.023 \pm 0.001$ |
| | RWExpSign | $0.039 \pm 0.002$ | $0.059 \pm 0.001$ | $0.040 \pm 0.001$ | $0.060 \pm 0.001$ |
| | RWRelu | $0.015 \pm 0.000$ | $0.029 \pm 0.000$ | $\mathbf{0.016 \pm 0.001}$ | $0.029 \pm 0.000$ |
| | RWSign | $0.039 \pm 0.002$ | $0.049 \pm 0.000$ | $0.040 \pm 0.001$ | $0.050 \pm 0.000$ |
| | RWStumps | $0.040 \pm 0.000$ | $0.040 \pm 0.000$ | $0.040 \pm 0.000$ | $0.040 \pm 0.000$ |
| telescope | RWExpRelu | $0.131 \pm 0.001$ | $0.125 \pm 0.001$ | $\mathbf{0.137 \pm 0.002}$ | $\mathbf{0.135 \pm 0.002}$ |
| | RWExpSign | $0.162 \pm 0.003$ | $0.139 \pm 0.001$ | $0.172 \pm 0.003$ | $0.146 \pm 0.003$ |
| | RWRelu | $0.131 \pm 0.001$ | $0.127 \pm 0.001$ | $\mathbf{0.137 \pm 0.002}$ | $\mathbf{0.135 \pm 0.001}$ |
| | RWSign | $0.162 \pm 0.003$ | $0.143 \pm 0.002$ | $0.172 \pm 0.003$ | $0.147 \pm 0.002$ |
| | RWStumps | $0.139 \pm 0.002$ | $0.139 \pm 0.001$ | $0.149 \pm 0.003$ | $0.146 \pm 0.001$ |

Table 17: (Train and test MSE columns of Table 11 with standard deviations.) Regression performance comparison of RKHS weightings (RW) to random kitchen sinks (RKS). For each combination of dataset and instantiation, an RKHS weighting was learned by cross-validation and using Algorithm 4 (the least squares fit) in order to select the best model hyperparameters. The sampling size $T$ was 500, and each value in the table is the average of 10 independently seeded runs.

| Dataset | Instantiation | Train MSE | | Test MSE | |
| | | RKS | RW | RKS | RW |
| --- | --- | --- | --- | --- | --- |
| abalone | RWExpRelu | $0.391 \pm 0.003$ | $0.396 \pm 0.004$ | $\mathbf{0.427 \pm 0.002}$ | $\mathbf{0.426 \pm 0.001}$ |
| | RWExpSign | $0.394 \pm 0.007$ | $0.413 \pm 0.003$ | $0.441 \pm 0.006$ | $0.433 \pm 0.000$ |
| | RWRelu | $0.391 \pm 0.003$ | $0.398 \pm 0.002$ | $\mathbf{0.427 \pm 0.002}$ | $0.430 \pm 0.001$ |
| | RWSign | $0.394 \pm 0.007$ | $0.406 \pm 0.002$ | $0.441 \pm 0.006$ | $0.431 \pm 0.001$ |
| | RWStumps | $0.416 \pm 0.008$ | $0.421 \pm 0.000$ | $0.460 \pm 0.008$ | $0.441 \pm 0.000$ |
| california housing | RWExpRelu | $0.249 \pm 0.003$ | $0.239 \pm 0.003$ | $0.269 \pm 0.002$ | $\mathbf{0.259 \pm 0.001}$ |
| | RWExpSign | $0.317 \pm 0.003$ | $0.281 \pm 0.001$ | $0.337 \pm 0.004$ | $0.298 \pm 0.001$ |
| | RWRelu | $0.249 \pm 0.003$ | $0.243 \pm 0.001$ | $0.269 \pm 0.002$ | $0.261 \pm 0.001$ |
| | RWSign | $0.317 \pm 0.003$ | $0.280 \pm 0.001$ | $0.337 \pm 0.004$ | $0.296 \pm 0.001$ |
| | RWStumps | $0.248 \pm 0.009$ | $0.234 \pm 0.008$ | $\mathbf{0.266 \pm 0.009}$ | $\mathbf{0.252 \pm 0.007}$ |
| concrete | RWExpRelu | $0.068 \pm 0.007$ | $0.065 \pm 0.015$ | $0.144 \pm 0.008$ | $0.155 \pm 0.012$ |
| | RWExpSign | $0.126 \pm 0.009$ | $0.113 \pm 0.050$ | $0.255 \pm 0.009$ | $0.210 \pm 0.033$ |
| | RWRelu | $0.068 \pm 0.007$ | $0.087 \pm 0.015$ | $0.144 \pm 0.008$ | $0.153 \pm 0.009$ |
| | RWSign | $0.126 \pm 0.009$ | $0.083 \pm 0.005$ | $0.255 \pm 0.009$ | $0.189 \pm 0.012$ |
| | RWStumps | $0.068 \pm 0.007$ | $0.064 \pm 0.013$ | $\mathbf{0.090 \pm 0.004}$ | $\mathbf{0.098 \pm 0.023}$ |
| conductivity | RWExpRelu | $0.176 \pm 0.002$ | $0.167 \pm 0.002$ | $0.193 \pm 0.003$ | $\mathbf{0.185 \pm 0.002}$ |
| | RWExpSign | $0.207 \pm 0.003$ | $0.217 \pm 0.001$ | $0.224 \pm 0.005$ | $0.222 \pm 0.002$ |
| | RWRelu | $0.176 \pm 0.002$ | $0.165 \pm 0.001$ | $0.193 \pm 0.003$ | $\mathbf{0.183 \pm 0.001}$ |
| | RWSign | $0.207 \pm 0.003$ | $0.268 \pm 0.001$ | $0.224 \pm 0.005$ | $0.270 \pm 0.001$ |
| | RWStumps | $0.187 \pm 0.004$ | $0.182 \pm 0.002$ | $0.198 \pm 0.003$ | $0.193 \pm 0.003$ |
| diabetes | RWExpRelu | $0.375 \pm 0.007$ | $0.355 \pm 0.001$ | $0.529 \pm 0.008$ | $0.544 \pm 0.002$ |
| | RWExpSign | $0.391 \pm 0.012$ | $0.452 \pm 0.000$ | $\mathbf{0.514 \pm 0.011}$ | $0.530 \pm 0.000$ |
| | RWRelu | $0.375 \pm 0.007$ | $0.382 \pm 0.000$ | $0.529 \pm 0.008$ | $0.529 \pm 0.000$ |
| | RWSign | $0.391 \pm 0.012$ | $0.453 \pm 0.000$ | $\mathbf{0.514 \pm 0.011}$ | $0.514 \pm 0.001$ |
| | RWStumps | $0.375 \pm 0.036$ | $0.427 \pm 0.000$ | $\mathbf{0.530 \pm 0.028}$ | $\mathbf{0.510 \pm 0.000}$ |
| wine | RWExpRelu | $0.032 \pm 0.004$ | $0.032 \pm 0.000$ | $0.113 \pm 0.010$ | $0.101 \pm 0.002$ |
| | RWExpSign | $0.009 \pm 0.006$ | $0.026 \pm 0.006$ | $\mathbf{0.091 \pm 0.011}$ | $\mathbf{0.088 \pm 0.004}$ |
| | RWRelu | $0.032 \pm 0.004$ | $0.039 \pm 0.003$ | $0.113 \pm 0.010$ | $0.147 \pm 0.004$ |
| | RWSign | $0.009 \pm 0.006$ | $0.028 \pm 0.008$ | $\mathbf{0.091 \pm 0.011}$ | $\mathbf{0.089 \pm 0.006}$ |
| | RWStumps | $0.042 \pm 0.006$ | $0.066 \pm 0.000$ | $0.108 \pm 0.012$ | $0.136 \pm 0.000$ |

Table 18: (Table 12 with standard deviations.) Binary classification performance comparison of RKHS weightings to AdaBoost (**AB**), **SVM** and the random kitchen sinks (**RKS**) on various datasets. Instantiations were chosen based on their performance in Table 10. Algorithm 4 (the least squares fit of the coefficient) was used to learn RKHS weightings with $T = 2000$ random features. Train error and Test error are the misclassification rates on the training and test sets. Inference time is the computation time of the model on the training set and test sets combined. Every line (except SVM, which is deterministic) is the average of 10 independent runs.

| Dataset | Algorithm | Instantiation | Train error | Test error | Train time (s) | Inference time |
|---|---|---|---|---|---|---|
| adults | AdaBoost | | $0.141 \pm 0.0$ | $\mathbf{0.14 \pm 0.0}$ | $15.225 \pm 0.242$ | $3.663 \pm 0.058$ |
| | RKHS Weighting | RWExpRelu | $0.137 \pm 0.002$ | $0.148 \pm 0.001$ | $3.967 \pm 0.032$ | $4.473 \pm 0.041$ |
| | | RWRelu | $0.138 \pm 0.002$ | $0.147 \pm 0.001$ | $3.984 \pm 0.054$ | $4.357 \pm 0.054$ |
| | | RWStumps | $0.146 \pm 0.001$ | $0.145 \pm 0.001$ | $2.163 \pm 0.196$ | $1.824 \pm 0.291$ |
| | RKS | RWRelu | $0.137 \pm 0.002$ | $0.15 \pm 0.001$ | $1.685 \pm 0.036$ | $1.246 \pm 0.007$ |
| | | RWStumps | $0.144 \pm 0.001$ | $0.144 \pm 0.001$ | $1.335 \pm 0.043$ | $0.809 \pm 0.013$ |
| | SVM | | $0.143$ | $0.146$ | $75.187$ | $57.222$ |
| cancer | AdaBoost | | $0.0 \pm 0.0$ | $\mathbf{0.021 \pm 0.0}$ | $0.512 \pm 0.017$ | $0.04 \pm 0.0$ |
| | RKHS Weighting | RWExpSign | $0.011 \pm 0.002$ | $0.045 \pm 0.008$ | $0.259 \pm 0.01$ | $0.032 \pm 0.007$ |
| | | RWSign | $0.01 \pm 0.002$ | $0.048 \pm 0.006$ | $0.265 \pm 0.006$ | $0.033 \pm 0.008$ |
| | RKS | RWSign | $0.006 \pm 0.005$ | $0.038 \pm 0.009$ | $0.185 \pm 0.012$ | $0.023 \pm 0.009$ |
| | SVM | | $0.014$ | $\mathbf{0.021}$ | $0.002$ | $0.002$ |
| marketing | AdaBoost | | $0.097 \pm 0.0$ | $0.101 \pm 0.0$ | $11.803 \pm 0.19$ | $1.607 \pm 0.014$ |
| | RKHS Weighting | RWExpRelu | $0.083 \pm 0.003$ | $\mathbf{0.097 \pm 0.001}$ | $4.02 \pm 0.034$ | $4.052 \pm 0.051$ |
| | | RWRelu | $0.089 \pm 0.001$ | $\mathbf{0.097 \pm 0.001}$ | $3.98 \pm 0.058$ | $3.958 \pm 0.053$ |
| | | RWStumps | $0.096 \pm 0.0$ | $0.1 \pm 0.001$ | $2.75 \pm 0.111$ | $2.303 \pm 0.149$ |
| | RKS | RWRelu | $0.085 \pm 0.002$ | $\mathbf{0.097 \pm 0.001}$ | $1.7 \pm 0.024$ | $1.113 \pm 0.006$ |
| | | RWStumps | $0.096 \pm 0.001$ | $0.1 \pm 0.001$ | $1.692 \pm 0.043$ | $1.069 \pm 0.028$ |
| | SVM | | $0.077$ | $\mathbf{0.097}$ | $28.927$ | $17.778$ |
| mnist17 | AdaBoost | | $0.0 \pm 0.0$ | $\mathbf{0.004 \pm 0.0}$ | $90.048 \pm 2.329$ | $9.455 \pm 0.3$ |
| | RKHS Weighting | RWExpRelu | $0.002 \pm 0.0$ | $0.008 \pm 0.001$ | $2.066 \pm 0.033$ | $1.686 \pm 0.024$ |
| | | RWRelu | $0.003 \pm 0.0$ | $0.01 \pm 0.001$ | $3.47 \pm 0.055$ | $1.642 \pm 0.024$ |
| | | RWStumps | $0.006 \pm 0.001$ | $0.009 \pm 0.001$ | $1.085 \pm 0.073$ | $0.711 \pm 0.082$ |
| | RKS | RWRelu | $0.003 \pm 0.001$ | $0.009 \pm 0.002$ | $1.015 \pm 0.019$ | $0.585 \pm 0.011$ |
| | | RWStumps | $0.006 \pm 0.0$ | $0.01 \pm 0.001$ | $0.734 \pm 0.021$ | $0.41 \pm 0.008$ |
| | SVM | | $0.0$ | $0.008$ | $6.37$ | $3.935$ |
| phishing | AdaBoost | | $0.061 \pm 0.0$ | $0.067 \pm 0.0$ | $1.687 \pm 0.025$ | $0.225 \pm 0.003$ |
| | RKHS Weighting | RWExpRelu | $0.027 \pm 0.002$ | $0.047 \pm 0.002$ | $1.17 \pm 0.016$ | $1.058 \pm 0.009$ |
| | | RWRelu | $0.04 \pm 0.001$ | $0.052 \pm 0.001$ | $1.167 \pm 0.011$ | $1.032 \pm 0.01$ |
| | RKS | RWRelu | $0.025 \pm 0.002$ | $\mathbf{0.045 \pm 0.002}$ | $0.519 \pm 0.012$ | $0.276 \pm 0.004$ |
| | SVM | | $0.024$ | $\mathbf{0.043}$ | $0.709$ | $0.824$ |
| skin | AdaBoost | | $0.043 \pm 0.0$ | $0.043 \pm 0.0$ | $10.941 \pm 0.267$ | $1.287 \pm 0.016$ |
| | RKHS Weighting | RWExpRelu | $0.02 \pm 0.0$ | $0.02 \pm 0.001$ | $21.153 \pm 0.311$ | $21.928 \pm 0.164$ |
| | | RWRelu | $0.022 \pm 0.0$ | $0.022 \pm 0.001$ | $23.677 \pm 0.264$ | $25.434 \pm 0.21$ |
| | RKS | RWRelu | $0.012 \pm 0.0$ | $0.013 \pm 0.0$ | $8.424 \pm 0.244$ | $5.763 \pm 0.043$ |
| | SVM | | $0.0$ | $\mathbf{0.0}$ | $533.843$ | $53.736$ |
| telescope | AdaBoost | | $0.144 \pm 0.0$ | $0.157 \pm 0.0$ | $10.14 \pm 0.096$ | $0.234 \pm 0.004$ |
| | RKHS Weighting | RWExpRelu | $0.119 \pm 0.003$ | $0.133 \pm 0.002$ | $1.79 \pm 0.018$ | $1.726 \pm 0.02$ |
| | | RWRelu | $0.12 \pm 0.001$ | $\mathbf{0.131 \pm 0.001}$ | $1.768 \pm 0.031$ | $1.689 \pm 0.021$ |
| | RKS | RWRelu | $0.12 \pm 0.003$ | $0.135 \pm 0.002$ | $0.779 \pm 0.014$ | $0.465 \pm 0.006$ |
| | SVM | | $0.095$ | $\mathbf{0.13}$ | $3.298$ | $4.067$ |

Table 19: (Table 13 with standard deviations.) Regression performance comparison of RKHS weightings to AdaBoost (**AB**), **SVM** and the random kitchen sinks (**RKS**) on various datasets. Instantiations were chosen based on their performance in Table 11. Algorithm 4 (the least squares fit of the coefficient) was used to learn RKHS weightings with $T = 2000$ random features. Inference time is the computation time of the model on the training set and test sets combined. Every line (except SVM, which is deterministic) is the average of 10 independent runs.

| Dataset | Algorithm | Instantiation | Train $R^2$ | Test $R^2$ | Train time (s) | Inference time |
|---|---|---|---|---|---|---|
| abalone | AdaBoost | | $0.44 \pm 0.019$ | $0.444 \pm 0.016$ | $0.067 \pm 0.021$ | $0.003 \pm 0.0$ |
| | RKHS Weighting | RWExpRelu | $0.604 \pm 0.0$ | $\mathbf{0.583 \pm 0.0}$ | $0.601 \pm 0.008$ | $0.42 \pm 0.006$ |
| | | RWRelu | $0.603 \pm 0.001$ | $0.576 \pm 0.0$ | $0.574 \pm 0.007$ | $0.403 \pm 0.007$ |
| | | RWSign | $0.591 \pm 0.0$ | $0.576 \pm 0.0$ | $0.457 \pm 0.007$ | $0.223 \pm 0.01$ |
| | RKS | RWRelu | $0.618 \pm 0.001$ | $0.581 \pm 0.001$ | $0.314 \pm 0.018$ | $0.1 \pm 0.005$ |
| | | RWSign | $0.647 \pm 0.011$ | $0.578 \pm 0.003$ | $0.343 \pm 0.011$ | $0.139 \pm 0.004$ |
| | SVM | | $0.579$ | $0.571$ | $0.35$ | $0.4$ |
| concrete | AdaBoost | | $0.819 \pm 0.004$ | $0.767 \pm 0.011$ | $0.129 \pm 0.038$ | $0.017 \pm 0.007$ |
| | RKHS Weighting | RWExpRelu | $0.955 \pm 0.006$ | $0.836 \pm 0.01$ | $0.314 \pm 0.009$ | $0.114 \pm 0.013$ |
| | | RWRelu | $0.925 \pm 0.0$ | $0.836 \pm 0.002$ | $0.297 \pm 0.009$ | $0.105 \pm 0.007$ |
| | | RWStumps | $0.96 \pm 0.0$ | $\mathbf{0.912 \pm 0.001}$ | $0.311 \pm 0.01$ | $0.058 \pm 0.003$ |
| | RKS | RWRelu | $0.959 \pm 0.007$ | $0.85 \pm 0.005$ | $0.193 \pm 0.012$ | $0.024 \pm 0.006$ |
| | | RWStumps | $0.945 \pm 0.001$ | $\mathbf{0.91 \pm 0.002}$ | $0.148 \pm 0.009$ | $0.025 \pm 0.006$ |
| | SVM | | $0.957$ | $0.851$ | $0.025$ | $0.02$ |
| conductivity | AdaBoost | | $0.731 \pm 0.005$ | $0.723 \pm 0.004$ | $2.239 \pm 0.02$ | $0.028 \pm 0.0$ |
| | RKHS Weighting | RWExpRelu | $0.877 \pm 0.001$ | $0.849 \pm 0.001$ | $2.068 \pm 0.021$ | $1.957 \pm 0.015$ |
| | | RWRelu | $0.87 \pm 0.001$ | $0.845 \pm 0.001$ | $2.095 \pm 0.015$ | $1.903 \pm 0.018$ |
| | | RWStumps | $0.867 \pm 0.003$ | $0.848 \pm 0.003$ | $1.493 \pm 0.012$ | $1.154 \pm 0.008$ |
| | RKS | RWRelu | $0.878 \pm 0.001$ | $0.849 \pm 0.001$ | $0.896 \pm 0.016$ | $0.517 \pm 0.002$ |
| | | RWStumps | $0.876 \pm 0.002$ | $0.852 \pm 0.003$ | $0.882 \pm 0.018$ | $0.533 \pm 0.006$ |
| | SVM | | $0.903$ | $\mathbf{0.871}$ | $13.384$ | $11.28$ |
| diabetes | AdaBoost | | $0.669 \pm 0.013$ | $0.286 \pm 0.018$ | $0.101 \pm 0.066$ | $0.013 \pm 0.01$ |
| | RKHS Weighting | RWExpRelu | $0.692 \pm 0.001$ | $0.3 \pm 0.001$ | $0.287 \pm 0.012$ | $0.061 \pm 0.013$ |
| | | RWRelu | $0.618 \pm 0.0$ | $0.334 \pm 0.0$ | $0.261 \pm 0.011$ | $0.055 \pm 0.012$ |
| | | RWStumps | $0.573 \pm 0.0$ | $\mathbf{0.358 \pm 0.0}$ | $0.228 \pm 0.007$ | $0.018 \pm 0.0$ |
| | RKS | RWRelu | $0.628 \pm 0.002$ | $0.335 \pm 0.003$ | $0.181 \pm 0.003$ | $0.017 \pm 0.002$ |
| | | RWStumps | $0.626 \pm 0.03$ | $\mathbf{0.338 \pm 0.021}$ | $0.14 \pm 0.007$ | $0.016 \pm 0.006$ |
| | SVM | | $0.609$ | $0.343$ | $0.008$ | $0.01$ |
| housing | AdaBoost | | $0.569 \pm 0.009$ | $0.546 \pm 0.009$ | $0.323 \pm 0.021$ | $0.01 \pm 0.0$ |
| | RKHS Weighting | RWExpRelu | $0.77 \pm 0.001$ | $0.743 \pm 0.001$ | $1.977 \pm 0.021$ | $1.809 \pm 0.031$ |
| | | RWRelu | $0.753 \pm 0.002$ | $0.735 \pm 0.001$ | $1.925 \pm 0.025$ | $1.747 \pm 0.029$ |
| | | RWStumps | $0.787 \pm 0.002$ | $0.764 \pm 0.002$ | $1.29 \pm 0.018$ | $0.955 \pm 0.007$ |
| | RKS | RWRelu | $0.779 \pm 0.004$ | $0.741 \pm 0.003$ | $0.846 \pm 0.02$ | $0.437 \pm 0.022$ |
| | | RWStumps | $0.788 \pm 0.005$ | $0.759 \pm 0.002$ | $0.725 \pm 0.022$ | $0.401 \pm 0.012$ |
| | SVM | | $0.815$ | $\mathbf{0.775}$ | $9.278$ | $8.181$ |
| wine | AdaBoost | | $1.0 \pm 0.0$ | $\mathbf{0.956 \pm 0.0}$ | $0.16 \pm 0.08$ | $0.026 \pm 0.012$ |
| | RKHS Weighting | RWExpSign | $0.979 \pm 0.002$ | $0.893 \pm 0.001$ | $0.259 \pm 0.01$ | $0.019 \pm 0.011$ |
| | | RWSign | $0.978 \pm 0.0$ | $0.894 \pm 0.0$ | $0.244 \pm 0.006$ | $0.019 \pm 0.01$ |
| | RKS | RWSign | $1.0 \pm 0.0$ | $0.904 \pm 0.008$ | $0.167 \pm 0.012$ | $0.013 \pm 0.006$ |
| | SVM | | $0.99$ | $0.942$ | $0.004$ | $0.002$ |

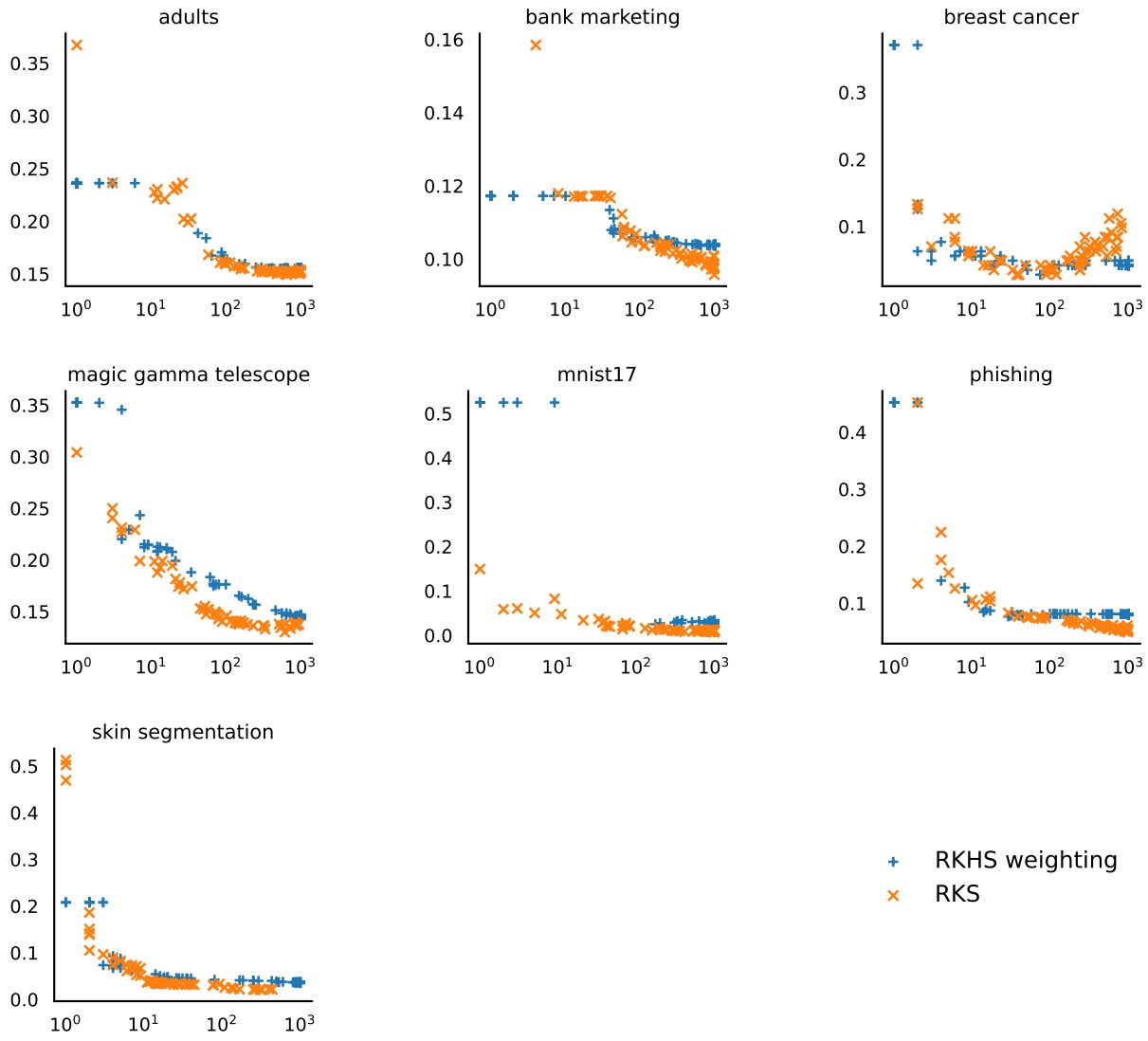

Figure 9: Comparison of the classification performance of **RKHS weightings** (learned using Algorithm 5) and random kitchen sinks (**RKS**, learned using Algorithm 2). The number of sampled features was $T = 1000$. The x-axis values are the number of features retained by the Lasso. On the y-axis is the test classification error. The RKHS weightings instantiation is RWRelu. The random kitchen sinks used the same distribution (Gaussian) and base predictor (ReLU). Each point was obtained by randomly sampling a regularization parameter for the Lasso, then selecting other parameters through cross-validation.

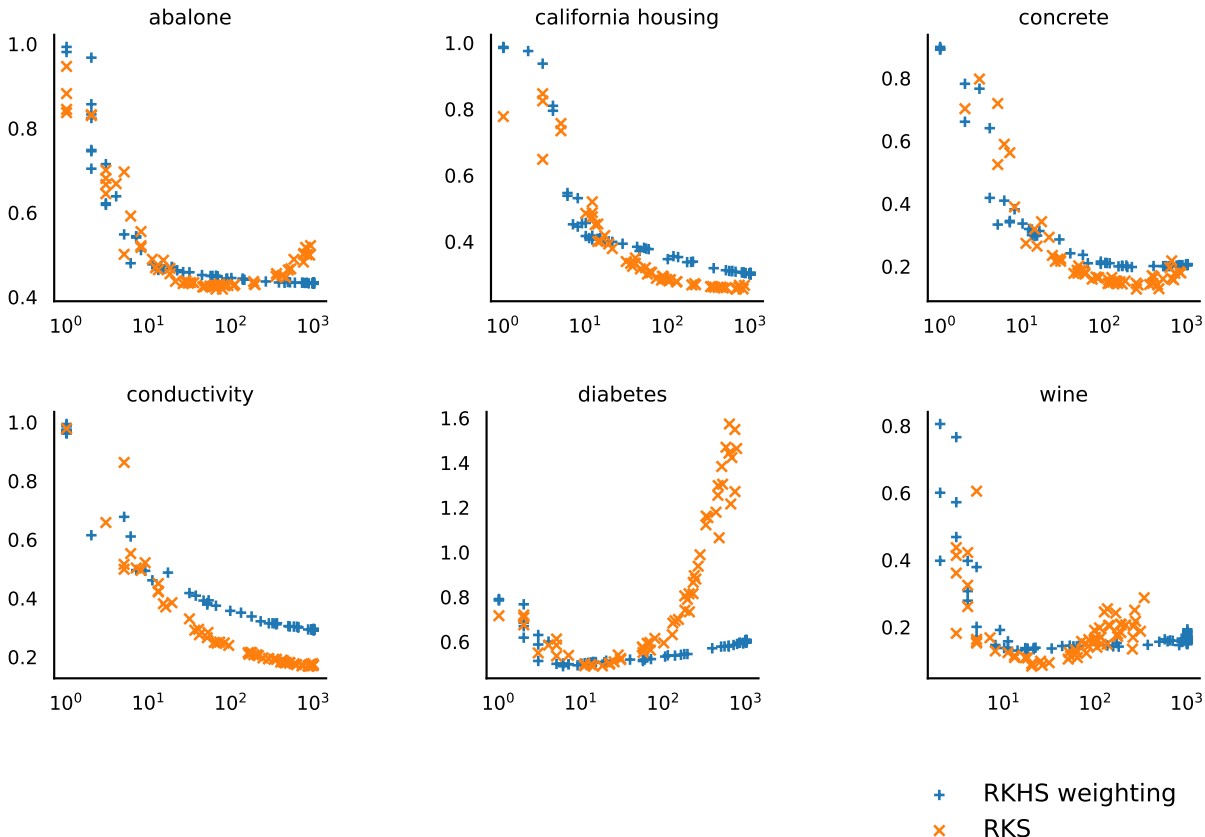

Figure 10: Comparison of the regression performance of **RKHS weightings** (learned using Algorithm 5) and random kitchen sinks (**RKS**, learned using Algorithm 2). The number of sampled features was $T = 1000$. The x-axis values are the number of features retained by the Lasso. On the y-axis is the test mean squared error. The RKHS weightings instantiation is RWRelu. The random kitchen sinks used the same distribution (Gaussian) and base predictor (ReLU). Each point was obtained by randomly sampling a regularization parameter for the Lasso, then selecting other parameters through cross-validation.

