# OpenReview forum: "Using RKHS Weight Functions in Random Feature Models"
_TMLR — Accepted by TMLR_

### Review · Reviewer_LoAR · 2026-03-02

**Summary Of Contributions:**

In this paper, the authors provide an extensive theoretical and practical analysis of “RKHS Weightings” (RKHSW), a random feature model where the weight function lies in a RKHS, both in the context of regression and classification. They provide the following contributions:
1) under this RKHS assumption, according to the choice of the weight function, they show that one can compute the exact model, and they provide many examples (Tables 2 and 3);
2) they provide learning algorithms (Stochastic Functional Gradient Descent (SFGD), Least-Squares, Lasso) and show how to choose the hyperparameters to deal with the curse of dimensionality under some circumstances;
3) they conduct a thorough theoretical analysis, based on standard assumptions on the model and the loss function, providing generalisation and PAC bounds through the Rademacher complexity theory, as well as a stability analysis of SFGD and approximation analysis of the empirical risk minimizer. They also discuss how RKHSW compare with Random Kitchen Sinks (RKS) and Square-integrable weight functions in theory;
4) they provide experiments on many real-world regression and classification datasets to compare RKHSW with RKS and other standard methods.

**Audience:**

Yes

**Audience Explanation:**

The TMLR’s audience would be highly interested in this paper’s findings, as kernel methods and random feature models are regularly the topics of TMLR published papers. The authors here provide a very interesting avenue: focusing on cases where the model can be exactly computed, partly thanks to this RKHS assumption, and studying it thoroughly. Theoretically, these models are promising as they seem more flexible and expressive that RKS. Moreover, compared to  Square-integrable weight functions theory, they provide a more flexible approximation guarantee. They also discuss the limitations of such models and leave many interesting future work directions.

**Claims And Evidence:**

Yes

**Claims Explanation:**

The claims made in the submission are supported with extensive theoretical and practical analysis, and all theoretical results are supported by well-detailed proofs. Moreover, all these results are compared with similar methods and are discussed relevantly by the authors, who also highlight the limitations of their work.

**Requested Changes:**

My main question concerns the limitation you point out about the unclear advantage over RKS that you leave as an open problem. In the conclusion, you mention that RKHSW could be relevant for interpretability purposes for instance. Could you elaborate a little more about this? And more generally, do you have more ideas to value RKHSW over RKS?

Minor changes:
- It would be nice to have a list of contributions in the end of the Introduction section.
- There is a term missing in the error decomposition in Equation 91: $\mathcal{L}_{\mathcal{D}}(\Lambda {\alpha}\_{\mathcal{S}}) - {\mathcal{L}}\_{\mathcal{D}}(\Lambda \alpha^{\star}) = \ldots$

---

> ### Author Response · Authors · 2026-04-08
> **Response to Reviewer LoAR**
>
> We thank the reviewer for their time.
>
>
> On the subject of interpretability, RKHS weightings, and likewise RKS, are additive models, which directly contributes to them being interpretable. Dubé & Marchand (2025) have built interpretable RKHS weightings based on an earlier version of this submission. (Note that interpretable RKS can be built in the same way, so this is not a difference between both methods.) Since the major distinction between both model formulations is the kernel $K$, we surmise that clever instantiation of the model (and especially of the kernel) could yield a model which is interpretable in a different way. Similarly, we think that it could be possible to use prior knowledge about a given problem in order to choose a relevant kernel. This would restrict the space of predictors, likely improving generalization guarantees, but would not hinder performance if the prior knowledge is adequate. We have not yet examined these avenues.
>
>
> We added some of these clarifications to the text of the updated manuscript. (Section 8, paragraphs “Instantiating the model” and “Unclear advantage over random kitchen sinks”. )
>
>
> We have also corrected Equation 91, and added an explicit list of contributions to the introduction.

---

### Review · Reviewer_MVbv · 2026-03-08

**Summary Of Contributions:**

This paper studies a variant of random feature models where the weight function in the representation $f(x) = \mathbb{E}_{w\sim p}[\alpha(w)\phi(w,x)]$ is assumed to lie in a reproducing kernel Hilbert space (RKHS). Under this assumption, the authors show that certain model instantiations allow the expectation to be computed analytically, avoiding the need for Monte Carlo approximation typically used in random kitchen sinks methods. The paper proposes three algorithms to learn the weight function, including a stochastic functional gradient descent method operating directly in the RKHS. The authors provide theoretical guarantees on generalization and convergence, showing that the excess risk scales as $O(\sqrt{\frac{1}{m}}+ \sqrt{\frac{1}{T}})$, matching the classical guarantees for random kitchen sinks. Finally, several instantiations of the model are presented together with numerical experiments comparing the proposed method with random kitchen sinks, SVMs, and AdaBoost.

**Audience:**

Yes

**Audience Explanation:**

Random feature models and kernel approximations remain an active topic in machine learning theory. The paper proposes an interesting perspective by placing an RKHS structure on the weight function of the random feature model, which connects functional optimization, kernel methods, and random features in a unified framework. Researchers interested in theoretical aspects of kernel methods, random feature models, and functional optimization may find the results relevant.

**Broader Impact Concerns:**

None.

**Claims And Evidence:**

Yes

**Claims Explanation:**

The paper presents a mathematically rigorous framework and several theoretical results supporting the proposed model class and learning algorithms. The theoretical analysis appears technically sound and follows established tools from RKHS theory and statistical learning theory. However, some claims about the practical advantages of the approach are not fully substantiated. In particular, the empirical evaluation is limited and does not clearly demonstrate when the proposed RKHS weighting approach provides significant advantages over standard random feature methods. Additionally, certain aspects of the methodology—such as the computational complexity and scalability of the stochastic functional gradient descent algorithm—would benefit from further clarification.

**Requested Changes:**

Major comments:

1. The proposed learning procedures appear to rely heavily on existing ideas from stochastic optimization and random feature learning. In particular, the stochastic functional gradient descent algorithm closely resembles standard stochastic gradient methods in RKHS spaces. The reviewer would appreciate a clearer discussion of what algorithmic insights are genuinely new in this work, as opposed to adaptations of existing techniques applied to the proposed model formulation.

2. The stochastic functional gradient descent algorithm involves maintaining an RKHS representation and performing projection steps that depend on the RKHS norm. Though the authors mentioned the dependency of the computational cost on the number of random features (which is similar to the least squares case), there's still an explicit gap in the simulation. It is also beneficial to discuss the dependency on other parameters and make a comparison with standard random feature training methods.

3. A key motivation of the paper is that, under certain instantiations, the expectation $\mathbb{E}_{w\sim p}[\alpha(w)\phi(w,x)]$ can be computed analytically. However, the examples provided seem somewhat specialized. It would strengthen the paper if the authors could discuss how broadly applicable this property is, and whether such analytical forms exist for commonly used feature maps and kernels in practice.

4. The experimental evaluation is relatively limited. The datasets appear small and the comparisons focus mainly on classical algorithms such as SVM and AdaBoost. Additional experiments on larger datasets or more challenging prediction tasks would help demonstrate the practical usefulness of the proposed method. In particular, it would be helpful to analyze how the performance depends on the number of sampled features and the RKHS parameters.

5. The paper provides a thoughtful discussion of the curse of dimensionality and proposes a heuristic method for choosing hyperparameters based on the constants appearing in the theoretical bounds.

To further strengthen this part, it would be useful if the authors could comment on:
 * How sensitive the model performance is to these hyperparameters in practice,
 * Whether the proposed parameter selection strategy was used in the experiments

6. The related work section provides a useful overview of the connections with Rahimi & Recht (2009) and Bach (2017). Since the proposed framework introduces a stronger structural assumption on the weight function, it would be helpful if the authors could more explicitly summarize the trade-offs between these formulations. For instance, a table comparing:

* Expressiveness of the hypothesis classes
* Assumptions on the weight function
* Resulting theoretical guarantees

This could help readers better understand the big picture and relative advantages of the RKHS-weighted formulation.


Minor comment:

1. For the case of diabetes in Figure 3, it is counter intuitive that the performance of RKHS weighting is getting worse as the number of random features goes up.

---

> ### Author Response · Authors · 2026-04-08
> **Response to Reviewer MVbv**
>
> We thank the reviewer for their thorough review of our paper. We submitted a revised manuscript containing changes which we hope address the reviewer’s concerns. Specifically:
>
>
> 1. We added a Section 5.4 explaining the differences between SFGD and the work of Dai et al. (2014).
>
>
> 2. We added detailed breakdowns of the algorithmic complexities (Tables 5 and 6 of the new manuscript). We added a new figure (Figure 2 of the new manuscript) which more clearly shows the similar complexities of algorithms SFGD and Least squares fit.
>
>
> 3. For the moment, obtaining the analytical expectation for a new instantiation requires time-consuming calculus. We have not yet found a simple method or rule to calculate these expectations more broadly than for the instantiations we considered in the paper. We mean to leave for future work the task of expanding the repertoire of analytical instantiations.
>
>
> 4. Our experiments are limited in scope since the principal goal of this paper is to lay the theoretical foundation for RKHS weightings. The experiments are there to assess the soundness of the method. Since the results are fairly unimpressive at the moment, we did not deem necessary to extend the comparison to more datasets or algorithms, as we do not think that a clear advantage will emerge without further work to improve the method. We do not claim in this paper that RKHS weightings yet have a clear practical advantage over other methods.
>
>
> Additionally, Figures 2 and 3 of the submission (5 and 6 in the new manuscript) show the performance with regard to the number of sampled features. Could the reviewer clarify what they mean by  “analyze how the performance depends on the number of sampled features”?
>
>
> Finally, see below for our comments on the subject of the RKHS parameters.
>
>
> 5. The performance is highly sensitive to the choice of kernel and distribution parameters. We added an experiment to illustrate this (Section 7.2, Figures 3 and 4 of the new manuscript). In short, only a limited window of values can be used successfully, and our proposed parameter selection strategy (which was used in our experiments) can automatically find values within that window.
>
>
> 6.  We have made the requested table (Table 7 of the new manuscript).
>
>
> Finally, the reviewer commented on the case of diabetes in Figure 3. We assume that simple overfitting explains the observed behavior. We surmise that the cross-validation parameters, which were the same for all datasets, were not quite adequate for diabetes. However, since the test set has now been seen, we cannot rerun the experiments with different parameters, as that would introduce test set leakage.

---

### Review · Reviewer_mtSf · 2026-04-01

**Summary Of Contributions:**

This paper proposes to analyze the random features model under the assumption that the prediction model coefficients belong to an RKHS associated with a kernel function on random features. This formulation is a little different to prominent random features literature from Rahimi & recht 2007 and Bach 2017. So, this is a novel setting specifically for random features models (assuming parameters being a function in a Hilbert space is, however, a common practice in functional analysis). This formulation has allowed the authors to derive a suite of associated theoretical and algorithmic results, from optimization methods (e.g., functional gradient descent, among others) to convergence and generalization guarantees. Empirical evidence is provided to show that learning a random features model under this perspective yield on par and sometime better prediction performance compared to the basic RKS algorithm.

Overall, I have a mix feeling about this work, as I believe some of the derivations in this work could be valuable to the community, yet the main message, narrative, and implications are not well-supported and well-motivated.

Strength:
1. The paper does a good job in introducing the necessary and relevant mathematical background.
2. Formulating the parameters of random features model as functions in an RKHS, to my best knowledge, is new in the realm of random features models (however, the setup is standard in functional analysis as I have mentioned ealier).
3. The analytical results regarding the new random features formulation are rather comprehensive, where the authors presented a full suite of results, from algorithmic optimization to convergence and generalization guarantees. The authors also provided a selection analytical examples for their proposed formulation in Table 2 and 3.
4. Overall, I would say the paper is well-written in its mathematical reasoning and quite easy to follow.

Weakness:
I will organize my thoughts into three categories. Main: something that I feel is necessary to be address or clarified before being considered for publication. Technical: Non-critical technical problems, including suggestions and statements of my view of the work (not necessarily addressable), and Structural: Presentation problems, with which I think could help improving the readability of the paper.

Main:
1. There seems to be a factor of 2 missing in equation (70), where the gradient of alpha hilbert norm is 2 alpha, see Lemma 2.2. This does not make the following results invalid as the factor can be absorbed into \lambda_H in principle, but the missing factor does propagate and impact several following derivations. This issue needs to be fixed.
2. The statement of K is universal leads to dense H in L^2(P) statement is a little unclear to me, as the authors definition of universal kernel (above Lemma 6.3) is a little different from my understanding, which requires a compact domain. Maybe the authors could correct my understanding here, but this might create some issue for Lemma 6.3 and the following results.
3. The title feels a bit too broad relative to the actual scope of the paper. “RKHS weighting of functions” suggests a general framework applicable to arbitrary function classes, whereas the paper focuses specifically on random feature models. I understand this may be for simplicity, but as it stands, the title gives the impression of a more general result than what is actually developed. It might be worth either narrowing the title or clarifying the scope more explicitly early on to avoid misleading readers.
4. Given the current length of the paper and the empirical behavior of SFGD, I am not fully convinced about the necessity of including it as a main component. SFGD itself is not a new algorithm, and in this setting it appears to have relatively slow convergence (as seen in Figure 1), which also tends to compress the performance differences among other methods. More importantly, several key results in Section 6 (e.g., Theorems 6.7 and 6.8) depend on the SFGD-based construction and its convergence. Since SFGD performs poorly in practice under the current design, it makes the motivation and practical interpretation of these results less clear. In particular, it is not obvious how these guarantees translate to the more effective methods considered in the experiments. It would help to either better justify the role of SFGD in the overall framework.
5. This is more of a conceptual question which I would appreciate the authors could clarify. Conceptually, the proposed formulation seems to transform a random feature model back into a kernel method, by inducing a new kernel on the input space through an RKHS structure over the feature weights. While this construction is mathematically valid, it raises the question of whether the framework is introducing a genuinely new modeling approach, or simply defining a new kernel in a more indirect way (Kernel Regression (KR) --- RF approximation --- RKHS Weighting --- KR with a new induced kernel). I believe clarifying this question would also help clarifying the fundamental contribution of this new perspective.

Technical:
1. A major technical concern I have is the practical relevance of the proposed formulation. Random feature models are typically used for approximation and computational efficiency, especially due to their simplicity and parallelizability. In contrast, the RKHS weighting formulation introduces additional complexity, such as gradient-based optimization in function space or numerical integration, which appears to reduce these practical advantages. At the same time, the paper does not seem to provide new insights into the approximation or convergence properties of random feature models (e.g., no improvement in order-wise rates). This makes it unclear what is gained theoretically. From a practical standpoint, the empirical motivation in Section 7.4 is also not fully convincing. If the goal is improved performance, one could often achieve similar or better results by simply increasing the number of random features in standard RKS, without incurring the additional training complexity introduced here. Given this, I do not yet see a clear practical incentive for adopting the proposed formulation. It would be helpful if the authors could more clearly articulate the intended use cases or scenarios where RKHS weighting provides a concrete advantage over standard random feature approaches, rather than presenting it primarily as an alternative formulation.
2. The choice of K_w is not very clear to me. While Table 2 lists several (phi, K_w) pairs, it seems that the main criterion is whether the expectation in Table 3 admits a closed-form expression. This makes the construction feel driven more by analytical convenience than by a clear modeling principle. Relatedly, it is unclear whether standard random Fourier features (e.g., cosine features used to approximate the Gaussian kernel) can be incorporated into this framework. If not, is it because there is no suitable K_w that leads to a tractable expression? Clarifying this would help better understand the generality and limitations of the proposed approach.
3. Given the relatively small performance differences in Tables 7 and 8, it would be helpful to report standard deviations over the 10 runs, especially considering the randomness in random feature models. Additionally, I am not sure about the usefulness of including the numerical values of the Theorem 6.2 bounds in these tables. As presented, these bounds appear quite loose and do not seem to provide meaningful insight into the empirical results. It might be more helpful to either omit them or better explain how they should be interpreted in practice.
4. This go back to main weakness 4. The design of SFGD is somewhat unclear to me. In particular, it is not clear why only a single random feature is sampled at each iteration. As shown in Figure 1, this leads to relatively slow convergence. It seems plausible that there is a tradeoff between the number of random features used per iteration and the total number of iterations (e.g., using mini-batches of random features). Exploring this tradeoff or providing justification for the current design would help clarify the efficiency of the method.


Structural:
1. Section 6 is quite dense and not very well organized. Many results are introduced in sequence without sufficient follow-up interpretation, which makes it difficult to understand their role and significance. While I understand that some of these results are intermediate lemmas used for later theorems, it might improve readability to move more of these technical details (including some proofs) to the appendix. This would allow the main text to focus more on the key ideas and takeaways.
2. It would be helpful if the authors could include a dedicated paragraph or short section early in the paper (e.g., at the end of Section 3.1) to clearly explain the scope of novelty of this work. Currently, the main differentiating idea appears only in Section 4.1 and has to be inferred by the reader. Making this more explicit upfront would significantly improve clarity and positioning.
3. There are some minor presentation issues including sentences and notations. For example, line 2 of Section 4.5, "That assumed that we were minimizing...". The norm in Eq (52) is not properly defined. Around Eq. (60), it would be helpful to clarify that \alpha_S \in H_S.
4. Number of random features used in the simulations of Section 7.2 are not mentioned. This should be a key issue given the results in Section 7.4.
5. There is red text "New Section" in Section 7.4 title.

**Audience:**

Yes

**Audience Explanation:**

The topic of random features model and Hilbert space analysis for function parameters should be of interest to TMLR community.

**Broader Impact Concerns:**

I do not see any concern on broader impact.

**Claims And Evidence:**

No

**Claims Explanation:**

While most of the individual results appear technically correct, there are several key issues (see main weaknesses) that affect the overall motivation of the work. In particular, these drawbacks make the theoretical and algorithmic contributions feel insufficiently justified in the broader context.

Moreover, the experimental results do not clearly demonstrate the practical relevance of the proposed formulation. They appear largely disconnected from the theoretical developments, aside from reporting bound values, and it is unclear how the theory translates into practical advantages.

**Requested Changes:**

Please see main weaknesses.

---

> ### Author Response · Authors · 2026-04-08
> **Response to Reviewer mtSf**
>
> We thank the reviewer for their thorough and detailed review of our paper. We submitted a new manuscript addressing many of the reviewer’s concerns. Below are our answers to each of these concerns.
>
>
> Main
> 1. We thank the reviewer for spotting the missing factor 2. It has been added to the relevant equations in Section 5.1, and the changes were propagated through the proof of Theorem 6.7. (The theorem statement has changed very slightly.)
> 2. The manuscript has been updated to use the notion of L^p-universality, using Sriperumbudur et al. (2011) as reference.
> 3. We changed the title to “Using RKHS Weight Functions in Random Feature Models”.
> 4. The goal of this paper is to explore the consequences of using an RKHS weight function in random feature models. Since the SFGD algorithm naturally arises in this context, it is necessary to include it in our analysis and experiments, even if its performance is inferior to alternatives.
>
> Also, a recent bug fix has led to significant improvement of the performance of SFGD in Figure 1 (see the new manuscript), placing it closer to the other methods.
>
> 5. The reviewer is correct. As we explain in Section 4.5, RKHS weightings in theory boil down to a kernel method of kernel K_X, given by Equation (63) of the submission. Indeed, all random feature models approximate a kernel. RKHS weightings are no different.
>
>
>
>
> Technical comments
>    1. RKHS weightings, when using an instantiation with an analytical formula (e.g. Tables 2 and 3) and learned with the least squares fit, are almost as fast as standard random kitchen sinks, and our experiments show similar results for both methods. This is pertinent information in itself, to prevent future researchers from repeating the same experiments.
>
> In theory, RKHS weightings add structure to the weight function, which can have interesting implications. Finding those implications or advantages are outside the scope of this paper, which provides the necessary foundation for this future work.
>
>    2. The reviewer is correct that the instantiations in Tables 2 and 3 are those for which we found a closed-form solution to the integrals. We have not yet found a simple method or rule to calculate these expectations more broadly than for the instantiations we considered in the paper. We mean to leave for future work the task of expanding the repertoire of analytical instantiations.
>
> Random Fourier Features (RFF) are used to approximate shift-invariant kernels, most notably the Gaussian kernel. As the reviewer points out in their main comment #5, RKHS weightings can be seen as a kernel method, with the kernel defined by Equation 63 of the submission manuscript. If the RFF already approximates the desired kernel, then there is no need to use an RKHS weighting. From this point of view, the purpose of RKHS weightings is to get access to these more exotic kernels.
>
>    3. Tables 13 and 14 of the appendix of the submission manuscript are Tables 7 and 8 with added standard deviations. We also comment on the looseness of the bounds in the second-to-last paragraph of Section 7.2 (“As for the generalization bounds, [...]”). (Note that these table and section numbers are changed in the updated manuscript.)
>    4. The updated manuscript contains a new Section 5.1.1 explaining why Algorithm 3 samples only one random feature at each iteration. We added new Figures 7 and 8 to the appendix in the new manuscript to supplement the argument.
>
>
> Structural comments
> 1. We streamlined Section 6.3 by moving both intermediate lemmas to the appendix.
>
> Streamlining Section 6.1 is more difficult, as most results are referenced elsewhere in the paper. We added a small paragraph at the end of Section 6.1 to help the reader understand the purpose of that section’s results.
>
> 2. The new manuscript has a list of contributions added to the introduction.
>
> 3. For the norm in Equation (52), the norm of an operator is defined in Equation (8) of the submission manuscript.
>
> The  first paragraph of Section 4.5 and lead-up to Equation 60 have been adjusted in the new manuscript.
>
> 4. The number of random features in the experiments of Section 7.2 is stated in the Tables 7 and 8 captions: “The sampling size T was 500…”. (Note that these section and table numbers are changed in the updated manuscript.)
>
> 5. The red text has been removed in the new manuscript.

---

### Decision · Action_Editor_Kmy6 · 2026-06-12

**Recommendation:** Accept as is

**Audience:**

Yes

**Audience Explanation:**

Random Feature models have a long history now in Machine Learning, and there are still members of the audience who will be interested in the findings of this paper.

**Claims And Evidence:**

Yes

**Claims Explanation:**

Both theoretical and empirical claims are supported by mathematical proofs and experimental evaluations.